# INTRINSIC TRAINING DYNAMICS OF DEEP NEURAL NETWORKS

**Sibylle Marcotte**
ENS - PSL Univ.
sibylle.marcotte@ens.fr

**Gabriel Peyré**
CNRS, ENS - PSL Univ.
gabriel.peyre@ens.fr

**Rémi Gribonval**
Inria, CNRS, ENS de Lyon,
Université Claude Bernard Lyon 1,
LIP, UMR 5668, 69342, Lyon cedex 07, France
remi.gribonval@inria.fr

## ABSTRACT

A fundamental challenge in the theory of deep learning is to understand whether gradient-based training can promote parameters belonging to certain lower-dimensional structures (e.g., sparse or low-rank sets), leading to so-called implicit bias. As a stepping stone, motivated by the proof structure of existing implicit bias analyses, we study when a gradient flow on a parameter $\theta$ implies an intrinsic gradient flow on a "lifted" variable $z = \phi(\theta)$, for an architecture-related function $\phi$. We express a so-called intrinsic dynamic property and show how it is related to the study of conservation laws associated with the factorization $\phi$. This leads to a simple criterion based on the inclusion of kernels of linear maps, which yields a necessary condition for this property to hold. We then apply our theory to general ReLU networks of arbitrary depth and show that, for a dense set of initializations, it is possible to rewrite the flow as an intrinsic dynamic in a lower dimension that depends only on $z$ and the initialization, when $\phi$ is the so-called path-lifting. In the case of linear networks with $\phi$, the product of weight matrices, the intrinsic dynamic is known to hold under so-called balanced initializations; we generalize this to a broader class of *relaxed balanced* initializations, showing that, in certain configurations, these are the *only* initializations that ensure the intrinsic metric property. Finally, for the linear neural ODE associated with the limit of infinitely deep linear networks, with relaxed balanced initialization, we make explicit the corresponding intrinsic dynamics.

## 1 INTRODUCTION

A central question in deep learning theory is how the complexity of gradient-based training can give rise to simpler, lower-dimensional dynamics. In this work, we explore when the gradient flow on parameters $\theta$ naturally induces a gradient flow on a "lifted" variable $z = \phi(\theta)$, where $\phi$ captures structural aspects of the model.

**Intrinsic lifted flow.** The study of optimization flows arising in the training of neural networks often benefits from the identification of lower-dimensional intrinsic dynamics. Specifically, due to the natural symmetries of linear and ReLU networks, it is of considerable interest to rewrite a parameter flow $\theta(t)$ in terms of an representation $z(t) = \phi(\theta(t))$, using a suitable architecture-related reparametrization $\phi$ (often called a *lifting*) that factors out certain symmetries.

When dissected, the most advanced recent results characterizing the implicit bias induced by gradient-based optimization algorithms notably rely on two key analysis ingredients: (i) establishing that

the dynamics of $z(t)$ is *intrinsic*, i.e., that it can be expressed as a Riemannian gradient flow with a metric depending only on $z$ and the initial parameters $\theta(0)$; (ii) further proving that this flow on $z(t)$ admits a *mirror flow representation*. With the combination of these two ingredients one gains access to powerful analytical tools rooted in convex optimization theory, allowing explicit characterization of the induced implicit bias. In particular, prior research has successfully leveraged mirror flow formulations to rigorously demonstrate implicit regularization effects, such as sparsity in scalar linear neural networks and two-layer networks with a single neuron (Gunasekar et al., 2018), as well as maximum-margin classification for logistic regression problems in separable data scenarios (Soudry et al., 2018; Chizat & Bach, 2020).

Recent work by Li et al. (2022) identifies sufficient conditions under which (i)+(ii) can both be established, requiring that the parametrization $\phi$ be *commuting*. However, this commuting condition is rarely satisfied in practical scenarios. This work focuses on characterizing weaker conditions ensuring that the flow on $z(t)$ is still driven by an intrinsic Riemannian gradient flow (but not necessarily a mirror flow anymore), which we believe is an important step forward and a starting point for future investigations encompassing variants of (ii) with *warped* mirror flows (Azulay et al., 2021) for practical (deep) network architectures. A first sufficient condition for (i), introduced by Marcotte et al. (2023), demands merely that the parametrization be involutive. Marcotte et al. (2023) have shown that this weaker condition applies specifically to the parametrization used in *two-layer* ReLU networks (Stock & Gribonval, 2022). As we will see, a consequence of the analysis conducted in our paper is *the extension of this result to arbitrary DAG ReLU networks* (Gonon et al., 2024).

**Conservation laws.** The functions conserved during the training dynamics play a crucial role in establishing that the dynamics of $z(t)$ is governed by an (intrinsic) Riemannian metric that depends only on $z$ and the initialization $\theta_0 := \theta(0)$. Indeed, when a trajectory $\theta(t)$ is known to remain within level sets $\{\theta : \mathbf{h}(\theta) = \mathbf{h}(\theta_0)\}$ where $\mathbf{h}$ is a (vector-valued) conserved function, the dynamics are effectively restricted to a manifold of lower dimension that is entirely determined by the initialization. A particularly important class of conserved functions along these trajectories is given by the conservation laws associated with a certain architecture-dependent parametrization $\phi$, a concept introduced in Marcotte et al. (2023). These laws depend exclusively on $\phi$, and notably, in the context of neural network training dynamics, they represent quantities preserved across trajectories irrespective of the initial conditions or the training dataset. In the specific case of linear and ReLU neural networks, these conservation laws correspond exactly to previously known "canonical" conserved functions identified in Du et al. (2018); Arora et al. (2019), as demonstrated by Marcotte et al. (2023); see also Le & Jegelka (2022) for canonical conservation laws for general DAG ReLU architectures. Furthermore, Marcotte et al. (2023) establish that *if the parametrization $\phi$ is involutive, there exist sufficiently many scalar conservation laws to fully rewrite the original trajectory $\theta(t)$ in terms of $\phi(\theta(t))$ and the initial conditions alone*. In the linear network case, when so-called balanced conditions (a notion introduced in Arora et al. (2019)) are satisfied (i.e., when the initialization sets all canonical conservation laws (Chitour et al., 2018) to zero, $\mathbf{h}(\theta_0) = 0$), it becomes possible to rewrite the flow in terms of $z = \phi(\theta)$, where $\phi$ corresponds to the product of weight matrices, as an intrinsic Riemannian metric (Arora et al., 2018; Bah et al., 2022). Moreover, Achour et al. (2025) extended this result to linear *convolutional* networks in the case of a mean squared loss, but this time for arbitrary initializations, with the Riemannian metric depending on the initialization. For linear networks and in the particular case when the loss function is the square loss, Bah et al. (2022) show that the trajectory evolves on the manifold of matrices having some fixed rank under balanced condition. Still in the square-loss setting, and in the case of two-layer linear networks, Tarmoun et al. (2021); Braun et al. (2022); Dominé et al. (2025) exploit the conservation laws to obtain an exact closed-form expression for $z(t)$ under specific configurations, whereas Varre et al. (2023) use the same laws to analyse an implicit bias of this dynamic.

**Our main contributions.** We first define the notion of intrinsic *dynamic* property (Definition 2.6), then the notion of intrinsic *metric* property (Definition 2.10) and finally the one of intrinsic *recoverability* property (Definition 2.15), and we show the implications (Lemma 2.11 and Lemma 2.16):

$$\text{Intrinsic Recoverability} \implies \text{Intrinsic Metric} \implies \text{Intrinsic Dynamic}.$$

We then provide a simple criterion that characterizes the intrinsic recoverability property (Theorem 2.17), and show (Proposition 2.21) that this criterion is *quasi* equivalent to the *Frobenius property*, which is slightly weaker than involutivity (Definition 2.20). We prove that the so-called path-lifting (Gonon et al., 2024) reparametrization for general ReLU networks of arbitrary depth satisfies this property (Theorem 3.1), characterizing *a dense set of initializations of a general ReLU network*

*that satisfies the intrinsic recoverability property* (Corollary 3.3), as illustrated by a characterization of the intrinsic dynamic of a 3-layer neural network (Proposition 3.5). Next, by establishing a necessary condition for the intrinsic metric property to hold based on the study of kernels of linear mappings Theorem 2.14, we show that the *intrinsic metric property fails to hold for the natural reparametrizations corresponding to 2-layer linear networks (resp. of attention layers)*, unless the initialization satisfies the *relaxed balanced condition* introduced in Definition 3.6 (Theorem 3.9). We then show that relaxed balanced initializations do satisfy the intrinsic metric property, not only in 2-layer networks (Theorem 3.8) but also in linear networks of arbitrary depth (Theorem 3.11), and we characterize the resulting intrinsic dynamic. Finally, we extend our analysis to the infinite-depth limit of linear networks. We show that a set of functions is conserved along the trajectory (Proposition 3.12), and, in contrast to the case of $L > 2$-layer, we derive a closed-form expression for the metric in the case of relaxed balanced initializations (Theorem 3.13).

## 2 DYNAMICS OF OVER-PARAMETERIZED MODELS

In most machine learning models, overparameterization occurs due to inherent symmetries (such as rescaling) within the parameter space $\theta \in \mathbb{R}^D$. In practice, this redundancy can be factored out through a function $\phi$ (often called a lifting (Candès et al., 2013; Gonon et al., 2024)) that captures these symmetries. Although the resulting lifted variable $z = \phi(\theta) \in \mathbb{R}^d$ often lives in higher dimension $d \gg D$, it also belongs to a lower dimensional manifold $\mathcal{Z}$ of dimension $d' < D$, and provides a representation of the essential structure of the model. We consider parameters $\theta(t) \in \mathbb{R}^D$ that satisfy the **gradient flow** dynamic with some initialization $\theta_0$:

$$\dot{\theta}(t) = -\nabla \ell(\theta(t)), \quad \theta(0) = \theta_0 \tag{1}$$

to minimize the function $\ell$. In machine learning, $\ell(\theta)$ is typically defined as the empirical average over training samples $(x_i, y_i)$ of a quantity that depends on the output $g(\theta, x_i)$ of a neural network with weights and biases collected in the parameter vector $\theta$. The function $g(\theta, x)$ can be locally reparameterized via the above architecture-dependent lifting $\phi(\theta)$, leading to the same factorization for the global loss $\ell$. This is the starting point of our analysis, captured via the following assumption:

**Assumption 2.1** (Local reparameterization of $\ell$ via $\phi \in \mathcal{C}^2(\mathbb{R}^D, \mathbb{R}^d)$ on an open set $\Omega \subseteq \mathbb{R}^D$)**.** There is a function $f \in \mathcal{C}^2(\Omega, \mathbb{R})$ such that

$$\forall \theta \in \Omega, \quad \ell(\theta) = f(\phi(\theta)). \tag{2}$$

The following examples illustrate common choices of $\phi$ for various neural network architectures.

**Example 2.2** (Linear neural networks)**.** For a two-layer network with $r$ hidden neurons and $\theta = (U, V) \in \mathbb{R}^{n \times r} \times \mathbb{R}^{m \times r}$ (where $D = (n + m)r$), the model $g(\theta, x) := UV^\top x$ is factorized via the map $\phi_{\mathtt{Lin}}(\theta) := UV^\top \in \mathbb{R}^{n \times m}$, thus the empirical risk $\ell$ can also be factorized by $\phi_{\mathtt{Lin}}$. This extends to $L$ layers where $\theta = (U_L, \cdots, U_1)$, with $g(\theta, x) := U_L \cdots U_1 x$ and $\phi_{\mathtt{Lin}}(\theta) := U_L \cdots U_1$. The resulting factorization of $\ell$ holds globally on $\Omega = \mathbb{R}^D$.

**Example 2.3** (ReLU neural networks)**.** Consider $g(\theta, x) = U\sigma(V^\top x)$, with $\sigma(y) := (\max(y_i, 0))_i$ the ReLU activation function. Denote $\theta = (U, V)$ with $U = (u_1, \cdots, u_r) \in \mathbb{R}^{n \times r}$, $V = (v_1, \cdots, v_r) \in \mathbb{R}^{m \times r}$ (so that $D = (n+m)r$). Given training vectors $x_i \in \mathbb{R}^m$ consider $\Omega \subseteq \mathbb{R}^D$ the set of all parameters $\theta = (U, V)$ such that $v_j^\top x_i \neq 0$ for every $i, j$. Then, $\theta \mapsto 1(v_j^\top x > 0) = \epsilon_{j,x}$ is locally constant over $\theta \in \Omega$, and the model $g_\theta(x)$ can be factorized by the reparametrization $\phi_{\mathtt{ReLU}}(\theta) = (u_j v_j^\top)_{j=1}^r$ (here $d = rmn$) using $g(\theta, x) = \sum_j \epsilon_{j,x} \phi_j x$ with $\phi_j(\theta) := u_j v_j^\top$, so $\ell$ can be factorized by $\phi_{\mathtt{ReLU}}$ with some forward function $f$: the reparametrization $\phi_{\mathtt{ReLU}}(\theta)$ contains $r$ matrices of size $n \times m$ (each of rank at most one, so in particular one has $d' \leq D - r$) associated to a "local" $f$ valid in a neighborhood of $\theta$. A similar factorization is possible for deeper ReLU networks (cf Neyshabur et al. (2015); Stock & Gribonval (2022); Gonon et al. (2024)) and we still write it $\phi_{\mathtt{ReLU}}$, as further discussed in the proof of Theorem 3.1.

**Example 2.4** (Attention layer)**.** For an attention layer, the input $X \in \mathbb{R}^{N \times \dim}$ is the vertical stacking of $N$ tokens $x^{(i)} \in \mathbb{R}^{\dim}$. The layer output is

$$g(\theta, X) = \mathrm{softmax}(XQ^\top KX^\top)XV^\top O \in \mathbb{R}^{N \times \dim} \quad \text{with} \quad \big(\mathrm{softmax}(A)\big)_{ij} := \frac{\exp(A_{ij})}{\sum_{k=1}^N \exp(A_{ik})},$$

with $Q, K, V, O \in \mathbb{R}^{d_1 \times \dim}$. We use the reparameterization $\phi_{\mathtt{Att}}(\theta) := (\phi_1, \phi_2)$ where $\phi_1 := Q^\top K$ and $\phi_2 := V^\top O$, such that $g(\theta, X) = \mathrm{softmax}(X\phi_1 X^\top)X\phi_2$, as done by Marcotte et al. (2025).

Thus, similarly to the linear case Example 2.2, $\ell$ can be globally factorized by $\phi_{\mathtt{Att}}$ as $f$ exhibits no dependency on the specific parameter configuration $\theta_0$. This naturally extends to multiple attention layers by concatenating the corresponding factorizations.

## 2.1 Dynamics of lifted parameters: to be or not to be intrinsic?

This paper addresses a fundamental question underlying much of the efforts to characterize the implicit bias of gradient-based methods: under what conditions does the gradient flow dynamics equation 1 in parameter space $\theta$ lead to a dynamics on the lifted parameters $z(t) := \phi(\theta(t))$ that can be expressed as an intrinsic gradient flow on $z$? This is crucial when attempting to establish that $z(t)$ follows a mirror flow (Gunasekar et al., 2017), which is a key step to characterize the implicit bias of gradient-based optimization. We specifically examine when $z(t)$ follows a flow with respect to a Riemannian metric which, by definition depends only on $z$ (and the initial parameter configuration $\theta_0$), thereby eliminating explicit dependence on the parameter trajectory $\theta(t)$.

A starting point of the analysis is that, under Assumption 2.1 and by the chain rule when $\theta(t) \in \Omega$

$$\dot{z}(t) = \partial\phi(\theta(t))\dot{\theta}(t) = -\partial\phi(\theta(t))\partial\phi(\theta(t))^\top \nabla f(z(t)). \tag{3}$$

Thus our goal is to understand when the symmetric, positive semi-definite matrix

$$M(\theta) := \partial\phi(\theta)\partial\phi(\theta)^\top \tag{4}$$

(corresponding to the so-called *path kernel* (Gebhart et al., 2021) when $\phi$ is the path-lifting associated to ReLU networks) can be solely expressed in terms of $z$ and $\theta_0$ during the trajectory, i.e. do we have a function $K = K_{\theta_0}$ such that $M(\theta(t)) = K(z(t))$? When this is the case equation 3 becomes

$$\dot{z}(t) = -K(z(t))\nabla f(z(t)), \tag{5}$$

an ordinary differential equation (ODE) which is a Riemannian flow (Boumal, 2023) for the metric $K^{-1}(z)$ (or a sub-Riemannian flow for the pseudo-inverse $K^+(z)$ when $K(z)$ is not invertible) , hence associated to an *intrinsic* dynamic on the lifted parameters $z(t)$.

As illustrated next, rewriting $M(\theta(t))$ as a function of $z(t)$ along the trajectory $\theta(t)$ is indeed possible for simple linear networks, with a function $K(\cdot)$ that depends on the initialization $\theta_0$.

**Example 2.5** (A simple linear network). Consider $g(\theta, x) = uvx$, with $\theta := (u, v) \in \mathbb{R}_* \times \mathbb{R}^m$, and $z = \phi(\theta) = uv \in \mathbb{R}^m$ (cf Example 2.2). Then $M(\theta) = \partial\phi(\theta)\partial\phi(\theta)^\top = vv^\top + u^2 \mathrm{Id}_m$. During the trajectory $u^2 - \|v\|^2 = u_0^2 - \|v_0\|^2 =: \lambda$ (as $h(\theta) := u^2 - \|v\|^2$ is conserved (Arora et al., 2019), and as $vv^\top = u^{-2}zz^\top$ we have $(u^2)^2 - \lambda u^2 - \|z\|^2 = 0$ so that $u^2 = \frac{\lambda + \sqrt{\lambda^2 + 4\|z\|^2}}{2}$. As a result along the whole trajectory we have $M(\theta) = K_{\theta_0}(z)$ so that $z(t)$ satisfies the ODE equation 5 with

$$K_{\theta_0}(z) = \frac{2}{\lambda + \sqrt{\lambda^2 + 4\|z\|^2}}zz^\top + \frac{\lambda + \sqrt{\lambda^2 + 4\|z\|^2}}{2}\mathrm{Id}_m, \quad \forall z.$$

In particular when $m = 1$ one has $K_{\theta_0}(z) = \sqrt{(u_0^2 - v_0^2)^2 + 4z^2}$ hence $\dot{z} = -\sqrt{\lambda^2 + 4z^2}\nabla f(z)$.

See Section B for more comments on that example. In the above example the function $K_{\theta_0}$, as its notation suggests, only depends on the initialization $\theta_0$ *but not on the function $f$ such that $\ell = f \circ \phi$*. In machine learning scenarios, $f$ typically captures dependence on the training dataset. The intrinsic metric $K_{\theta_0}(z)$ thus captures parts of the dynamics of $z(t)$ due to the network architecture (via $\phi$) and of the training algorithm (the gradient flow equation 1) *irrespective of the dataset and the learning task* (of course the latter still play a role via the $\nabla f(z)$ term in the ODE $\dot{z} = -K_{\theta_0}(z)\nabla f(z)$). This motivates the introduction of the following definition.

**Definition 2.6** (Intrinsic dynamic property). $\theta_0$ verifies the *intrinsic dynamic property* with respect to $\phi$ on some open set $\Omega \ni \theta_0$, if there is $K_{\theta_0} : \mathbb{R}^d \to \mathbb{R}^{d \times d}$ such that, for every $f \in \mathcal{C}^2$, the maximal solution $\theta(\cdot)$ of equation 1 on $\Omega$ with $\ell = f \circ \phi$ satisfies $M(\theta(t)) = K_{\theta_0}(\phi(\theta(t)))$ for each $t$.

## 2.2 Conservation laws

Example 2.5 illustrates a phenomenon that we will systematically exploit in our analysis: with the typical reparameterizations $\phi$ mentioned above, there exists a vector-valued function $\mathbf{h} : \theta \mapsto \mathbf{h}(\theta) \in \mathbb{R}^N$ that is conserved along the trajectory and allows to exhibit a function $K_{\theta_0}$ such that $M(\theta(t)) = K_{\theta_0}(z(t))$ along the trajectory. As these will play a key role in our analysis we now remind the essential concepts related to *conservation laws*.

We denote $\phi_1, \cdots, \phi_d$ the $d$ coordinate functions of the reparameterization $\phi : \mathbb{R}^D \mapsto \phi(\theta) \in \mathbb{R}^d \in \mathcal{C}^\infty(\mathbb{R}^D, \mathbb{R}^d)$. Since $\phi$ yields a factorization of the loss, functions $h$ such that $\nabla h(\theta) \perp \nabla \phi_i(\theta)$ for each $i$ and each $\theta$ remain constant along the trajectory. This has been thoroughly analyzed, see e.g. Marcotte et al. (2023; 2024), using the following definition.

**Definition 2.7** (Conservation law for $\phi$). A function $h \in \mathcal{C}^1(\Omega, \mathbb{R})$ is a conservation law for $\phi$ on $\Omega$ if for any $\theta \in \Omega$ one has $\partial \phi(\theta) \nabla h(\theta) = 0$, i.e. for each $\theta \in \Omega$ and $i$, $\langle \nabla \phi_i(\theta), \nabla h(\theta) \rangle = 0$.

**Proposition 2.8.** *Under Assumption 2.1 on $\ell$, $\phi$ and $\Omega$, if $h \in \mathcal{C}^1(\Omega, \mathbb{R})$ satisfies $\partial \phi(\theta) \nabla h(\theta) = 0$ on $\Omega$, then $h$ remains constant during the trajectory $\theta(t)$ of equation 1 for any initialization $\theta_0 \in \Omega$.*

The conservation laws associated with a given parameterization $\phi$ have been almost exhaustively studied for parameterizations corresponding to linear networks, ReLU networks, and attention layers. In particular, prior work has shown that all conservation laws for $\phi$ in the cases of ReLU (cf Example 2.3) and linear (cf Example 2.2) networks (see Marcotte et al. (2023)) as well as for an attention layer (see Marcotte et al. (2025)) are captured by the following proposition (Marcotte et al., 2023). In Le & Jegelka (2022), the authors also give canonical conservation laws for general DAG ReLU architectures. We recall these conservation laws in Lemma F.9. This has been proven theoretically for two-layer networks and empirically validated for deeper architectures using symbolic computation (see Marcotte et al. (2023)). One of the contributions of this paper (see Corollary 3.4 and Corollary F.15) is to establish the completeness of conservation laws for $\phi$ associated with any DAG ReLU network. It is worth noticing that all conservation laws in such cases are polynomials.

**Proposition 2.9** (Conservation laws for classical $\phi$ on $\mathbb{R}^D$). *Consider $\theta = (U_L, \cdots, U_1)$ and $\phi_{\mathtt{Lin}}(\theta) := U_L \cdots U_1$ from Example 2.2 (resp. $\phi_{\mathtt{ReLU}}$ from Example 2.3). The functions*

$$\mathbf{h}_i : \theta \mapsto U_{i+1}^\top U_{i+1} - U_i U_i^\top \text{ (resp. } \mathbf{h}_i : \theta \mapsto \mathrm{Diag}(U_{i+1}^\top U_{i+1} - U_i U_i^\top))$$

*are conservation laws for $\phi_{\mathtt{Lin}}$ (resp. $\phi_{\mathtt{ReLU}}$). Similarly, considering $\theta := (Q, K, V, O)$ and $\phi_{\mathtt{Att}}$ from Example 2.4, $\mathbf{h} : \theta \mapsto (QQ^\top - KK^\top, VV^\top - OO^\top)$ is a set of conservation laws for $\phi_{\mathtt{Att}}$.*

### 2.3 INTRINSIC DYNAMICS VIA CONSERVATION LAWS

Given conservation laws $\mathbf{h}(\theta)$ for $\phi$, any trajectory $\theta(t)$ satisfying equation 1 remains at all times on the set

$$\mathcal{M}_{\theta_0} := \{\theta : \mathbf{h}(\theta) = \mathbf{h}(\theta_0)\}, \tag{6}$$

determined by $\theta_0$. This holds true *for any function $f$ such that $\ell = f \circ \phi$* (hence, in machine learning: for any task/loss and any dataset, provided that the network model is (locally) factorized via $\phi$).

To establish the existence of a function $K_{\theta_0}(\cdot)$ such that $M(\theta(t)) = K_{\theta_0}(z(t))$ on the whole trajectory, a natural relaxation is thus to establish a related equality on the whole set $\mathcal{M}_{\theta_0}$ rather than only on a specific trajectory. This leads to the following definition and its immediate consequence.

**Definition 2.10** (Intrinsic metric property). We say that $\theta_0$ verifies the *intrinsic metric property* with respect to $\phi$ on an open set $U \ni \theta_0$ if there exists conservation laws $\mathbf{h}(\theta) \in \mathbb{R}^N$ for $\phi$ and a function $K_{\theta_0} \in \mathcal{C}^1(\mathbb{R}^d, \mathbb{R}^{d \times d})$ such that $M(\theta) = K_{\theta_0}(\phi(\theta))$ for each $\theta \in \mathcal{M}_{\theta_0} \cap U$.

**Lemma 2.11.** *If $\theta_0$ verifies the intrinsic metric property 2.10 on $U$ with respect to $\phi$, then it also verifies the intrinsic dynamic property 2.6 on $U$ with respect to $\phi$.*

*Remark* 2.12. It is not difficult to check on all examples considered in this paper that if $\theta_0$ satisfies the intrinsic metric property with respect to $\phi$ on *some* open set $U$, then any $\theta_0' \in \mathcal{M}_{\theta_0}$ also satisfies the property on a properly modified open set $U'$, with the same function $K$. This function thus only depends on $\mathbf{h}(\theta_0)$, and we denote it $K_{\mathbf{h}(\theta_0)}$ when needed to highlight this fact.

*Remark* 2.13. Lemma 2.11 remains valid with a slightly weakened version of Definition 2.10, where $K_{\theta_0}$ is not required to be smooth. Yet, since the existence of a smooth solution to the resulting intrinsic ODE equation 5 is simplified when $K_{\theta_0}$ is $\mathcal{C}^1$, we chose to include this in the definition.

The following theorem (proved in Section C) establishes a necessary condition for the intrinsic metric property to hold. We use it to show that the property *does not always* hold for linear networks.

**Theorem 2.14.** *Consider $\mathbf{h} \in \mathcal{C}^1(\mathbb{R}^D, \mathbb{R}^N)$, $\phi \in \mathcal{C}^2(\mathbb{R}^D, \mathbb{R}^d)$, and $\theta_0 \in \mathbb{R}^D$ such that the matrix $\partial \mathbf{h}(\theta) \in \mathbb{R}^{N \times D}$ has constant rank on $\mathcal{M}_{\theta_0} \cap U$ with $U \ni \theta_0$ an open subset of $\mathbb{R}^D$ and $\mathcal{M}_{\theta_0} := \mathbf{h}^{-1}(\{\mathbf{h}(\theta_0)\})$. Then (i) $\implies$ (ii), where*

*(i) There exists an open set $O \supset \phi(\mathcal{M}_{\theta_0} \cap U)$ and a map $K_{\theta_0} \in \mathcal{C}^1(O, \mathbb{R}^{d \times d})$ such that:*

$$M(\theta) = K_{\theta_0}(\phi(\theta)), \qquad \forall \theta \in \mathcal{M}_{\theta_0} \cap U;$$

*(ii)*  $$\ker \partial \phi(\theta) \cap \ker \partial \mathbf{h}(\theta) \subseteq \ker \partial M(\theta), \quad \forall \theta \in \mathcal{M}_{\theta_0} \cap U. \tag{7}$$

A trivial case where equation 7 holds is when the intersection of kernels on the left hand side is zero:

$$\ker \partial \phi(\theta) \cap \ker \partial \mathbf{h}(\theta) = \{0\}. \tag{8}$$

This stronger assumption can in fact be shown to imply the intrinsic metric property (see Theorem 2.17 in the upcoming section), and we will show (cf Corollary 3.3) that, with $\phi_{\texttt{ReLU}}$ associated to general DAG (directed acyclic graph) ReLU networks of any depth, there exists a set of conservation laws such that equation 8 indeed holds *for any initialization*. This implies the intrinsic metric property and therefore the intrinsic dynamic property irrespective of the initialization for ReLU networks with $\phi_{\texttt{ReLU}}$.

For linear networks with more than one hidden neuron, equation 8 *does not hold* (as a consequence of Steps 2,3 in the proof of Theorem 3.9), and we will show that equation 7 *does not always hold*. Nevertheless, certain initializations, known as balanced conditions (Arora et al., 2019), are known to satisfy the intrinsic dynamic property with respect to the reparametrisation $\phi_{\texttt{Lin}}$ (cf (Arora et al., 2018, Theorem 1), (Bah et al., 2022, Lemma 2)). In this paper, we generalize this result to so-called *relaxed balanced initializations* (see Definition 3.6). Moreover, we show that in certain configurations, relaxed balanced initializations *are exactly the only ones that satisfy the intrinsic metric property* (cf Theorem 3.8 and Theorem 3.9).

## 2.4 INTRINSIC RECOVERABILITY

In this section we consider a stronger condition called *intrinsic recoverability property* which requires not only that $M(\theta(t))$ can be rewritten as a function of $z(t)$ and the initialization, but also that at each point of the trajectory $\theta(t)$ itself can be fully expressed in terms of $z(t)$ and the initialization $\theta_0$. In other words, in this scenario, $\theta(t)$ can be *completely recovered* from the parameterization $\phi$ and the initialization alone, hence the name. As we will establish, this apparently strong property indeed holds when equation 8 is satisfied, which is always the case for ReLU networks.

### 2.4.1 INTRINSIC RECOVERABILITY IMPLIES INTRINSIC METRIC

**Definition 2.15** (Intrinsic recoverability property). We say that $\theta_0$ verifies the intrinsic recoverability property on an open set $U \ni \theta_0$ with respect to $\phi$, if there exists conservation laws $\mathbf{h}(\theta) \in \mathbb{R}^N$ for $\phi$ and a function $\Gamma(\cdot) \in \mathcal{C}^1(\mathbb{R}^d \times \mathbb{R}^N, \mathbb{R}^D)$ such that $\theta = \Gamma(\phi(\theta), \mathbf{h}(\theta))$ for each $\theta \in U$.

When this property holds, each $\theta \in \mathcal{M}_{\theta_0}$ satisfies $M(\theta) = M[\Gamma(\phi(\theta), \mathbf{h}(\theta))] = M[\Gamma(\phi(\theta), \mathbf{h}(\theta_0))] = K_{\mathbf{h}(\theta_0)}(\phi(\theta))$ (with $K_{\mathbf{h}(\theta_0)}(\cdot) := M[\Gamma(\cdot, \mathbf{h}(\theta_0))]$), hence the following result.

**Lemma 2.16.** *If $\theta_0$ satisfies the intrinsic recoverability property on an open set $U \ni \theta_0$ with respect to $\phi$, then $\theta_0$ satisfies the intrinsic metric property on $U$ with respect to $\phi$.*

The intrinsic recoverability property is equivalent to equation 8 (see Section D for a proof):

**Theorem 2.17.** *Given $\phi \in \mathcal{C}^2(\mathbb{R}^D, \mathbb{R}^d)$ and $\theta_0 \in \mathbb{R}^D$, the following are equivalent: (i) there are conservation laws $\mathbf{h} \in \mathcal{C}^1(\Omega, \mathbb{R}^N)$ for $\phi$ on a neighborhood $\Omega$ of $\theta_0$ such that equation 8 holds for each $\theta \in \mathcal{M}_{\theta_0} \cap \Omega$; (ii) there is an open set $U \subseteq \Omega$ on which $\theta_0$ satisfies the intrinsic recoverability property Definition 2.15 (and thus the intrinsic metric property Definition 2.10) with respect to $\phi$.*

### 2.4.2 THE FROBENIUS PROPERTY IS ALMOST EQUIVALENT TO INTRINSIC RECOVERABILITY

We are interested in equation 8, as it implies the *intrinsic recoverability property*, and thus an intrinsic dynamic. It may not seem obvious a priori how to verify whether such a condition can hold, nor how to construct suitable conservation laws $\mathbf{h}$ in practice. Intuitively, one should select as many conservation laws as possible while ensuring they remain independent, in a specific sense (Marcotte et al., 2023, Definition 2.18). As shown by Marcotte et al. (2023), knowing the maximal number of such conservation laws can be checked using Lie brackets of the associated vector fields. We recall the relevant definitions and explain how this criterion applies in our setting.

**Definition 2.18** (Lie brackets). Given two vector fields $\chi_1, \chi_2 \in \mathcal{C}^\infty(\Theta, \mathbb{R}^d)$, the *Lie bracket* $[\chi_1, \chi_2]$ is the vector field defined by $[\chi_1, \chi_2](\theta) := \partial \chi_2(\theta) \chi_1(\theta) - \partial \chi_1(\theta) \chi_2(\theta)$.

**Definition 2.19** (Generated Lie algebra). Given some function space $\mathbb{W} \subseteq \mathcal{C}^\infty(\Theta, \mathbb{R}^d)$, the *generated Lie algebra* of $\mathbb{W}$ is the smallest subspace of $\mathcal{C}^\infty(\Theta, \mathbb{R}^d)$ that contains $\mathbb{W}$ and that is stable under Lie brackets, and is denoted $\mathrm{Lie}(\mathbb{W})$.

The *trace* at $\theta \in \Theta$ of any set $\mathbb{W} \subset \mathcal{C}^\infty(\Theta, \mathbb{R}^D)$ of vector fields is defined as the linear space

$$\mathbb{W}(\theta) := \mathrm{span}\{\chi(\theta) : \chi \in \mathbb{W}\} \subseteq \mathbb{R}^D, \tag{9}$$

and for any infinitely smooth $\phi$ we denote $\mathbb{W}_\phi := \mathrm{span}\{\nabla\phi_i(\cdot), \ 1 \le i \le d\} \subseteq \mathcal{C}^\infty(\Theta, \mathbb{R}^D)$.

**Definition 2.20** (Frobenius property). A $\mathcal{C}^\infty$ function $\phi$ satisfies the *Frobenius property* on $\Omega$ if for all $\theta \in \Omega$, $\mathrm{Lie}(\mathbb{W}_\phi)(\theta) = \mathbb{W}_\phi(\theta)$. This property is slightly weaker than involutivity (Isidori, 1995).

The following proposition (proved in Section E) relates this property to the intrinsic dynamic property of $\theta_0$. In particular, as the Frobenius property does not hold for $\phi_{\mathtt{Lin}}$ (Marcotte et al., 2023, Proposition I.1), it is not possible to have the intrinsic recoverability for linear networks with classical $\phi_{\mathtt{Lin}}$.

**Proposition 2.21.** *We have the following implications* $(i) \implies (ii) \implies (iii)$*:* $(i)$ *$\phi$ satisfies the Frobenius property on $\Omega$ and the trace of $\mathbb{W}_\phi$ has its dimension that is constant on $\Omega$;* $(ii)$ *For any $\theta_0 \in \Omega$, there exists conservation laws $\mathbf{h}$ for $\phi$ on a neighborhood $U \subset \Omega$ of $\theta_0$ such that for each $\theta \in \mathcal{M}_{\theta_0}$ equation 8 holds;* $(iii)$ *$\phi$ satisfies the Frobenius property on $\Omega$.*

**Link with the commuting property.** The result presented in Li et al. (2022) can be recovered as a special case of Proposition 2.21: the authors require the reparametrization $\phi$ to be *commuting*, meaning that for all pairs $\phi_i, \phi_j$, the Lie bracket $[\nabla\phi_i, \nabla\phi_j]$ is zero. In this setting, $\phi$ naturally satisfies the Frobenius property, and the result of Li et al. (2022) establishes an even stronger property: the dynamics on $\phi(\theta)$ form a mirror flow. In particular, it is worth noting that diagonal networks satisfy that their parametrization (product of the diagonals) is commuting (as all coordinates functions are separable), which thus (Li et al., 2022) implies a mirror flow dynamic, and thus an implicit bias (see e.g. Azulay et al. (2021)). In contrast, we consider a weaker condition; we seek only to determine whether the dynamics on $z = \phi(\theta)$ can be expressed intrinsically as a Riemannian gradient flow.

## 3 APPLICATION FOR GENERAL RELU NETWORKS AND LINEAR NETWORKS

We now show that the intrinsic recoverability property is satisfied *for any initialization* for the parametrization $\phi$ associated to a large class of (deep) ReLU neural networks. While this result is already known in the two-layer case (Marcotte et al., 2023, Examples 3.5 and 3.8), *here we establish it for the general model of ReLU networks of Gonon et al. (2024), associated to a directed acyclic graph (DAG) of any depth*, including skip connexions and arbitrary mixes of ReLU/linear/max-pooling activations. We first establish that $\phi_{\mathtt{ReLU}}$ satisfies the Frobenius property (see Section F.1 for a proof):

**Theorem 3.1.** *The parameterization $\phi_{\mathtt{ReLU}}$ used for ReLU neural networks with any DAG architecture (see Gonon et al. (2024) and our Example 2.3)) satisfies the Frobenius property on $(\mathbb{R}\backslash\{0\})^D$.*

The following theorem (proved in Section F.2 for general DAG ReLU networks) refines this result by allowing some zero entries in the parameters.

**Theorem 3.2.** *Consider an MLP with at least two neurons in each layer (including input and output layers), and denote $\Theta \subseteq \mathbb{R}^D$ the set of all parameters with at most one zero component. On this set, which is open, dense, and with a single connected component, the parameterization $\phi_{\mathtt{ReLU}}$ of Gonon et al. (2024) satisfies the Frobenius property.*

This leads to the following corollary (proved under weaker assumptions in Section F.3) which guarantees the existence of a maximal set of conservation laws big enough to ensure the intrinsic recoverability property.

**Corollary 3.3.** *For a general DAG ReLU network (resp. for an MLP with at least two neurons per layer), every element $\theta$ of the set $(\mathbb{R} \setminus \{0\})^D$ (resp. of the set $\Theta$ of Theorem 3.2) admits an open neighborhood $U$ in this set on which the intrinsic recoverability property (and thus the intrinsic dynamic property with respect to $\phi_{\mathtt{ReLU}}$) holds.*

Moreover, the known conservation laws given in Proposition 2.9 are independent on $\Theta$ from Theorem 3.2 (actually on an even larger set containing $\Theta$). Thus, they yield $m$ independent conservation laws on $\Theta$, where $m$ corresponds to the number of hidden neurons. To verify that these are indeed

the only ones, one must check that the trace of $\mathbb{W}_{\phi_{\text{ReLU}}}$ has dimension $D - m$; while it is empirically supported by Marcotte et al. (2023), which confirms that $\text{Lie}\big(\mathbb{W}_{\phi_{\text{ReLU}}}\big)(\theta) = \mathbb{W}_{\phi_{\text{ReLU}}}(\theta)$ has dimension $D - m$ when sampling random values of $\theta$, as well as random dimensions and depths, the following corollary establishes it theoretically (see Section F.4 for a proof).

**Corollary 3.4.** *Let us consider a MLP architecture with at least two neurons per layer, and with $m$ hidden neurons. Then the $m$ independent conservation laws for $\phi_{ReLU}$ given in Proposition 2.9 are exhaustive on the set $\Theta$ from Theorem 3.2: there is no more conservation laws than these ones.*

As a concrete example the following proposition (proved in Section G) provides the first closed-form formula for the intrinsic dynamic for a three-layer ReLU network with scalar input and output.

**Proposition 3.5.** *For a 3-layer ReLU MLP with scalar input/output, the factorization $\phi_{ReLU}$ reads[1]*

$$Z = \phi_{\text{ReLU}}(u, V, w) := \text{diag}(u)\, V\, \text{diag}(w) \in \mathbb{R}^{n \times m},$$

*with $u \in \mathbb{R}^n$, $V \in \mathbb{R}^{n \times m}$, and $w \in \mathbb{R}^m$. Define $\Theta := \{(u, V, w) : u_i, V_{ij}, w_j \neq 0\ \forall i, j\}$, and consider the $n + m$ conservation laws $\mathbf{h}(\theta) := \big((u_i^2 - \sum_j V_{ij}^2)_{i=1}^n, (w_j^2 - \sum_i V_{ij}^2)_{j=1}^m\big)$ for $\phi_{ReLU}$. Every $\theta_0 \in \Theta$ satisfies the intrinsic dynamics with respect to $\phi_{ReLU}$, which reads $\dot{z} = -K_{\theta_0}(z)\nabla f(z)$ with $z = \text{vec}(Z)$ corresponding to*

$$\dot{Z} = -\text{ddiag}(\nabla f(Z)Z^\top)\,\text{diag}(\boldsymbol{\alpha})^{-1}\, Z - \text{diag}(\boldsymbol{\alpha})\,\nabla f(Z)\,\text{diag}(\boldsymbol{\beta}) - Z\,\text{diag}(\boldsymbol{\beta})^{-1}\,\text{ddiag}(Z^\top \nabla f(Z)),$$

*where: a) for any matrix $M$, $\text{ddiag}(M) := \text{diag}\big(\text{Diag}(M)\big)$, where $\text{Diag}(M)$ extracts its diagonal as a vector and $\text{diag}(v)$ is the diagonal matrix with entries of $v$; and b) the vectors $\boldsymbol{\alpha} = \boldsymbol{\alpha}(Z, \mathbf{h}(\theta_0)) \in \mathbb{R}_{>0}^n$ and $\boldsymbol{\beta} := \boldsymbol{\beta}(Z, \mathbf{h}(\theta_0)) \in \mathbb{R}_{>0}^m$ (uniquely determined by $Z$ and $\mathbf{h}(\theta_0)$) satisfy*

$$\boldsymbol{\alpha}^2 - |Z|^2\,\text{diag}(\boldsymbol{\beta})^{-1}\mathbf{1}_n - \boldsymbol{\lambda} \odot \boldsymbol{\alpha} = 0, \quad \boldsymbol{\beta}^2 - (|Z|^2)^\top\,\text{diag}(\boldsymbol{\alpha})^{-1}\mathbf{1}_m - \boldsymbol{\mu} \odot \boldsymbol{\beta} = 0, \quad (10)$$

*with $|Z|^2 \in \mathbb{R}^{n \times m}$ the element-wise square on the matrix $Z \in \mathbb{R}^{n \times m}$ and with $\boldsymbol{\lambda} \in \mathbb{R}^n$, $\boldsymbol{\mu} \in \mathbb{R}^m$ such that $\mathbf{h}(\theta_0) = (\boldsymbol{\lambda}, \boldsymbol{\mu})$. When $\boldsymbol{\lambda}, \boldsymbol{\mu} = 0$, equation 10 entirely characterizes $(\boldsymbol{\alpha}, \boldsymbol{\beta})$.*

## 3.1 DEEP LINEAR NEURAL NETWORKS AND LINEAR NEURAL ODES

For $L$-layer linear networks, $\theta = (U_1, \ldots, U_L)$ and the path-lifting formalism (Gonon et al., 2024) yields a factorization via $\phi_{\text{ReLU}}$, leading to an intrinsic dynamic by the results of the previous section. It is more common, however, to consider the dynamics of $Z_L := \phi_{\text{Lin}}(\theta_L) = U_L \cdots U_1$, since $\phi_{\text{Lin}}$ is more efficient than $\phi_{\text{ReLU}}$ in terms of dimension reduction. We now analyze the dynamics of $Z_L(t)$. The gradient flow $\dot{\theta}_L = -\nabla\ell(\theta_L)$ gives the evolution of $Z_L$ (see e.g. (Bah et al., 2022, Lemma 2)):

$$\dot{Z}_L = -\sum_{j=1}^L S_j\,\nabla f(Z_L)\,T_j, \quad \text{with} \quad \begin{cases} S_j := U_L \cdots U_{j+1}\,U_{j+1}^\top \cdots U_L^\top, & S_L = \text{Id}, \\ T_j := U_1^\top \cdots U_{j-1}^\top\,U_{j-1} \cdots U_1, & T_1 = \text{Id}. \end{cases} \quad (11)$$

The metric $M(\theta_L)$ on $z_L = \text{vec}(Z_L)$ is thus entirely characterized by $(S_j(\theta_L), T_{j+1}(\theta_L))_{j=1}^{L-1}$.

**Definition 3.6** (Relaxed balanced condition). We say that $\theta_L := (U_L, \cdots, U_1)$ satisfies the relaxed balanced condition if there exists $\boldsymbol{\lambda} := (\lambda_i)_i \in \mathbb{R}^{L-1}$ such that

$$U_{i+1}^\top U_{i+1} - U_i U_i^\top = \lambda_i \text{Id}, \quad \forall 1 \leq i \leq L-1. \quad (12)$$

(0-)balanced conditions (Bah et al., 2022, Def 1) (Arora et al., 2019, Def 1) correspond to $\boldsymbol{\lambda} = 0$.

*Remark* 3.7. It is worth noting that Dominé et al. (2025) used this exact same condition and called it the $\lambda$-balanced condition. However, the definition of $\lambda$-balanced condition is already used (see (Arora et al., 2019, Def 1)) by the literature to refer to the weaker condition $\|U_{i+1}^\top U_{i+1} - U_i U_i^\top\| \leq \lambda_i$. Other works (see e.g. Tarmoun et al. (2021); Braun et al. (2022); Varre et al. (2023)) use stronger conditions on the initializations, that in particular satisfy the relaxed balanced conditions of Definition 3.6.

### 3.1.1 DEEP LINEAR NEURAL NETWORKS

We first detail the study of the two-layer case, and then generalize it to the deep case.

**Matrix factorization.** We consider the two-layer case where $\theta := (U, V) \in \mathbb{R}^{n \times r} \times \mathbb{R}^{m \times r}$ and with $Z = \phi_{\text{Lin}}(\theta) := UV^\top \in \mathbb{R}^{n \times m}$. We assume $\theta(t)$ satisfies the gradient flow equation 1 with $\theta(0) = (U_{t=0}, V_{t=0})$. We denote $S := U_{t=0}^\top U_{t=0} - V_{t=0}^\top V_{t=0} \in \mathbb{R}^{r \times r}$.

---

[1]When written as a $n \times m$ matrix, we denote $Z$ instead of $z$ and also view $\nabla f(Z)$ as an $n \times m$ matrix.

If $\theta_0$ satisfies the balanced condition equation 12 $S = 0$, then (Arora et al., 2018, Theorem 1) (Bah et al., 2022, Lemma 2) $\theta_0$ satisfies the intrinsic metric property with respect to $\phi_{\texttt{Lin}}$ and

$$\dot{Z} = -\sqrt{ZZ^\top}\nabla f(Z) - \nabla f(Z)\sqrt{Z^\top Z}. \tag{13}$$

We generalize this result (see Section H for a proof) to a broader class of initializations: all initializations satisfying the relaxed balanced condition equation 12 possess the intrinsic metric property.

**Theorem 3.8.** *Consider $\theta_0 := (U_{t=0}, V_{t=0})$ where both $U_{t=0} \in \mathbb{R}^{n \times r}$ and $V_{t=0} \in \mathbb{R}^{m \times r}$ have full rank $r \leq \min(n, m)$, and assume $S = \lambda\mathrm{Id}_r$ for some $\lambda \in \mathbb{R}$. Then, on a neighborhood $\Omega$ of $\theta_{t=0}$:*

$$\dot{Z} = -\Pi_Z\left[\frac{\lambda}{2}\mathrm{Id}_n + \frac{1}{2}\sqrt{\lambda^2\mathrm{Id}_n + 4\,ZZ^\top}\right]\nabla f(Z) - \nabla f(Z)\Pi_{Z^\top Z}\left[-\frac{\lambda}{2}\mathrm{Id}_m + \frac{1}{2}\sqrt{\lambda^2\mathrm{Id}_m + 4\,Z^\top Z}\right],$$
$$\tag{14}$$

*where $\Pi_A$ is the orthogonal projector on $\mathrm{range}\,A$.*

Note that equation 13 corresponds indeed to equation 14 with $\lambda = 0$ and that $r \leq \min(n, m)$ is necessary to have $S = \lambda\mathrm{Id}_r$ for some $\lambda \neq 0$. Note also that Theorem 3.8 generalizes to the case $r \leq \min(n, m)$ the expression obtained in (Dominé et al., 2025, Theorem 5.2) for the special case $r = \min(n, m)$ (if Dominé et al. (2025) focus in general on the squared loss, the proof of their Theorem 5.2 does not rely on the use of this specific loss: this result can be applied for any loss, as ours). The following theorem shows that the *relaxed balanced condition* is actually a necessary condition when $r \leq \max(n, m)$ to have the *strong* intrinsic dynamic property (see Definition 2.10 in Section I), a variant of Definition 2.6 where $f$ is *piecewise* $\mathcal{C}^2$. Its proof (see Section I) relies on showing the non-inclusion of the kernels of equation 7.

**Theorem 3.9.** *Let $\theta_0 := (U_{t=0}, V_{t=0})$. Assume that both $U_{t=0} \in \mathbb{R}^{n \times r}$ and $V_{t=0} \in \mathbb{R}^{m \times r}$ have full rank and that $r \leq \max(n, m)$. If $S := U_{t=0}^\top U_{t=0} - V_{t=0}^\top V_{t=0} \neq \lambda\mathrm{Id}_r$, then $\theta_0$ does not satisfy the strong intrinsic dynamic property with respect to $\phi_{\texttt{Lin}}$.*

The case $r > \max(n, m)$ is still open. For $n = m = 1$ and any $r$, the following proposition (proved in Section J) shows that all initializations *do* satisfy the intrinsic metric property with respect to $\phi_{\texttt{Lin}}$.

**Proposition 3.10.** *Let $\theta := (u, v)$ with $u \in \mathbb{R}^r$ and $v \in \mathbb{R}^r$. Then $z := \phi_{\texttt{Lin}}(\theta) = \langle u, v \rangle \in \mathbb{R}$. We denote $S := u_{t=0}u_{t=0}^\top - v_{t=0}v_{t=0}^\top \in \mathbb{R}^{r \times r}$. Then one has $\dot{z} = -\sqrt{2\mathrm{tr}(S^2) - \mathrm{tr}(S)^2 + 4z^2}\nabla f(z)$.*

In particular, it is important to note that the two-layer linear analysis allows these results to be applied directly to networks composed of attention layers (Example 2.4).

**Deep linear neural networks.** Consider linear neural networks of arbitrary depth, with square weight matrices $\theta_L := (U_L, \ldots, U_1)$, $U_i \in \mathbb{R}^{n \times n}$. Indeed, this is precisely the main case where a non-trivial relaxed balanced condition ($\lambda_i \neq 0$) can occur at every layer $i$, since $n_i = n$ and thus $n_i \leq \min(n_{i-1}, n_{i+1})$ for all $i$. The following theorem (proved in Section K) generalizes Theorem 3.8 to this setting. In the case of balanced conditions ($\boldsymbol{\lambda} = 0$), our theorem recovers the dynamics described in (Arora et al., 2018, Theorem 1), (Bah et al., 2022, Lemma 2).

**Theorem 3.11.** *If $\theta_L(0)$ satisfies the relaxed balanced condition (Definition 3.6) with $\boldsymbol{\lambda} = (\lambda_i)_i$ then during the trajectory $\theta_L(t)$ of equation 1, the matrices in equation 11 satisfy $S_j(\theta_L(t)) = Q_j(U_L(t)U_L(t)^\top)$ and $T_j(\theta_L(t)) = R_j(U_1(t)^\top U_1(t))$, where $Q_j(x) := \prod_{k=0}^{L-j-1}(x - a_k)$ with $a_0 := 0$ and $a_k := \sum_{i=1}^{k}\lambda_{L-i}$ for $k = 1, \cdots L-1$ and $R_j(x) := \prod_{k=0}^{j-2}(x - b_k)$ with $b_0 := 0$ and $b_k := -\sum_{i=1}^{k}\lambda_i$. Moreover $U_L U_L^\top$ (resp. $U_1^\top U_1$) is the unique root of $Z_L Z_L^\top = Q_0(U_L U_L^\top)$ (resp. of $Z_L^\top Z_L = R_{L-1}(U_1^\top U_1)$) with spectrum lower bounded by $\max_{0 \leq k \leq L-1} a_k$ (resp. by $\max_{0 \leq k \leq L-2} b_k$). This implies that all matrices in equation 11 are entirely characterized by $Z_L$ and the initialization, hence $\theta_L(0)$ satisfies the intrinsic dynamic property on $\mathbb{R}^D$ with respect to $\phi_{\texttt{Lin}}$.*

### 3.1.2 INFINITELY DEEP LINEAR NETWORKS

We next consider the limit when $L \to +\infty$ of deep linear residual networks with parameters $U_k = \mathrm{Id}_n + \mathcal{A}_{\frac{k}{L}}$, and thus focus on the analysis of the parameter $\theta = (\mathcal{A}_s)_{s \in [0,1]}$, where $\mathcal{A}_s \in \mathbb{R}^{n \times n}$, which corresponds to a linear neural ODEs model (introduced by Chen et al. (2018)). Remarkably, our theoretical approach still applies in this regime, and yields a closed-form formula for the metric. We thus study the dynamics of parameters $\theta(t) \in \mathcal{X}$ where $\mathcal{X}$ corresponds to the Banach

space $(\mathcal{C}^1([0,1],\mathbb{R}^{n\times n}), \|\cdot\|_{\mathcal{C}^1})$ where $\|f\|_{\mathcal{C}^1} := \max\{\|f\|_\infty, \|f'\|_\infty\}$, and such that the trajectory $t \mapsto \theta(t) = (\mathcal{A}_s(t))_{s\in[0,1]}$ is the solution of the gradient flow on $\ell(\theta)$, given by the (family of coupled) ODEs

$$\forall s \in [0,1], \quad \frac{\partial \mathcal{A}_s}{\partial t}(t) = -\mathfrak{g}_s(t), \quad \text{with} \quad \mathfrak{g}_s(t) := \frac{\partial \ell}{\partial \mathcal{A}_s}\left(\theta(t)\right) \in \mathbb{R}^{n\times n}, \tag{15}$$

where we assume that the loss function $\ell : \mathcal{X} \mapsto \mathbb{R}$ is such that $\theta \mapsto (\frac{\partial \ell}{\partial \mathcal{A}_s}(\theta))_{s\in[0,1]}$ is locally Lipschitz on $\mathcal{X}$ (to ensure by the Cauchy-Lipschitz theorem that indeed there exists a unique maximal solution $\theta(\cdot) \in \mathcal{C}^1([0,T), \mathcal{X})$ of equation 15 with a given $\theta(0)$).

As an infinite-depth analog of $Z_L = U_L \dots U_1$, given any $\theta \in \mathcal{X}$ we consider $s \in [0,1] \mapsto Z_s = Z_s[\theta] \in \mathbb{R}^{n\times n}$ the unique global solution (as $\theta = (\mathcal{A}_s)_{s\in[0,1]} \in \mathcal{X}$) of the *state equation*

$$\frac{\mathrm{d}}{\mathrm{d}s}Z_s = \mathcal{A}_s Z_s, \quad Z_0 = \mathrm{Id}_n. \tag{16}$$

The analog to Assumption 2.1 is to assume that $\ell(\theta) = f(Z_{s=1})$ with $f \in \mathcal{C}^1$, and we now want to know if it is possible to rewrite the dynamics $\frac{\partial Z_{s=1}}{\partial t}(t)$ as an intrinsic dynamic that only depends on $Z_{s=1}(t)$ and the initialization $\theta(0)$. The following proposition (see Section L for a proof) gives a set of conserved functions during all trajectories of equation 15.

**Proposition 3.12.** *For any $s \in [0,1]$, consider $\mathbf{h}_s : \theta := (\mathcal{A}_s)_{s\in[0,1]} \in \mathcal{X} \mapsto \mathcal{A}'_s + \mathcal{A}'^\top_s + [\mathcal{A}^\top_s, \mathcal{A}_s] \in \mathbb{R}^{n\times n}$, where we denote $\mathcal{A}'_s := \frac{\mathrm{d}}{\mathrm{d}s}\mathcal{A}_s$. Then for any $s \in [0,1]$, one has for any $t$: $\mathbf{h}_s(\theta(t)) = \mathbf{h}_s(\theta(0))$, where $\theta(t)$ is the maximal solution of equation 15 with initialization $\theta(0)$.*

Moreover, the following theorem (see Section M for a proof) shows that for relaxed balanced initializations, the evolution of $Z_1(t) = Z_{s=1}(t)$ is intrinsic as it depends only on $Z_1$ and the initialization $\theta(0)$:

**Theorem 3.13.** *If the initialization $\theta(0)$ satisfies that for each $s \in [0,1]$ $\mathbf{h}_s(\theta(0)) = \lambda(s)\mathrm{Id}_n$ for some $\lambda(\cdot) \in \mathcal{C}^0([0,1],\mathbb{R})$, then one has*

$$\dot{Z}_1 = -\int_0^1 (Z_1 Z_1^\top)^{1-s} \exp(\gamma(s))\nabla f(Z_1)(Z_1^\top Z_1)^s \mathrm{d}s,$$

*with $\gamma(s) := (1-s)\psi_1(1) - \psi_1(1-s) - s\psi_2(1) + \psi_2(s)$, where $\psi_1 : s \in [0,1] \mapsto \int_0^s \int_0^u \lambda(1-v)\mathrm{d}v\mathrm{d}u$ and $\psi_2 : s \in [0,1] \mapsto \int_0^s \int_0^u \lambda(v)\mathrm{d}v\mathrm{d}u$. If $\lambda(\cdot) \equiv 0$ (balanced-condition), then $\gamma(\cdot) \equiv 0$.*

In a sense, this theorem captures the infinite-depth limit ($L \to +\infty$) of Theorem 3.11, while offering the key advantage of an explicit closed-form expression for the associated metric.

## CONCLUSION

In this paper, we investigated when high-dimensional gradient flows can be recast as intrinsic Riemannian flows in lower-dimensional spaces. Our results show that such reductions are always possible for ReLU networks under path-lifting parametrization, and for linear networks under relaxed balanced initializations. A central contribution is our analysis of the "path-lifting metric", a recently introduced and still largely unexplored object, for which we provide an intrinsic characterization in the 3-layer case. Extending this analysis to deeper or more general architectures could shed new light on the geometry of gradient dynamics for general ReLU networks.

## ACKNOWLEDGMENTS

The work of G. Peyré was supported by the French government under the management of Agence Nationale de la Recherche as part of the "Investissements d'avenir" program, reference ANR-19-P3IA-0001 (PRAIRIE 3IA Institute) and by the European Research Council (ERC project WOLF). The work of R. Gribonval was supported by the AllegroAssai ANR-19-CHIA-0009 project of the French Agence Nationale de la Recherche and by the SHARP ANR project ANR-23-PEIA-0008 in the context of the France 2030 program. The authors thank Stephan Thomassé, Gilles Blanchard and Laurent Jacques for discussions on rank properties of DAGs that helped simplify the proof of Proposition F.7 in Section F.2.

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

# A   A TABLE THAT SUMMARIZES WHICH PARAMETRIZATIONS CAN BE USED TO ANALYZE WHICH TYPE OF NEURAL NETWORK, ALONG WITH THE CORRESPONDING RESULTS.

| Parametrization / Network type | $\phi_{\text{Lin}}$ | $\phi_{\text{ReLU}}$ |
|---|---|---|
| Linear network | IMP: **only for relaxed balanced** $\theta_0$ 
 Dimension: $d' \ll D$ | IMP: **for all** $\theta_0$ 
 $d' = D - k \approx D$ |
| DAG ReLU | **N/A** | IMP: **for all** $\theta_0$ 
 $d' = D - k \approx D$ |

IMP = intrinsic metric property ; $d'$ = dimension of the manifold of $\phi(\theta)$; $k$= # hidden neurons

Table 1: *The table summarizes which parametrizations can be used to analyze which type of neural network, along with the corresponding results.*

# B   MORE COMMENTS ON EXAMPLE 2.5

In example equation 2.5, when $\lambda \to +\infty$ $K_{\theta_0}/\lambda \to \text{Id}_m$ (the Euclidean metric). When $\lambda = 0$, one has $K_{\theta_0}(z) = \|z\|\text{Id}_m + \frac{zz^\top}{\|z\|}$ for every $z \neq 0$, and in particular, by the uniqueness result in the Cauchy-Lipschitz theorem, 0 is reachable only if $z_0 = 0$. When $\lambda \to -\infty$, $K_{\theta_0}/|\lambda| \to \frac{zz^\top}{\|z\|^2}$. See `https://github.com/sibyllema/Intrinsic_training_dynamic` for numerical illustrations of these different behaviors.

# C   PROOF OF THEOREM 2.14

**Theorem 2.14.** *Consider* $\mathbf{h} \in \mathcal{C}^1(\mathbb{R}^D, \mathbb{R}^N)$, $\phi \in \mathcal{C}^2(\mathbb{R}^D, \mathbb{R}^d)$, *and* $\theta_0 \in \mathbb{R}^D$ *such that the matrix* $\partial\mathbf{h}(\theta) \in \mathbb{R}^{N \times D}$ *has constant rank on* $\mathcal{M}_{\theta_0} \cap U$ *with* $U \ni \theta_0$ *an open subset of* $\mathbb{R}^D$ *and* $\mathcal{M}_{\theta_0} := \mathbf{h}^{-1}(\{\mathbf{h}(\theta_0)\})$. *Then (i)* $\implies$ *(ii), where*

*(i) There exists an open set* $O \supset \phi(\mathcal{M}_{\theta_0} \cap U)$ *and a map* $K_{\theta_0} \in \mathcal{C}^1(O, \mathbb{R}^{d \times d})$ *such that:*

$$M(\theta) = K_{\theta_0}(\phi(\theta)), \qquad \forall \theta \in \mathcal{M}_{\theta_0} \cap U;$$

*(ii)*
$$\ker\partial\phi(\theta) \cap \ker\partial\mathbf{h}(\theta) \subseteq \ker\partial M(\theta), \quad \forall \theta \in \mathcal{M}_{\theta_0} \cap U. \tag{7}$$

*Proof.* **(i)** $\Rightarrow$ **(ii).**

Assume (i) and fix $\theta \in \mathcal{M}_{\theta_0} \cap U$ and a vector $v \in \ker \partial\phi(\theta) \cap \ker \partial\mathbf{h}(\theta)$. Applying the chain rule in the ambient space $\mathbb{R}^D$ (possible as $v \in \ker \partial\mathbf{h}(\theta) = T_\theta\mathcal{M}_{\theta_0}$ because $\partial\mathbf{h}(\theta)$ has its rank that remains constant on $\mathcal{M}_{\theta_0} \cap U$ by hypothesis) gives

$$\partial M(\theta) \cdot v = \partial K_{\theta_0}(\phi(\theta)) \cdot \left(\partial\phi(\theta) \cdot v\right) = \partial K_{\theta_0}(\theta) \cdot 0 = 0,$$

hence $v \in \ker \partial M(\theta)$ and (ii) holds.

$\square$

# D   PROOF OF THEOREM 2.17

**Theorem 2.17.** *Given* $\phi \in \mathcal{C}^2(\mathbb{R}^D, \mathbb{R}^d)$ *and* $\theta_0 \in \mathbb{R}^D$, *the following are equivalent: (i) there are conservation laws* $\mathbf{h} \in \mathcal{C}^1(\Omega, \mathbb{R}^N)$ *for* $\phi$ *on a neighborhood* $\Omega$ *of* $\theta_0$ *such that equation 8 holds for each* $\theta \in \mathcal{M}_{\theta_0} \cap \Omega$; *(ii) there is an open set* $U \subseteq \Omega$ *on which* $\theta_0$ *satisfies the intrinsic recoverability property Definition 2.15 (and thus the intrinsic metric property Definition 2.10) with respect to* $\phi$.

*Proof.* We first show $(i) \implies (ii)$. We assume $(i)$. Observe that given any $\theta \in \mathbb{R}^D$ equation 8 is equivalent (by rank theorem) to

$$\text{rank}\begin{pmatrix} \partial\phi(\theta) \\ \partial\mathbf{h}(\theta) \end{pmatrix} = D. \tag{17}$$

By smoothness of $\phi$ and $\mathbf{h}$, if equation 17 holds at $\theta_0$ it also holds in a whole neighborhood $U$ of $\theta_0$. By the implicit function theorem, denoting $F(\theta) := (\phi(\theta), \mathbf{h}(\theta))$, it implies that $\theta$ can be expressed on $U$ as $\theta = F^{-1}(\phi(\theta), \mathbf{h}(\theta)) = \Gamma(\phi(\theta), \mathbf{h}(\theta))$.

We now show $(ii) \implies (i)$. We assume $(ii)$. Then on $U$ one has $\theta = \Gamma(\phi(\theta), \mathbf{h}(\theta))$. Thus on $\mathcal{M}_{\theta_0} \cap U$, one has $\theta = \Gamma(\phi(\theta), \mathbf{h}(\theta_0))$. We now fix some $\theta \in \mathcal{M}_{\theta_0} \cap U$ and we consider a vector $v \in \ker \partial\phi(\theta) \cap \ker \partial\mathbf{h}(\theta)$.

Applying the chain rule in the ambient space $\mathbb{R}^D$ on $\Gamma$ gives

$$v = \text{Id}_D v = \partial_{\phi(\theta)}\Gamma(\phi(\theta), \mathbf{h}(\theta_0)) \cdot (\partial\phi(\theta) \cdot v) = \partial\Gamma(\theta) \cdot 0 = 0,$$

and thus $v = 0$. $\qquad\square$

# E    PROOF OF PROPOSITION 2.21

**Proposition 2.21.** *We have the following implications $(i) \implies (ii) \implies (iii)$: $(i)$ $\phi$ satisfies the Frobenius property on $\Omega$ and the trace of $\mathbb{W}_\phi$ has its dimension that is constant on $\Omega$; $(ii)$ For any $\theta_0 \in \Omega$, there exists conservation laws $\mathbf{h}$ for $\phi$ on a neighborhood $U \subset \Omega$ of $\theta_0$ such that for each $\theta \in \mathcal{M}_{\theta_0}$ equation 8 holds; $(iii)$ $\phi$ satisfies the Frobenius property on $\Omega$.*

*Proof.* $(i) \implies (ii)$ is a direct consequence of the proof of (Marcotte et al., 2023, Proposition 3.7).

We now show $(ii) \implies (iii)$. Let us assume $(ii)$. We fix $\theta_0$. Then by assumption on $U \ni \theta_0$, $\theta = \Gamma(\mathbf{h}(\theta), \phi(\theta))$, and by using Proposition 2.21 one has $\ker \partial\phi(\theta) \cap \ker \partial\mathbf{h}(\theta) = \{0\}$ on $\mathcal{M}_{\theta_0} \cap U$. Thus equation 17 holds on an open neighborhood $O$ of $\theta_0$. As $\mathbf{h}$ are conservation laws for $\phi$ on $U$, one has on $O \cap U$ that $\dim(\mathbb{W}_\phi(\theta)) = D - \text{rank}\partial\mathbf{h}(\theta)$.

But as for any conservation law $h$ for $\phi$, one has $\langle \nabla h(\theta), \nabla\phi_i(\theta) \rangle = \langle \nabla h(\theta), \nabla\phi_j(\theta) \rangle = 0 \implies \langle \nabla h(\theta), [\nabla\phi_i, \nabla\phi_j](\theta) \rangle = 0$, one has necessarily $\dim \text{Lie}\mathbb{W}_\phi(\theta_0) \leq D - \text{rank}\partial\mathbf{h}(\theta_0)$ and as one also has $D - \text{rank}\partial\mathbf{h}(\theta_0) = \dim\mathbb{W}_\phi(\theta_0) \leq \dim \text{Lie}\mathbb{W}_\phi(\theta_0)$ then one has $\mathbb{W}_\phi(\theta_0) = \text{Lie}\mathbb{W}_\phi(\theta_0)$. This holds for any $\theta_0$, which concludes the proof. $\qquad\square$

# F    PROOF OF THEOREM 3.1, THEOREM 3.2, COROLLARY 3.3 AND COROLLARY 3.4

## F.1    PROOF OF THEOREM 3.1

**Theorem 3.1.** *The parameterization $\phi_{\texttt{ReLU}}$ used for ReLU neural networks with any DAG architecture (see Gonon et al. (2024) and our Example 2.3)) satisfies the Frobenius property on $(\mathbb{R}\backslash\{0\})^D$.*

*Proof.* We consider a parametrization $\phi : \theta \mapsto (\phi_i(\theta))_{i=1}^d$, where all $\phi_i$ are monomials in $\theta = (\theta_1, \cdots, \theta_D) \in (\mathbb{R}\backslash\{0\})^D$, i.e. $\phi_i(\theta) = \prod_{\ell=1}^D \theta_\ell^{\alpha_\ell^{(i)}}$. Moreover, if a variable $\theta_\ell$ appears in some coordinate with exponent $\alpha_\ell^{(i)} > 0$, then every other coordinate that contains $\theta_\ell$ uses the *same* exponent $\alpha_\ell^{(k)} = \alpha_\ell^{(i)}$. These assumptions are indeed satisfied for the path-lifting parametrization $\phi_{\texttt{ReLU}}$ associated to general ReLU networks (Gonon et al., 2024; Stock & Gribonval, 2022), associated to a directed acyclic graph (DAG) of any depth, including skip connexions and arbitrary mixes of ReLU/linear/max-pooling activations (and even slight generalizations of max-pooling).

Now let us consider two indices $i, j \in \{1, \ldots, d\}$. Denote $I$ (resp. $J$) the subset of all indices $\ell$ such that $\alpha_\ell^{(i)} \neq 0$ (resp. $\alpha_\ell^{(j)} \neq 0$). By abuse of notation we write $i \cap j$ (resp. $i\backslash j$ etc.) the set $I \cap J$ (resp. $I\backslash J$) and denote $\theta_{i\cap j}$ etc. the restriction of $\theta$ to the corresponding entries. In particular, one can decompose

$$\theta = \left(\theta_{i\cap j}, \; \theta_{i\backslash j}, \; \theta_{j\backslash i}, \; \theta_{(i\cap j)^c}\right).$$

We write
$$\phi_i(\theta) = \phi_{i\cap j}\big(\theta_{i\cap j}\big)\,\phi_{i\setminus j}\big(\theta_{i\setminus j}\big), \quad \phi_j(\theta) = \phi_{i\cap j}\big(\theta_{i\cap j}\big)\,\phi_{j\setminus i}\big(\theta_{j\setminus i}\big),$$
where $\phi_{i\cap j}(\cdot)$ is the maximal monomial factoring both $\phi_i(\cdot)$ and $\phi_j(\cdot)$, and $\phi_{i\setminus j}$ (resp. $\phi_{j\setminus i}$) is the unique monomial such that
$$\phi_i(\cdot) = \phi_{i\cap j}(\cdot)\,\phi_{i\setminus j}(\cdot), \qquad \phi_j(\cdot) = \phi_{i\cap j}(\cdot)\,\phi_{j\setminus i}(\cdot).$$

Then one has:
$$\nabla\phi_i(\theta) = \begin{pmatrix} \nabla\phi_{i\cap j}\phi_{i\setminus j} \\ \phi_{i\cap j}\nabla\phi_{i\setminus j} \\ 0 \\ 0 \end{pmatrix} \quad \text{and} \quad \nabla\phi_j(\theta) = \begin{pmatrix} \nabla\phi_{i\cap j}\phi_{j\setminus i} \\ 0 \\ \phi_{i\cap j}\nabla\phi_{j\setminus i} \\ 0 \end{pmatrix}$$

and
$$\partial^2\phi_i(\theta) = \begin{pmatrix} \partial^2\phi_{i\cap j}\phi_{i\setminus j} & \nabla\phi_{i\cap j}\nabla\phi_{i\setminus j}^\top & 0 & 0 \\ \nabla\phi_{i\setminus j}\nabla\phi_{i\cap j}^\top & \partial^2\phi_{i\setminus j}\phi_{i\cap j} & 0 & 0 \\ 0 & 0 & 0 & 0 \\ 0 & 0 & 0 & 0 \end{pmatrix}$$

Thus
$$\partial^2\phi_i(\theta)\nabla\phi_j(\theta) = \begin{pmatrix} \partial^2\phi_{i\cap j}\phi_{i\setminus j} & \nabla\phi_{i\cap j}\nabla\phi_{i\setminus j}^\top & 0 & 0 \\ \nabla\phi_{i\setminus j}\nabla\phi_{i\cap j}^\top & \partial^2\phi_{i\setminus j}\phi_{i\cap j} & 0 & 0 \\ 0 & 0 & 0 & 0 \\ 0 & 0 & 0 & 0 \end{pmatrix} \begin{pmatrix} \nabla\phi_{i\cap j}\phi_{j\setminus i} \\ 0 \\ \phi_{i\cap j}\nabla\phi_{j\setminus i} \\ 0 \end{pmatrix}$$
$$= \begin{pmatrix} \partial^2\phi_{i\cap j}\nabla\phi_{i\cap j}\phi_{j\setminus i}\phi_{i\setminus j} \\ \nabla\phi_{i\setminus j}\|\nabla\phi_{i\cap j}\|^2\phi_{j\setminus i} \\ 0 \\ 0 \end{pmatrix}$$
$$= \phi_{j\setminus i} \begin{pmatrix} \partial^2\phi_{i\cap j}\nabla\phi_{i\cap j}\phi_{i\setminus j} \\ \nabla\phi_{i\setminus j}\|\nabla\phi_{i\cap j}\|^2 \\ 0 \\ 0 \end{pmatrix},$$

and similarly one has:
$$\partial^2\phi_j(\theta)\nabla\phi_i(\theta) = \phi_{i\setminus j} \begin{pmatrix} \partial^2\phi_{i\cap j}\nabla\phi_{i\cap j}\phi_{j\setminus i} \\ 0 \\ \nabla\phi_{j\setminus i}\|\nabla\phi_{i\cap j}\|^2 \\ 0 \end{pmatrix}$$

Finally one has:
$$[\nabla\phi_i, \nabla\phi_j](\theta) = \begin{pmatrix} 0 \\ -\nabla\phi_{i\setminus j}\|\nabla\phi_{i\cap j}\|^2\phi_{j\setminus i} \\ \nabla\phi_{j\setminus i}\|\nabla\phi_{i\cap j}\|^2\phi_{i\setminus j} \\ 0 \end{pmatrix}$$
$$= \|\nabla\phi_{i\cap j}\|^2 \begin{pmatrix} 0 \\ -\nabla\phi_{i\setminus j}\phi_{j\setminus i} \\ \nabla\phi_{j\setminus i}\phi_{i\setminus j} \\ 0 \end{pmatrix}$$

But as:
$$\phi_{j\setminus i}\nabla\phi_i - \phi_{i\setminus j}\nabla\phi_j = \phi_{i\cap j} \begin{pmatrix} 0 \\ \phi_{j\setminus i}\nabla\phi_{i\cap j} \\ -\phi_{i\setminus j}\nabla\phi_{j\cap i} \\ 0 \end{pmatrix}$$

As $\phi_{i\cap j} \neq 0$ (indeed $\theta \in (\mathbb{R}\setminus\{0\})^D$, one then has $\begin{pmatrix} 0 \\ \phi_{j\setminus i}\nabla\phi_{i\cap j} \\ -\phi_{i\setminus j}\nabla\phi_{j\cap i} \\ 0 \end{pmatrix} \in \mathbb{W}_\phi(\theta)$ and thus
$$[\nabla\phi_i, \nabla\phi_j](\theta) \in \mathbb{W}_\phi(\theta).$$

$\square$

### F.2 PROOF OF THEOREM 3.2.

*To prove Theorem 3.2, we introduce the notion of prunable DAGs and characterize them, before showing (see Corollary F.11) that the rank of $\partial\phi(\theta)$ is almost everywhere equal to that of $\partial\phi(\mathbf{1}_D)$*

#### F.2.1 PRUNABLE DAGS

**Definition F.1.** Given a DAG $G = (V, E)$ with vertices $V$ and edges $E$ and, for each edge $i \in E$ we denote $G_i' = (V, E_i')$ with $E_i' := E \setminus \{i\}$ the smaller DAG obtained by removing edge $i$. The input neurons of $G$ (neurons without any antecedent) are always also input neurons of $G_i'$, and similarly for the output neurons of $G$ (neurons without any successor), and we say that $G$ is *prunable at edge $i$* if $G_i'$ has no additional input/output neurons. The graph $G$ is said to be *prunable everywhere (resp. somewhere/nowhere)* if it is prunable at every (resp. at some/at no) edge $i$.

Recalling that, by definition, a path in a graph is a sequence of consecutive edges starting from an input neuron and ending at an output neuron, we denote $\mathcal{P}_i := \{p : p \ni i\}$ the set of all paths of $G$ containing an edge $i \in E$, and $\mathcal{P}_i^c$ the other paths of $G$.

**Lemma F.2.** *The set $\mathcal{P}_i^c$ is always a subset of the paths of $G_i'$. Moreover, if $G$ is prunable at edge $i$, then $\mathcal{P}_i^c$ is exactly the set of all paths of $G_i'$.*

*Proof.* If $p \in \mathcal{P}_i^c$ then its edges are distinct from $i$, hence $p$ is a path of $G_i'$. Conversely, when $G$ is prunable at $i$, $G_i'$ has the same input and output neurons as $G$, hence if $p$ is a path of $G_i'$ it is a path of $G$ with edges in $E_i'$, i.e. with no edge equal to $i$, i.e. $p \in \mathcal{P}_i^c$. $\square$

The following straightforward characterizations will be useful.

**Lemma F.3.** *A graph $G$ is prunable at edge $i$ if, and only if, the output neuron of edge $i$ has at least another incoming edge, and the input neuron of edge $i$ has at least another outgoing edge.*

**Corollary F.4.** *A graph $G$ is prunable everywhere if, and only if, every non-output neuron has at least two successors, and every non-input neurons has at least two antecedents.*

**Corollary F.5.** *A graph $G$ associated to a multilayer perceptron (MLP) of depth $L$ with layer widths $N_\ell$, $0 \le \ell \le L$, is prunable everywhere if, and only if, $N_\ell \ge 2$ for every layer $\ell$ (including input and output layers).*

#### F.2.2 RANK PROPERTIES

**Lemma F.6.** *The parameterization $\phi = \phi_{\texttt{ReLU}}$ used for any general DAG ReLU neural network satisfies that for any $\theta \in (\mathbb{R} \setminus \{0\})^D$, one has $\operatorname{rank}(\partial\phi(\theta)) = \operatorname{rank}(\partial\phi(\mathbf{1}_D))$, where $\mathbf{1}_D$ is the vector with all its coordinates equal to $1$.*

*Proof.* By definition of the component $\phi_p$ of $\phi = (\phi_p)_p$ for each path $p$ as a product Gonon et al. (2024), one has:

$$\frac{\partial\phi_p}{\partial\theta_i}(\theta)\theta_i = \phi_p(\theta)1_{\{i \in p\}}.$$

In particular for $\theta = \mathbf{1}_D$ one gets:

$$S := \partial\phi(\mathbf{1}_D) = (1_{\{i \in p\}})_{p,i},$$

and thus:

$$D_{\phi(\theta)}S = \partial\phi(\theta)D_\theta, \tag{18}$$

where $D_{\phi(\theta)} = \operatorname{diag}(\phi(\theta))$ and $D_\theta = \operatorname{diag}(\theta)$. *Note that equation 18 holds for each $\theta \in \mathbb{R}^D$.*

Finally, as all coordinates of $\theta$ are in $\mathbb{R} \setminus \{0\}$, both $D_{\phi(\theta)}$ and $D_\theta$ are invertible and thus $\operatorname{rank}(S) = \operatorname{rank}(\partial\phi(\theta))$, which concludes the proof. $\square$

**Proposition F.7.** *For any DAG ReLU architecture, one has $\operatorname{rank}(\partial\phi_{ReLU}(\mathbf{1}_D)) \ge D - m$ with $m$ the number of hidden neurons and $D$ the number of parameters.*

*Proof.* Without loss of generality we prove the result on a connected DAG. Indeed, if the DAG has $C$ connected components, $G_k$, $1 \leq k \leq C$, each with $D_k$ edges and $m_k$ hidden neurons, then as $\sum_k D_k = D$ and $\sum_k m_k = m$ the result for the whole graph follows from the result on each component: up to row/column permutations, it is easy to check that the matrix $\partial\phi(\mathbf{1}_D)$ is the block-diagonal concatenation of the corresponding matrices for each component, hence its rank is $\sum_k(D_k - m_k) = D - m$.

Assume that $G$ is a connected DAG, and denote $S' := \partial\phi(\mathbf{1}_D)[F, :]$, the incidence matrix of $G$ restricted to $F$, the set of all full paths (i.e., paths starting from an input neuron and ending at an output neuron). As a direct consequence of the definition of $S'$, one gets: $\mathrm{range}(\partial\phi(\mathbf{1}_D)) \geq \mathrm{range}(S')$. Thus, to conclude the proof it is enough to show that $\mathrm{range}(S') = D - m$.

We now show that $\mathrm{range}(S') = D - m$. We will use the known fact (see (Gleiss et al., 2003, Proposition 6)) that in a strongly connected[2] directed graph $G' = (E', V')$, the rank of the cycle matrix $C$ is $|E'| - |V'| + 1$, where the cycle matrix has as many rows as the number of cycles in $G'$ and each row has binary entries indicating which edge (indexing the columns of $C$) belongs to the corresponding cycle. To exploit this fact, we complete $G$ (as a DAG, it has no cycle!) into a strongly connected graph $G'$ with cycle matrix $C$ satisfying $\mathrm{rank}(S') = \mathrm{rank}(C)$, and $|E'| = |E| + |V_{\mathrm{out}}| + |V_{\mathrm{in}}| + 1$, $|V'| = |V| + 2$, so that $\mathrm{rank}(S') = \mathrm{rank}(C) = |E| - |V| + |V_{\mathrm{out}}| + |V_{\mathrm{in}}| = D - m$ as claimed, since $m = |V| - (|V_{\mathrm{out}}| + |V_{\mathrm{in}}|)$. Above, we denoted $V_{\mathrm{in}}$ (resp. $V_{\mathrm{out}}$) the set of input (resp. output) neurons of $G$.

The construction is as follows: we add two fresh neurons $\mu, \nu$ ($V' := V \cup \{\mu, \nu\}$); the first one is a "source" neuron with an outgoing edge towards each input neuron in $V_{\mathrm{in}}$; the second is a "sink" neuron with an incoming edge from each output neuron in $V_{\mathrm{out}}$; another edge is added $\nu \to \mu$. It is not difficult to check that cycles of $G'$ are in bijection with paths in $G$: each cycle $c$ of $G'$ passes through exactly one input neuron and one output neuron, and its restriction to the edges of $G$ is precisely a path $p$ between these neurons. Vice-versa, every path $p$ in $G$, starting at $\gamma \in V_{\mathrm{in}}$ and ending at $\gamma' \in V_{\mathrm{out}}$ is uniquely prolongated into a cycle of $G'$ by adding the three edges $\gamma' \to \nu$, $\nu \to \mu$, and $\mu \to \gamma$. With a slight abuse of notation we will thus identify $c$ and $p$.

Equipped with this construction we first show that $S' = C[:, E]$, so that $\mathrm{rank}(S') \leq \mathrm{rank}(C)$. Indeed, $S'$ has binary entries $S'[p, e] = \mathbf{1}_{p \ni e}$, so that $S'[p, e] = C[c, e]$ for every path and $e \in E$.

To conclude, we only need to show $\mathrm{rank}(C) \leq \mathrm{rank}(S')$. For this we prove that each column $C[:, e']$, $e' \in E' \setminus E$, is in the span of $S'$. Denoting $\mathrm{out}(\gamma)$ (resp. $\mathrm{in}(\gamma)$) the set of outgoing (resp. incoming) edges of neuron $\gamma \in V$, and $\mathrm{suc}(\gamma)$ (resp. $\mathrm{ant}(\gamma)$) the set of successors (resp. antecedent) neurons of $\gamma$, i.e. neurons connected via an outgoing (resp. incoming) edge, we distinguish three cases:

- $e' = \mu \to \gamma$ with $\gamma \in V_{\mathrm{in}}$ some input neuron; a cycle $c$ contains this edge $e'$ if, and only if, the corresponding path $p$ of $G$ starts at the input neuron $\gamma$, or equivalently if there exists some neuron $\gamma' \in \mathrm{suc}(\gamma)$ such that $p$ contains the edge $\gamma \to \gamma'$. As $p$ can only pass through at most one of the outgoing edges of $\gamma$, we obtain $C[c, e'] = \sum_{e \in \mathrm{out}(\gamma)} S'[p, e]$, and since this holds for every $c$ we get $C[:, e'] = \sum_{e \in \mathrm{out}(\gamma)} S'[:, e]$, showing indeed that $C[:, e']$ belongs to the span of $S'$;

- $e' = \gamma \to \nu$, with $\gamma \in V_{\mathrm{out}}$; similarly, $C[:, e'] = \sum_{e \in \mathrm{in}(\gamma)} S'[:, e]$;

- $e' = \nu \to \mu$: this edge appears in all cycles hence $C[:, e'] = \mathbf{1}$. Since we can partition the cycles according to which edge $e'' = \mu \to \gamma$, $\gamma \in V_{\mathrm{in}}$ they contain, we have $\mathbf{1} = \sum_{\gamma \in V_{\mathrm{in}}} C[:, \mu \to \gamma]$, and since the first case have established that such $C[:, \mu \to \gamma]$ belong to the span of $S'$, so does $C[:, e'] = \mathbf{1}$.

$\square$

**Corollary F.8.** *For any DAG ReLU architecture, one has* $\mathrm{rank}(\partial\phi_{ReLU}(\mathbf{1}_D)) = D - m$ *with $m$ the number of hidden neurons and $D$ the number of parameters.*

Before showing this result, we first establish that the $m$ "canonical" conservation laws (Le & Jegelka, 2022) associated to each hidden neuron $1 \leq j \leq m$ are independent on a set that contains $\mathbf{1}_D$.

---

[2] a directed graph where for every nodes $\mu, \nu$ there exists a *directed* path starting at $\mu$ and ending at $\nu$

**Lemma F.9.** *For any DAG ReLU architecture, the "canonical" conservation laws obtained in Le & Jegelka (2022) associated to each hidden neuron $1 \leq j \leq m$, expressed (denoting $\theta_j$ the bias of neuron $j$, $\mathrm{succ}(j)/\mathrm{ant}(j)$ its outgoing/incoming edges, and $\theta_j$ the corresponding weights) as*

$$h_j(\theta) := \sum_{i \in \mathrm{succ}(j)} \theta_i^2 - \sum_{i \in \mathrm{ant}(j)} \theta_i^2 - \theta_j^2 \tag{19}$$

*are independent in the neighborhood of each parameter $\theta$ such that $\nabla h_j(\theta) \neq 0$ for all $j$ (so in particular around any $\theta \in \Theta$ with $\Theta$ the set of Theorem 3.2).*

*Proof.* Let us consider a parameter $\theta$ such that $\nabla h_j(\theta) \neq 0$ for all hidden neurons $j$.

For each edge $i = \mu \to \nu$ one has

$$\frac{\partial h_j}{\partial \theta_i}(\theta) = \begin{cases} 2\theta_i \text{ if } j = \mu \\ -2\theta_i \text{ if } j = \nu \\ 0 \quad \text{otherwise} \end{cases}$$

and similarly for each hidden neuron $j'$ with a bias

$$\frac{\partial h_j}{\partial \theta_{j'}}(\theta) = \begin{cases} -2\theta_{j'} \text{ if } j = j' \\ 0 \quad \text{otherwise} \end{cases},$$

denoting $h(\theta) := (h_j(\theta))_j \in \mathbb{R}^m$ the Jacobian matrix $\partial h(\theta) \in \mathbb{R}^{m \times D}$ has one or two nonzero entries per row: each row corresponds to one edge or one bias, each column to a hidden neuron, and a row has two nonzero entries at the input and output neurons of the corresponding edge (resp. one nonzero entry at the neuron corresponding to the bias).

By standard linear algebra one can show that $\partial h(\theta)$ has a trivial kernel, hence its column are linearly independent. By continuity this remains the case for the $m$ columns of $\partial h(\theta')$ in a neighborhood of $\theta$. This is exactly the definition of independent conservation laws.

$\square$

We now are able to prove Corollary F.8.

*Proof.* By Lemma F.9, there exists (at least) $m$ independent conservation laws around $\mathbf{1}_D$. Let us denote them $h_i$, $1 \leq i \leq m$. By definition of a conservation law (Definition 2.7) $h$ of $\phi_{\mathrm{ReLU}}$ we have that $\nabla h_i(\theta) \in \ker(\partial \phi_{\mathrm{ReLU}}(\theta))$, $1 \leq i \leq m$, and by definition of the *independence* of conservation laws, the vectors $\nabla h_i(\theta)$, $1 \leq i \leq m$, are linearly independent. It follows that

$$\dim(\ker(\partial \phi_{\mathrm{ReLU}}(\mathbf{1}_D))) \geq m.$$

By the rank-nullity theorem this implies

$$\mathrm{rank}(\partial \phi_{\mathrm{ReLU}}(\mathbf{1}_D)) \leq D - m,$$

and thus, by Proposition F.7, one gets

$$\mathrm{rank}(\partial \phi_{\mathrm{ReLU}}(\mathbf{1}_D)) = D - m. \qquad \square$$

**Corollary F.10.** *Consider a DAG ReLU architecture with $D$ parameters and $m$ hidden neurons. Assume that the underlying DAG $G$ is prunable at edge $i$. If $\theta$ satisfies $\theta_j \neq 0$ for every $j \neq i$, then $\mathrm{rank}(\partial \phi_{ReLU}(\theta)) = \mathrm{rank}(\partial \phi_{ReLU}(\mathbf{1}_D)) = D - m$.*

*Proof of Corollary F.10.* By Lemma F.6, the result holds if $\theta_i \neq 0$, hence we focus on the case where $\theta_i = 0$. Denote $\phi = \phi_{\mathrm{ReLU}}$, and $\mathcal{P}_i := \{p : p \ni i\}$ the set of all paths containing the edge $i$.

**1st step: We show that** $\mathrm{rank}(\partial \phi(\theta)) \geq \mathrm{rank}(\partial \phi(\mathbf{1}_D)) \overset{\text{Corollary F.8}}{=} D - m$**.** As a preliminary, observe that $\phi_p(\theta) = 0$ if, and only if, $p \in \mathcal{P}_i$, and that, denoting $S := \partial \phi(\mathbf{1}_D)$, we have

$$\frac{\partial \phi_p}{\partial \theta_j}(\theta) = \begin{cases} 0 & p \in \mathcal{P}_i, j \neq i, \\ \phi_{p \setminus \{i\}}(\theta) & p \in \mathcal{P}_i, j = i, \\ \phi_p(\theta) S_{p,j} \theta_j^{-1} & p \notin \mathcal{P}_i, j \neq i, \\ 0 & p \notin \mathcal{P}_i, j = i. \end{cases}$$

Thus one can write by blocks:

$$\partial\phi(\theta) = \begin{pmatrix} \mathbf{0} & \mathbf{u} \\ D_{\phi'(\theta')}(\theta')S'D'^{-1}_{\theta'} & \mathbf{0,} \end{pmatrix} \tag{20}$$

where $\mathbf{u} := (\phi_{p\setminus\{i\}})_{p\in\mathcal{P}_i}$ has no coordinate equal to zero, $\theta' := (\theta_j)_{j\neq i}$, $D_{\phi'(\theta')} := \mathrm{diag}(\phi_{\mathcal{P}_i^c}(\theta'))$, $D_{\theta'} := \mathrm{diag}(\theta')$ is invertible, and $S' := S[\mathcal{P}_i^c, \{i\}^c]$. Thus, we have

$$\mathrm{rank}(\partial\phi(\theta)) = 1 + \mathrm{rank}(D_{\phi'(\theta')}S'D_{\theta'}^{-1}) = 1 + \mathrm{rank}(S').$$

Let us now consider the smaller DAG $G_i'$ obtained by removing edge $i$ from the graph $G$. Since $G$ is prunable at $i$, by Lemma F.2 the set $\mathcal{P}_i^c$ is exactly the set of all paths of $G_i'$, so that $S' = \partial\phi'(\mathbf{1}_{D'})$ with $\phi'$ the path-lifting associated to $G_i'$, which is a DAG with $D' = D - 1$ parameters and $m' = m$ hidden neurons. By Corollary F.8 we obtain:

$$\mathrm{rank}(\partial\phi(\theta)) = 1 + \mathrm{rank}(S') \geq 1 + (D-1) - m = D - m \stackrel{\text{Corollary F.8}}{=} \mathrm{rank}(\partial\phi(\mathbf{1}_D)).$$

**2d step: We now show that** $\mathrm{rank}(\partial\phi(\theta)) \leq \mathrm{rank}(\partial\phi(\mathbf{1}_D)) + 1$. Indeed

$$\mathrm{rank}(\partial\phi(\mathbf{1}_D)) =: \mathrm{rank}(S) \geq \mathrm{rank}(D_{\phi(\theta)}S) \stackrel{\text{Equation (18)}}{=} \mathrm{rank}(\partial\phi(\theta)D_\theta) \geq \mathrm{rank}(\partial\phi(\theta)) - 1.$$

where in the last inequality we used again Equation (20), recalling that $\theta_i$ is the only coordinate of $\theta$.

**3rd step:   Conclusion.**   Using   the   two   first   steps,   one   has   $\mathrm{rank}(\partial\phi(\theta))$   $\in$ $\{\mathrm{rank}(\partial\phi(\mathbf{1}_D)), \mathrm{rank}(\partial\phi(\mathbf{1}_D)) + 1\} = \{D - m, D + 1 - m\}$. Assume by contradiction $\mathrm{rank}(\partial\phi(\theta)) = D + 1 - m$, i.e., there exists a subset of $D + 1 - m$ linearly independent columns of $\partial\phi(\theta)$. Then, by the continuity of $\phi$, there exists an open neighborhood $U$ of $\theta$ such that on $U$, these columns remain linearly independent, hence $\mathrm{rank}(\partial\phi_{\mathrm{ReLU}}(\theta')) \geq D + 1 - m$ for every $\theta' \in U$. In particular for each $\theta'$ in the non-empty set $U \cap (\mathbb{R}\setminus\{0\})^D$, we have $\mathrm{rank}(\partial\phi(\theta')) \geq D + 1 - m$, which contradicts Lemma F.6. This concludes the proof. $\qquad\square$

**Corollary F.11.** *Consider a DAG ReLU architecture with $D$ parameters that is prunable everywhere. For any $\theta$ with at most one zero component, one has $\mathrm{rank}(\partial\phi_{\mathrm{ReLU}}(\theta)) = \mathrm{rank}(\partial\phi_{\mathrm{ReLU}}(\mathbf{1}_D)) = D - m$, with $m$ the number of hidden neurons.*

*Proof.* By Corollary F.8 we have $\mathrm{rank}(\partial\phi_{\mathrm{ReLU}}(\mathbf{1}_D)) = D - m$ with $m$ the number of hidden neurons and $D$ the number of parameters. Since $\theta$ has at most one zero component and since the underlying graph is prunable at edge $i$ (as it is everywhere by assumption), we conclude using Corollary F.10. $\quad\square$

In particular, for an MLP architecture, this corollary reads:

**Corollary F.12.** *Consider an MLP architecture with $D$ parameters where every layer (including input and output ones) contains at least two neurons. For any $\theta$ with at most one zero component, one has $\mathrm{rank}(\partial\phi_{\mathrm{ReLU}}(\theta)) = \mathrm{rank}(\partial\phi_{\mathrm{ReLU}}(\mathbf{1}_D)) = D - m$, with $m$ the number of hidden neurons.*

*Proof.* By Corollary F.8 we have $\mathrm{rank}(\partial\phi_{\mathrm{ReLU}}(\mathbf{1}_D)) = D - m$ with $m$ the number of hidden neurons and $D$ the number of parameters. Since $\theta$ has at most one zero component, there is an edge $i$ in the graph $G$ underlying the MLP architecture such that $\theta_j \neq 0$ for each $j \neq i$. Moreover, since we focus on an MLP, by Corollary F.5 and our assumption on the layer widths, the underlying graph is prunable everywhere, hence it is prunable at edge $i$. We conclude using Corollary F.10. $\qquad\square$

### F.2.3   PROOF OF THEOREM 3.2

We can now establish Theorem 3.2. In fact, we prove a more general version of Theorem 3.2:

**Theorem F.13.** *Consider a DAG ReLU architecture that is prunable everywhere, and denote $\Theta \subseteq \mathbb{R}^D$ the set of all parameters with at most one zero component. On this set, which is open, dense, and with a single connected component, the parameterization $\phi_{\mathrm{ReLU}}$ of Gonon et al. (2024) satisfies the Frobenius property.*

*Proof.* It is each to check that $\Theta$ is an dense open set with a single connected component. Moreover for any $\theta \in \Theta$, denoting $\phi = \phi_{\texttt{ReLU}}$, we have

$$\dim \mathrm{Lie}(\mathbb{W}_\phi)(\theta) \geq \dim(\mathbb{W}_\phi(\theta)) = \mathrm{rank}(\partial\phi(\theta)) \stackrel{\text{Corollary F.11}}{=} \mathrm{rank}(\partial\phi(\mathbf{1}_D)) =: d.$$

To conclude we need to prove that this is indeed an equality. Assume by contradiction that there exists a parameter $\theta_1 \in \Theta$ such that $\dim \mathrm{Lie}(\mathbb{W}_\phi)(\theta_1) > d$. Then, by standard continuity arguments, this will remain the case on an open neighborhood $U$ of $\theta_1$, which is absurd by Theorem 3.1 as the Frobenius propery holds on $(\mathbb{R} \setminus \{0\})^D$, a dense open subset of $\Theta$. $\qquad\square$

### F.3 PROOF OF COROLLARY 3.3

We prove a more general version of Corollary 3.3.

**Corollary F.14.** *For a general DAG ReLU network (resp. for a DAG ReLU network prunable everywhere), every element $\theta$ of the set $(\mathbb{R} \setminus \{0\})^D$ (resp. of the set $\Theta$ of Theorem 3.2/Theorem F.13) admits an open neighborhood $U$ in this set on which the intrinsic recoverability property (and thus the intrinsic dynamic property with respect to $\phi_{\texttt{ReLU}}$) holds.*

*Proof.* Let us consider a DAG ReLU network (resp. a DAG ReLU architecture prunable everywhere). We can apply Lemma F.6 (resp. Corollary F.11) to obtain an open dense set $\Theta$ (resp. with a single connected component) on which the dimension of the trace of $\mathbb{W}_\phi$ remains constant. By Theorem 3.1 (resp. Theorem F.13), $\phi_{\texttt{ReLU}}$ satisfies the Frobenius property on $\Theta$. By Proposition 2.21 every $\theta_0 \in \Theta$ admits a neighborhood $U$ on which it satisfies the intrinsic recoverability property with respect to $\phi_{\texttt{ReLU}}$. By Lemma 2.16 such a parameter $\theta_0$ also satisfies the intrinsic metric property on $U$ with respect to $\phi_{\texttt{ReLU}}$. $\qquad\square$

### F.4 PROOF OF COROLLARY 3.4.

We prove a more general version of Corollary 3.4.

**Corollary F.15.** *Let us consider a DAG ReLU architecture that is prunable everywhere, and with $m$ hidden neurons. Then the $m$ independent conservation laws for $\phi_{ReLU}$ given in Proposition 2.9 are exhaustive on the set $\Theta$ from Theorem F.13: there is no more conservation laws than these ones.*

*Proof.* On $\Theta$, we know that the dimension of $\mathrm{Lie}(\mathbb{W}_{\phi_{\texttt{ReLU}}})(\theta)$ is constant and equal to $\mathrm{range}(\partial\phi_{\texttt{ReLU}}(\mathbf{1}_D)) = D - m$ by Theorem 3.2 and Corollary F.11. Then by (Marcotte et al., 2023, Theorem 3.3), there exists exactly $m$ independent conservation laws locally on $\Theta$. Thus the $m$ independent conservation laws for $\phi_{\texttt{ReLU}}$ on $\Theta$ given in Lemma F.9 are exhaustive. $\qquad\square$

## G PROOF OF PROPOSITION 3.5

**Lemma G.1.** *Let $Y \in \mathbb{R}_{>0}^{n \times m}$. Then there exists a unique pair $(\boldsymbol{\alpha}, \boldsymbol{\beta}) =: \Gamma(Y)$ of vectors $\boldsymbol{\alpha} \in \mathbb{R}_{>0}^n$, $\boldsymbol{\beta} \in \mathbb{R}_{>0}^m$ such that*

$$\boldsymbol{\alpha}^2 = Y \mathrm{diag}(\boldsymbol{\beta})^{-1}\mathbf{1}_m, \quad and \quad \boldsymbol{\beta}^2 = Y^\top \mathrm{diag}(\boldsymbol{\alpha})^{-1}\mathbf{1}_n.$$

*Proof.* Define the mappings

$$S(\boldsymbol{\beta}) := \sqrt{Y\mathrm{diag}(\boldsymbol{\beta})^{-1}\mathbf{1}_m}, \quad T(\boldsymbol{\alpha}) := \sqrt{Y^\top \mathrm{diag}(\boldsymbol{\alpha})^{-1}\mathbf{1}_n}.$$

Let $D_n(a, a') := \|\log(a/a')\|_\infty$ denote the Thompson metric on $(\mathbb{R}_{>0})^n$ (and similarly $D_m$ on $(\mathbb{R}_{>0})^m$), where $\mathbb{R}_+^*$ is the set of positive real numbers. It is known that $((\mathbb{R}_{>0})^n, D_n)$ (resp. $((\mathbb{R}_{>0})^m, D_m)$) is a complete metric space. The linear operator $Y$ is 1-Lipschitz from $((\mathbb{R}_{>0})^m, D_m)$ to $((\mathbb{R}_{>0})^n, D_n)$, according to the Birkhoff contraction theorem. Moreover, the square root function is $\frac{1}{2}$-Lipschitz in this metric. Hence, the composition $S \circ T$ is $\frac{1}{4}$-contracting with respect to $D_n$. By the Banach fixed-point theorem, there exists a unique fixed point of $S \circ T$, which implies the existence and uniqueness of the pair $(\boldsymbol{\alpha}, \boldsymbol{\beta})$ solving the original equations. $\qquad\square$

**Proposition 3.5.** *For a 3-layer ReLU MLP with scalar input/output, the factorization $\phi_{\texttt{ReLU}}$ reads[3]*

$$Z = \phi_{\texttt{ReLU}}(u, V, w) := \operatorname{diag}(u)\, V \operatorname{diag}(w) \in \mathbb{R}^{n \times m},$$

*with $u \in \mathbb{R}^n$, $V \in \mathbb{R}^{n \times m}$, and $w \in \mathbb{R}^m$. Define $\Theta := \{(u, V, w) : u_i, V_{ij}, w_j \neq 0\ \forall i, j\}$, and consider the $n + m$ conservation laws $\mathbf{h}(\theta) := \big((u_i^2 - \sum_j V_{ij}^2)_{i=1}^n, (w_j^2 - \sum_i V_{ij}^2)_{j=1}^m\big)$ for $\phi_{ReLU}$. Every $\theta_0 \in \Theta$ satisfies the intrinsic dynamics with respect to $\phi_{ReLU}$, which reads $\dot{z} = -K_{\theta_0}(z)\nabla f(z)$ with $z = \texttt{vec}(Z)$ corresponding to*

$$\dot{Z} = -\operatorname{ddiag}(\nabla f(Z) Z^\top)\operatorname{diag}(\boldsymbol{\alpha})^{-1} Z - \operatorname{diag}(\boldsymbol{\alpha})\,\nabla f(Z)\operatorname{diag}(\boldsymbol{\beta}) - Z\operatorname{diag}(\boldsymbol{\beta})^{-1}\operatorname{ddiag}(Z^\top \nabla f(Z)),$$

*where: a) for any matrix $M$, $\operatorname{ddiag}(M) := \operatorname{diag}\big(\operatorname{Diag}(M)\big)$, where $\operatorname{Diag}(M)$ extracts its diagonal as a vector and $\operatorname{diag}(v)$ is the diagonal matrix with entries of $v$; and b) the vectors $\boldsymbol{\alpha} = \boldsymbol{\alpha}(Z, \mathbf{h}(\theta_0)) \in \mathbb{R}_{>0}^n$ and $\boldsymbol{\beta} := \boldsymbol{\beta}(Z, \mathbf{h}(\theta_0)) \in \mathbb{R}_{>0}^m$ (uniquely determined by $Z$ and $\mathbf{h}(\theta_0)$) satisfy*

$$\boldsymbol{\alpha}^2 - |Z|^2\operatorname{diag}(\boldsymbol{\beta})^{-1}\mathbf{1}_n - \boldsymbol{\lambda}\odot\boldsymbol{\alpha} = 0, \quad \boldsymbol{\beta}^2 - (|Z|^2)^\top\operatorname{diag}(\boldsymbol{\alpha})^{-1}\mathbf{1}_m - \boldsymbol{\mu}\odot\boldsymbol{\beta} = 0, \quad (10)$$

*with $|Z|^2 \in \mathbb{R}^{n \times m}$ the element-wise square on the matrix $Z \in \mathbb{R}^{n \times m}$ and with $\boldsymbol{\lambda} \in \mathbb{R}^n$, $\boldsymbol{\mu} \in \mathbb{R}^m$ such that $\mathbf{h}(\theta_0) = (\boldsymbol{\lambda}, \boldsymbol{\mu})$. When $\boldsymbol{\lambda}, \boldsymbol{\mu} = 0$, equation 10 entirely characterizes $(\boldsymbol{\alpha}, \boldsymbol{\beta})$.*

*Proof.* Given the general definition of $\phi_{\texttt{ReLU}}$ (see e.g. Neyshabur et al. (2015); Stock & Gribonval (2022); Gonon et al. (2024)), we study the factorization map

$$\phi(u, V, w) := \operatorname{diag}(u)\, V \operatorname{diag}(w),$$

where $u \in \mathbb{R}^n$, $V \in \mathbb{R}^{n \times m}$, $w \in \mathbb{R}^m$ with $u_i, w_j \neq 0$.

**Step 1: Gradient flow in parameters.**

Let $f : \mathbb{R}^{n \times m} \to \mathbb{R}$ and define the loss $\ell(u, V, w) = f\big(\phi(u, V, w)\big)$. Writing $Z = \phi(u, V, w)$ and its gradient $G = \nabla f(Z)$, the gradient-flow ODE equation 1 $\dot{u} = -\partial_u\ell$, $\dot{V} = -\partial_V\ell$, $\dot{w} = -\partial_w\ell$ is:

$$\dot{u} = -\operatorname{Diag}\big(G\operatorname{diag}(w)\, V^\top\big),$$
$$\dot{V} = -\operatorname{diag}(u)\, G\operatorname{diag}(w),$$
$$\dot{w} = -\operatorname{Diag}\big(V^\top\operatorname{diag}(u)\, G\big),$$

**Step 2: Induced flow on $z$.**

Since $Z = \operatorname{diag}(u) V \operatorname{diag}(w)$, we have

$$\dot{Z} = \operatorname{diag}(\dot{u}) V \operatorname{diag}(w) + \operatorname{diag}(u)\dot{V}\operatorname{diag}(w) + \operatorname{diag}(u) V \operatorname{diag}(\dot{w}).$$

Substituting the above yields

$$\dot{Z} = -\operatorname{ddiag}\big(G\operatorname{diag}(w) V^\top\big) V \operatorname{diag}(w) - \operatorname{diag}(u^2)\, G\operatorname{diag}(w^2) - \operatorname{diag}(u) V \operatorname{ddiag}\big(V^\top\operatorname{diag}(u) G\big),$$

where we set $\operatorname{ddiag}(M) = \operatorname{diag}\big(\operatorname{Diag}(M)\big)$.

Eliminating $V$ via $V = \operatorname{diag}(u)^{-1} Z \operatorname{diag}(w)^{-1}$ (possible as $u_i, w_j \neq 0$ on $\Theta$) and using $\operatorname{ddiag}(M\operatorname{diag}(a)) = \operatorname{ddiag}(M)\operatorname{diag}(a)$ one obtains

$$\dot{Z} = -\operatorname{ddiag}(G z^\top)\operatorname{diag}(u^{-2}) Z - \operatorname{diag}(u^2)\, G\operatorname{diag}(w^2) - Z\operatorname{diag}(w^{-2})\operatorname{ddiag}(Z^\top G).$$

Moreover by Corollary 3.3 there exists conservation laws $\mathbf{h}$ and a function $\Gamma$ such that $\theta = (u, V, w) = \Gamma(\phi(\theta), \mathbf{h}(\theta)) = \Gamma(Z, \mathbf{h}(\theta))$ so that $\boldsymbol{\alpha} := u^2$ and $\boldsymbol{\beta} := w^2$ (entrywise multiplication) can both be expressed as functions $\boldsymbol{\alpha}(Z, \mathbf{h}(\theta))$ and $\boldsymbol{\beta}(Z, \mathbf{h}(\theta))$. Below we explicit such conservation laws and characterize properties of $\boldsymbol{\alpha}$ and $\boldsymbol{\beta}$.

**Step 3: Conserved quantities and elimination of $\boldsymbol{\alpha}, \boldsymbol{\beta}$.**

The flow equation 1 preserves the following $n + m$ conservation laws:

$$\forall i = 1, \ldots, n : \quad u_i^2 - \sum_{j=1}^m V_{ij}^2 = \lambda_i,$$

---

[3]When written as a $n \times m$ matrix, we denote $Z$ instead of $z$ and also view $\nabla f(Z)$ as an $n \times m$ matrix.

$$\forall j = 1, \ldots, m: \quad w_j^2 - \sum_{i=1}^{n} V_{ij}^2 = \mu_j,$$

for given constants $\boldsymbol{\lambda} \in \mathbb{R}^n$ and $\boldsymbol{\mu} \in \mathbb{R}^m$ determined by $\theta_0$. Since $V_{ij} = Z_{ij}/(u_i w_j)$, then $(u^2, w^2) > 0$ is a solution of the coupled system

$$u^2: \ u_i^4 - \sum_{j=1}^{m} \frac{Z_{ij}^2}{w_j^2} - \lambda_i \, u_i^2 = 0,$$

$$w^2: \ w_j^4 - \sum_{i=1}^{n} \frac{Z_{ij}^2}{u_i^2} - \mu_j \, w_j^2 = 0.$$

In vector-matrix form (with entrywise squaring):

$$\boldsymbol{\alpha} = u^2, \quad \boldsymbol{\beta} = w^2,$$

$$\boldsymbol{\alpha}^2 - |Z|^2 \operatorname{diag}(\boldsymbol{\beta})^{-1} \mathbf{1}_m - \boldsymbol{\lambda} \odot \boldsymbol{\alpha} = 0, \quad \boldsymbol{\beta}^2 - (|Z|^2)^\top \operatorname{diag}(\boldsymbol{\alpha})^{-1} \mathbf{1}_n - \boldsymbol{\mu} \odot \boldsymbol{\beta} = 0,$$

where $|Z|^2$ is the elementwise square of $Z$ and $\odot$ is the element-wise product.

**Special case $\boldsymbol{\lambda} = 0, \boldsymbol{\mu} = 0$.**

Then the system reduces to

$$\boldsymbol{\alpha}^2 = (|Z|^2) \operatorname{diag}(\boldsymbol{\beta})^{-1} \mathbf{1}_m, \quad \boldsymbol{\beta}^2 = (|Z|^2)^\top \operatorname{diag}(\boldsymbol{\alpha})^{-1} \mathbf{1}_n. \tag{21}$$

By Lemma G.1 with $Y = |Z|^2$ (possible as $Z_{ij} = u_i V_{ij} w_j \neq 0$ since $\theta \in \Theta$), the exists a unique solution $(\boldsymbol{\alpha}, \boldsymbol{\beta}) > 0$ of the system equation 21.

In the scalar case ($n = m = 1$) with $|Z|^2 = z^2$ a scalar, the solution is $\boldsymbol{\alpha} = \boldsymbol{\beta} = (|Z|^2)^{1/3} = |z|^{2/3}$. $\qquad \square$

## H   PROOF OF THEOREM 3.8.

**Theorem 3.8.** *Consider $\theta_0 := (U_{t=0}, V_{t=0})$ where both $U_{t=0} \in \mathbb{R}^{n \times r}$ and $V_{t=0} \in \mathbb{R}^{m \times r}$ have full rank $r \leq \min(n, m)$, and assume $S = \lambda \operatorname{Id}_r$ for some $\lambda \in \mathbb{R}$. Then, on a neighborhood $\Omega$ of $\theta_{t=0}$:*

$$\dot{Z} = -\Pi_Z \left[ \frac{\lambda}{2} \operatorname{Id}_n + \frac{1}{2} \sqrt{\lambda^2 \operatorname{Id}_n + 4\, ZZ^\top} \right] \nabla f(Z) - \nabla f(Z) \Pi_{Z^\top Z} \left[ -\frac{\lambda}{2} \operatorname{Id}_m + \frac{1}{2} \sqrt{\lambda^2 \operatorname{Id}_m + 4\, Z^\top Z} \right], \tag{14}$$

*where $\Pi_A$ is the orthogonal projector on $\operatorname{range} A$.*

*Proof.* **Step 1: rank of $Z$.**

As $r \leq \min(n, m)$ and as both $U_{t=0} \in \mathbb{R}^{n \times r}$ and $V_{t=0} \in \mathbb{R}^{m \times r}$ have full rank equal to $r$, it remains the case in a neighborhood $\Omega$ of $\theta_0 := (U_{t=0}, V_{t=0})$, and it is also the case for $Z = UV^\top$.

**Step 2: A quadratic equation for $P := UU^\top$.**

Compute
$$ZZ^\top = UV^\top V U^\top = U(V^\top V)U^\top.$$

With the hypothesis $U^\top U - V^\top V = \lambda \operatorname{Id}_r$ we get $V^\top V = U^\top U - \lambda \operatorname{Id}_r$, hence

$$ZZ^\top = U(U^\top U - \lambda \operatorname{Id}_r)U^\top = UU^\top UU^\top - \lambda UU^\top = P^2 - \lambda P.$$

Thus $P$ satisfies the quadratic matrix equation

$$P^2 - \lambda P - ZZ^\top = 0. \tag{22}$$

**Step 3: Simultaneous diagonalisation and scalar reduction.**

Write $Z' := ZZ^\top$. Because

$$P = UU^\top, \qquad Z' = U(V^\top V)U^\top,$$

and $U^\top U$ differs from $V^\top V$ only by a scalar multiple of the identity, we have $(U^\top U)(V^\top V) = (V^\top V)(U^\top U)$. Encapsulating by $U$ and $U^\top$ yields $PZ' = Z'P$. Hence $P$ and $Z'$ are *simultaneously diagonalisable*: there exists an orthogonal matrix $W \in \mathbb{R}^{n \times n}$ such that

$$P = W \operatorname{diag}(\sigma_1, \ldots, \sigma_n) W^\top, \qquad Z' = W \operatorname{diag}(\mu_1, \ldots, \mu_n) W^\top,$$

with $\sigma_i, \mu_i \geq 0$ and where we assume $\sigma_1 \geq \cdots \geq \sigma_n$ and $\mu_1 \geq \cdots \geq \mu_n$.

In the common eigenbasis, equation 22 becomes for every $i$

$$\sigma_i^2 - \lambda \sigma_i - \mu_i = 0.$$

Its two roots are

$$\sigma_i^{\pm} = \frac{\lambda \pm \sqrt{\lambda^2 + 4\mu_i}}{2}.$$

By the first step, one already has that on $\Omega$, for any $i > r$: $\sigma_i = \mu_i = 0$ so that $\sigma_i = \sigma_i^-$, and that for any $i \leq r$, $\sigma_i > 0$ and $\mu_i > 0$. Thus $\sqrt{\lambda^2 + 4\mu_i} > |\lambda|$, the "$-$" root is negative, while $P = UU^\top$ is positive-semidefinite. Therefore $\sigma_i = \sigma_i^+$ for $i \leq r$. Let us define $\Pi_Z := W \operatorname{diag}(\underbrace{1, \cdots, 1}_{\times r}, 0, \cdots, 0) W^\top$ the orthogonal projector on $\operatorname{range}(Z) = \operatorname{range}(ZZ^\top)$. It follows that:

$$P = \Pi_Z \times \left[ \frac{\lambda}{2} \operatorname{Id}_n + \frac{1}{2} \sqrt{\lambda^2 \operatorname{Id}_n + 4 \, ZZ^\top} \right]. \tag{23}$$

**Step 4: The expression for $Q := VV^\top$.** A fully analogous computation gives

$$Z^\top Z = VU^\top UV^\top = V(V^\top V + \lambda \operatorname{Id}_r) V^\top = Q^2 + \lambda Q,$$

so that $Q$ satisfies

$$Q^2 + \lambda Q - Z^\top Z = 0. \tag{24}$$

Because $Q$ and $T := Z^\top Z$ commute, they share an orthonormal eigenbasis in which equation 24 reduces to

$$\tau_i^2 + \lambda \tau_i - \mu_i = 0 \quad (\tau_i \geq 0, \ \mu_i \geq 0).$$

By the first step, one already has that on $\Omega$, for any $i > r$: $\tau_i = \mu_i = 0$ and that for any $i \leq r$, $\tau_i \neq 0$ and $\mu_i \neq 0$. For $i \leq r$ the positive root (as $\sqrt{\lambda^2 + 4\mu_i} > |\lambda|$) is

$$\tau_i = \frac{-\lambda + \sqrt{\lambda^2 + 4\mu_i}}{2},$$

so that

$$Q = \Pi_{Z^\top Z} \times \left[ -\frac{\lambda}{2} \operatorname{Id}_m + \frac{1}{2} \sqrt{\lambda^2 \operatorname{Id}_m + 4T} \right], \tag{25}$$

with $T = Z^\top Z$ and where $\Pi_{Z^\top Z}$ is the orthogonal projector on $\operatorname{range}(Z^\top Z)$.

**Step 5: Uniqueness and conclusion** In both cases equation 23–equation 25 are the only solutions consistent with $UU^\top \succeq 0$ and $VV^\top \succeq 0$ and with $\operatorname{rank}(Z) = r$ on $\Omega$. Finally one has on $\Omega$:

$$\dot{Z} = -UU^\top \nabla f(Z) - \nabla f(Z) VV^\top$$

$$= -\Pi_Z \left[ \frac{\lambda}{2} \operatorname{Id}_n + \frac{1}{2} \sqrt{\lambda^2 \operatorname{Id}_n + 4 \, ZZ^\top} \right] \nabla f(X) - \nabla f(X) \Pi_{Z^\top Z} \left[ -\frac{\lambda}{2} \operatorname{Id}_m + \frac{1}{2} \sqrt{\lambda^2 \operatorname{Id}_m + 4 \, Z^\top Z} \right],$$

which concludes the proof. $\qquad \square$

## I    PROOF OF THEOREM 3.9

We first show the following lemma:

**Lemma I.1.** *If $S \neq \lambda \operatorname{Id}_r$ with $S$ a real symmetric matrix, then there exists a skew-symmetric matrix $A$ such that $[A, S] \neq 0$.*

*Proof.* Let us assume $S \neq \lambda \mathrm{Id}_r$ (in particular $r > 1$ necessarily). Thus there are at least two distinct eigenvalues of $S$ $\delta$ and $\mu$ associated to the eigenvectors $x$ and $y$. Then $A := xy^\top - yx^\top \neq 0$ is a skew-symmetric matrix that satisfies:

$$\begin{aligned}
[A, S] &= (xy^\top - yx^\top)S - S(xy^\top - yx^\top) \\
&= x(Sy)^\top - y(Sx)^\top - (Sx)y^\top + (Sy)x^\top \text{ as } S \text{ is symmetric} \\
&= \mu xy^\top - \delta yx^\top - \delta xy^\top + \mu yx^\top \\
&= \underbrace{(\mu - \delta)}_{\neq 0}(xy^\top + yx^\top) \neq 0,
\end{aligned}$$

as $\mu \neq \delta$, and which concludes the proof. $\qquad\square$

We introduce a slightly stronger version of Definition 2.6 involving *piecewise $\mathcal{C}^2$* functions $f$.

**Definition I.2** (Strong intrinsic dynamic property). $\theta_0$ verifies the *strong intrinsic dynamic property* with respect to $\phi$ on some open set $\Omega \ni \theta_0$, if there is $K_{\theta_0} : \mathbb{R}^d \to \mathbb{R}^{d \times d}$ such that: if $\theta(\cdot) \in \mathcal{C}^0$ satisfies $\theta(0) = \theta_0$ and $\dot\theta(t) = -\nabla \ell(\theta(t))$ with $\ell = f \circ \phi$, where $f$ is *piecewise $\mathcal{C}^2$*, whenever $\theta(t) \in \Omega$ and $f$ is differentiable at $\phi(\theta(t))$, then $M(\theta(t)) = K_{\theta_0}(\phi(\theta(t)))$ holds for each $t$ such that $\theta(t) \in \Omega$.

**Theorem 3.9.** *Let $\theta_0 := (U_{t=0}, V_{t=0})$. Assume that both $U_{t=0} \in \mathbb{R}^{n \times r}$ and $V_{t=0} \in \mathbb{R}^{m \times r}$ have full rank and that $r \leq max(n, m)$. If $S := U_{t=0}^\top U_{t=0} - V_{t=0}^\top V_{t=0} \neq \lambda \mathrm{Id}_r$, then $\theta_0$ does not satisfy the* strong *intrinsic dynamic property with respect to $\phi_{\mathtt{Lin}}$.*

*Proof.* **A. First we show that such an initialization does not satisfy the intrinsic *metric* property.** In light of the necessary condition of Theorem 2.14 we will first characterize $\ker \partial M(\theta)$ for any $\theta = (U, V)$. Then, with $\mathbf{h}(\theta) = U^\top U - V^\top V$ and $\phi(\theta) = \phi_{\mathtt{Lin}}(\theta) = UV^\top$, we will exhibit a subspace $\mathcal{V}$ of $\ker \partial \mathbf{h}(\theta) \cap \ker \partial \phi(\theta)$ such that $\mathcal{V} \not\subset \ker \partial M(\theta)$. We will then conclude using the needed calculus and Theorem 2.14.

**Step 1: Characterization of $\ker \partial M(\theta)$ for any $\theta = (U, V)$.**

By equation 11 (with $L = 2$, $U_2 = U, U_1 = V^\top$), one can write $M(\theta)\mathrm{vec}(X) = \mathrm{vec}(UU^\top X + XVV^\top)$ for any matrix $X \in \mathbb{R}^{n \times m}$. Using the Kronecker product and the fact that $(A \otimes B)\mathrm{vec}(X) = \mathrm{vec}(BXA^\top)$, this expression can be rewritten as:

$$M(\theta) = \mathrm{Id}_m \otimes (UU^\top) + (VV^\top) \otimes \mathrm{Id}_n. \tag{26}$$

Thus differentiating equation 26 yields that for any $(H, K)$ of the same dimensions as $(U, V)$ we have $\partial M(\theta).(H, K) = \mathrm{Id}_m \otimes (UH^\top + HU^\top) + (VK^\top + KV^\top) \otimes \mathrm{Id}_n$, and thus: $(H, K) \in \ker \partial M(\theta)$ if and only if $\mathrm{Id}_m \otimes (UH^\top + HU^\top) = -(VK^\top + KV^\top) \otimes \mathrm{Id}_n$. We now show that

$$\ker \partial M(\theta) = \left\{ (H, K) : \exists \mu \in \mathbb{R}, UH^\top + HU^\top = \mu \mathrm{Id}_n \text{ and } VK^\top + KV^\top = -\mu \mathrm{Id}_m \right\}. \tag{27}$$

The converse inclusion is clear. We now prove the direct inclusion. Let us consider $(H, K) \in \ker \partial M(\theta)$, then one has $\mathrm{Id}_m \otimes (UH^\top + HU^\top) = -(VK^\top + KV^\top) \otimes \mathrm{Id}_n$. Still using that $(A \otimes B)\mathrm{vec}(X) = \mathrm{vec}(BXA^\top)$ and denoting $U' := UH^\top + HU^\top$ and $V' := (VK^\top + KV^\top)$, this exactly means that for any matrix $X \in \mathbb{R}^{n \times m}$ one has $U'X = -XV'^\top$. To conclude, we only need to show that this implies the existence of $\mu, \mu' \in \mathbb{R}$ such that $U' = \mu \mathrm{Id}_n$ and $V' = -\mu' \mathrm{Id}_m$, since the equality $U'X = -XV'^\top$ then also implies $\mu = \mu'$. This is immediate if $V' = 0$ since in this case $U'$ must also be equal to zero as $U'X = 0$ for every $X$. Assume now that $V'$ is non-zero so there exists a vector $v$ such that $V'^\top v \neq 0$. Considering any such $v$ and any vector $u$, and setting $X = uv^\top$, we have

$$(U'u)v^\top = U'X = -XV' = -u(V'^\top v)^\top$$

hence $U'u$ is colinear with $u$. Since this holds for any choice of $u$, we deduce indeed that $U'$ is proportional to $\mathrm{Id}_n$. A similar reasoning yields that $V' \propto \mathrm{Id}_m$. This concludes the proof of equation 27.

**Step 2: Characterization of a subspace $\mathcal{V} \subseteq \ker \partial \mathbf{h}(\theta) \cap \ker \partial \phi(\theta)$.** Since $\mathbf{h}(\theta) = U^\top U - V^\top V$ and $\phi(\theta) = UV^\top$ we have

$$\partial h(\theta).(H, K) = U^\top H + H^\top U - V^\top K - K^\top V$$

$$\partial \phi(\theta).(H, K) = UK^\top + HV^\top$$

and one can easily check that for any $\theta$ such that $\mathbf{h}(\theta) = S$ we have

$$\mathcal{V} := \left\{ \begin{pmatrix} U\Delta \\ -V\Delta^\top \end{pmatrix} : \Delta \in \mathbb{R}^{r \times r}, (\Delta^\top + \Delta) U^\top U + U^\top U (\Delta + \Delta^\top) = \Delta S + S \Delta^\top \right\}$$
$$\subseteq \ker\partial\mathbf{h}(\theta) \cap \ker\partial\phi(\theta).$$

**Step 3: Proof that $\mathcal{V} \not\subset \ker\partial M(\theta)$.** The fact that a matrix $\Delta \in \mathbb{R}^{r \times r}$ satisfies

$$(\Delta^\top + \Delta) U^\top U + U^\top U (\Delta + \Delta^\top) = \Delta S + S \Delta^\top,$$

is equivalent to

$$\Delta_S (2U^\top U - S) + (2U^\top U - S)\Delta_S = [\Delta_A, S],$$

with $\Delta_S$ (resp. $\Delta_A$) the symmetric (resp. skew-symmetric) part of $\Delta$ (so that $\Delta = \Delta_S + \Delta_A$). Denote $\mathcal{S}_r$ (resp. $\mathcal{A}_r$) the set of $r \times r$ symmetric (resp. skew-symmetric) matrices and $L$ the Lyapunov operator defined by:

$$L : \Delta_S \in \mathcal{S}_r \mapsto L(\Delta_S) := \Delta_S (2U^\top U - S) + (2U^\top U - S)\Delta_S$$
$$= \Delta_S (U^\top U + V^\top V) + (U^\top U + V^\top V)\Delta_S \in \mathcal{S}_r$$

We obtain

$$\mathcal{V} = \left\{ \begin{pmatrix} U(\Delta_S + \Delta_A) \\ -V(\Delta_S - \Delta_A) \end{pmatrix} : (\Delta_S, \Delta_A) \in \mathcal{S}_r \times \mathcal{A}_r, L(\Delta_S) = [\Delta_A, S] \right\}.$$

As $S \neq \lambda\mathrm{Id}_r$, by Lemma I.1 there exists a skew-symmetric matrix $\Delta_A \in \mathcal{A}_r$ such that $[\Delta_A, S] \neq 0$. As $U^\top U + V^\top V$ is positive definite (as either $U$ or $V$ has full column-rank) its eigenvalues $\lambda_i > 0$ satisfy $\lambda_i + \lambda_j \neq 0$, so (see e.g. Bartels & Stewart (1972)) in particular the Lyapunov operator: $L : \mathcal{S}_r \to \mathcal{S}_r$ is invertible. Since $[\Delta_A, S] = \Delta_A S - S\Delta_A \in \mathcal{S}_r$, we obtain that there exists $\Delta_S \neq 0$ such that $L(\Delta_S) = [\Delta_A, S]$. This particular choice of $\Delta_S$ and $\Delta_A$ exhibits a parameter $\theta' = (U\Delta, -V\Delta^\top)$ that satisfies $\theta' \in \mathcal{V} \subseteq \ker\partial\phi(\theta)\cap\ker\partial\mathbf{h}(\theta)$. We now show that $\theta' \notin \ker\partial M(\theta)$. We proceed by contradiction: if $\theta' \in \ker\partial M(\theta)$ then, by equation 27, there exists $\mu \in \mathbb{R}$ such that $U(\Delta^\top + \Delta)U^\top = \mu\mathrm{Id}_n$ and $V(\Delta^\top + \Delta)V^\top = -\mu\mathrm{Id}_m$ that is to say

$$2U\Delta_S U^\top = \mu\mathrm{Id}_n \quad \text{and} \quad 2V\Delta_S V^\top = -\mu\mathrm{Id}_m. \tag{28}$$

When $r \leq \max(m, n)$ and since $U, V$ are full rank, at least one of the two matrices $U$ or $V$ is full column rank $r$. Without loss of generality let us assume that $U$ is full column rank. Then $U^\top U$ is invertible and we deduce that,

$$2\Delta_S = \mu(U^\top U)^{-1}. \tag{29}$$

Moreover if (as we indeed show below) $\mathrm{range}(U^\top) \cap \mathrm{range}(V^\top) \neq \{0\}$, then by considering $z = U^\top x = V^\top y \neq 0$ for some $x \in \mathbb{R}^n$ and $y \in \mathbb{R}^m$, one deduces from equation 28 that $\mu\|x\|_2^2 = 2x^\top U\Delta_S U^\top x = z^\top \Delta_S z = 2y^\top V\Delta_S V^\top y = -\mu\|y\|_2^2$ and thus $\mu = 0$. Hence $\Delta_S = 0$ by equation 29, contradicting $L(\Delta_S) = [\Delta_A, S] \neq 0$, which shows that $\theta' \notin \ker\partial M(\theta)$.

Thus we only need to prove that one has $\mathrm{range}U^\top \cap \mathrm{range}V^\top \neq \{0\}$, and indeed:

$$\dim(\mathrm{range}(U^\top) \cap \dim(\mathrm{range}(V^\top)) = \underbrace{\mathrm{rank}(U^\top)}_{=\mathrm{rank}(U)} + \underbrace{\mathrm{rank}(V^\top)}_{=\mathrm{rank}(V)} - \dim(\underbrace{\mathrm{range}(U^\top) + \mathrm{range}(V^\top)}_{\mathrm{range}((U^\top \mid V^\top))})$$

$$= \underbrace{\mathrm{rank}(U) + \mathrm{rank}(V)}_{\geq \min(r,n)+\min(r,m)\geq r+1} - \underbrace{\mathrm{rank}\left(\begin{pmatrix} U \\ V \end{pmatrix}\right)}_{=r} > 0,$$

where we used in the last line that $r \leq \max(n, m)$.

**Step 4: Conclusion.**

As both $U_{t=0}$ and $V_{t=0}$ have full rank it remains the case in a neighborhood $\Omega$ of $\theta_0$. Moreover as $r \leq \max(n, m)$ then one of the two matrices has full column rank on $\Omega$. In particular the vertical concatenation $\begin{pmatrix} U \\ V \end{pmatrix}$ has full rank (equal to $r$) on $\Omega$ as $r \leq \max(n, m) \leq n + m$.

Since $\left(\begin{smallmatrix} U \\ V \end{smallmatrix}\right)$ has full rank on $\Omega$, by (Marcotte et al., 2023, Proposition 4.2 and Corollary 4.4) the vector-valued function $\mathbf{h}$ contains a complete set of conservation laws.

We now show by contradiction that for any $\Omega' \subseteq \Omega$, $\theta_0$ does not satisfy the intrinsic metric property on $\Omega'$. Let us assume there exists a neighborhood $\Omega' \subseteq \Omega$ of $\theta_0$ and a set of conservation laws $\mathbf{h_0}$ for $\phi$ and a function $K_{\theta_0}$ such that $M(\theta) = K_{\theta_0}(\phi(\theta))$ for each $\theta \in \mathcal{M}_{\theta_0}^{\mathbf{ho}} \cap \Omega'$, where $\mathcal{M}_{\theta_0}^{\mathbf{ho}} := \{\theta : \mathbf{h_0}(\theta) = \mathbf{h_0}(\theta_0)\}$. As the family of conservation laws $\mathbf{h}$ is complete on $\Omega$ (and in particular on $\Omega'$) and as $\mathrm{Lie}(\mathbb{W}_\phi)(\theta)$ has a constant dimension on $\Omega$ (and thus on $\Omega'$) by (Marcotte et al., 2023, Proposition 4.3), using (Marcotte et al., 2025, Proposition 2.12) yields that $\mathcal{M}_{\theta_0}^{\mathbf{ho}} := \{\theta : \mathbf{h_0}(\theta) = \mathbf{h_0}(\theta_0)\} \supset \mathcal{M}_{\theta_0}^{\mathbf{h}} := \{\theta : \mathbf{h}(\theta) = \mathbf{h}(\theta_0)\}$. Thus the function $K_{\theta_0}$ also satisfies $M(\theta) = K_{\theta_0}(\phi(\theta))$ on $\mathcal{M}_{\theta_0}^{\mathbf{h}}$, hence $\mathbf{h}$ satisfies assumption $i)$ of Theorem 2.14. As the rank of $\partial h(\theta)$ is constant on $\Omega'$, we deduce by Theorem 2.14 the inclusion equation 7, which contradicts the previous step.

**B. Finally we show that such a $\theta_0$ does not satisfy the *strong intrinsic dynamic property either.***

This is a direct consequence of the following lemma.

**Lemma I.3.** *Consider a two-layer linear network parameterized by $\theta = (U, V)$, $\phi(\theta) = \phi_{\mathtt{Lin}}(\theta) = UV^\top$, and an initialization $\theta_0 = (U_{t=0}, V_{t=0})$ such that the vertical concatenation $\begin{pmatrix} U_{t=0} \\ V_{t=0} \end{pmatrix}$ has full rank. If $\theta_0$ satisfies the strong intrinsic dynamic property with respect to $\phi$ on $\Omega$, then it also satisfies the intrinsic metric property with respect to $\phi$ on some open neighborhood $\Omega'$ of $\theta_0$.*

*Proof of Lemma I.3.* Consider the function $K_{\theta_0}$ which existence is guaranteed by the fact that $\theta_0$ satisfies the strong intrinsic dynamic property. We will use the Chow-Rashevskii theorem to show that there is a neighborhood $\Omega' \subseteq \Omega$ of $\theta_0$ such that $\mathcal{M}_{\theta_0} \cap \Omega'$ is *attainable* by patching a finite number of trajectories $\dot{\theta}_k(t) = -\nabla w_k(\theta_k(t))$, each initiated at the ending point of the previous one, defined via fields

$$w_k(\cdot) \in \mathcal{F} := \{w(\cdot) = \partial\phi^\top(\cdot)\nabla f(\phi(\cdot)) : f \in C^\infty\}.$$

We will further show that there is a piecewise $\mathcal{C}^2$ function $f$ such that this is feasible with a continuous trajectory $\theta(t)$ such that $\dot{\theta}(t) = -\nabla \ell(\theta(t))$ with $\ell = f \circ \phi$ for each $t$ such $\theta(t) \in \Omega$ and $f$ is differentiable at $\phi(\theta(t))$. The strong intrinsic dynamic property of $\theta_0$ with respect to $\phi$ thus yields $M(\theta(t)) = K_{\theta_0}(\phi(\theta(t)))$ at every time, and in particular $M(\theta) = K_{\theta_0}(\phi(\theta))$ at the end point of the trajectory. This shows that $\theta_0$ satisfies the intrinsic metric property with respect to $\phi$ on $\Omega'$.

To exploit the Chow-Rashevskii theorem, we first observe that $\mathcal{F} = -\mathcal{F}$, and that by standard Lie algebra calculus, since every $w(\cdot) \in \mathcal{F}$ can be written as

$$w(\theta) = \sum_{i=1}^d a_i(\theta)\nabla\phi_i(\theta)$$

we have $\mathrm{Lie}(\mathcal{F})(\theta) \subseteq \mathrm{Lie}(\mathbb{W}_\phi)(\theta)$ where $\mathbb{W}_\phi := \mathrm{span}\{\nabla\phi_i(\cdot)\}$. Vice-versa, considering $e_i \in \mathbb{R}^d$ the $i$-th canonical vector, since $f_i(\phi) := \langle e_i, \phi \rangle$ is $\mathcal{C}^\infty$ with $\nabla f(\phi) = e_i$, we get $w_i(\cdot) := \partial\phi^\top(\cdot)\nabla f_i(\cdot) = \nabla\phi_i(\cdot) \in \mathcal{F}$, hence $\mathrm{Lie}(\mathbb{W}_\phi)(\theta) \subseteq \mathrm{Lie}(\mathcal{F})(\theta)$. Moreover, as $\begin{pmatrix} U_{t=0} \\ V_{t=0} \end{pmatrix}$ has full rank it remains the case in a neighborhood $\Omega'' \subset \Omega$ of $\theta_0$. With the same arguments as in step 4 of the proof of Theorem 3.9 above, the vector-valued function $\mathbf{h}$ contains a complete set of conservation laws of $\phi = \phi_{\mathtt{Lin}}$, and by (Marcotte et al., 2023, Propositions 4.2 and 4.3) one thus has $\mathrm{Lie}(\mathcal{F})(\theta) = \mathrm{Lie}(\mathbb{W}_\phi)(\theta) = T_\theta(\mathcal{M}_{\theta_0}^{\mathbf{h}} \cap \Omega')$ (with $T_\theta$ the tangent plane at $\theta$) for each $\theta \in \mathcal{M}_{\theta_0}^{\mathbf{h}} \cap \Omega'$. Finally, choose an open neighborhood $\Omega' \subseteq \Omega''$ of $\theta_0$ such that $\mathcal{M}_{\theta_0}^{\mathbf{h}} \cap \Omega'$ is a connected set: all the assumptions of the Chow-Rashevskii theorem (Jurdjevic, 1997, Theorem 3) hold on $\Omega'$, thus the attainable sets of $\mathcal{F}$ from $\theta_0$ on $\Omega'$ is exactly $\mathcal{M}_{\theta_0}^{\mathbf{h}} \cap \Omega'$. This means that for every $\theta \in \mathcal{M}_{\theta_0}^{\mathbf{h}} \cap \Omega'$ there is a trajectory $\theta(t), t \in [0, T]$ such that $\theta(t) \in \Omega'$ at every time, $\theta(0) = \theta_0$, $\theta(T) = \theta$, and $[0, T] = \cup_k[t_k, t_{k+1}]$ with $\dot{\theta}(t) = w_k(\theta(t))$ for $t \in (t_k, t_{k+1})$, with $w_k \in \mathcal{F}$. Without loss of generality the trajectory does not self-intersect (if it does at times $\tau < \tau'$, we can shorten it by concatenating the trajectories on $[0, \tau]$ and $[\tau', T]$). Since there are $\mathcal{C}^\infty$ functions $f_k$ such that $w_k = \nabla(f_k \circ \phi)$, and since the trajectory $\theta(t)$ does not self-intersect, there is a *piecewise $\mathcal{C}^\infty$* function

$f$ that matches each $f_k$ on each piece of the trajectory $\theta(\cdot)$, hence $\dot{\theta}(t) = -\nabla \ell(\theta(t))$ with $\ell = f \circ \phi$, for every $t \in (t_k, t_{k+1})$ and every $k$.

$\square$

$\square$

## J    PROOF OF PROPOSITION 3.10.

**Proposition 3.10.** *Let $\theta := (u, v)$ with $u \in \mathbb{R}^r$ and $v \in \mathbb{R}^r$. Then $z := \phi_{\mathtt{Lin}}(\theta) = \langle u, v \rangle \in \mathbb{R}$. We denote $S := u_{t=0} u_{t=0}^\top - v_{t=0} v_{t=0}^\top \in \mathbb{R}^{r \times r}$. Then one has $\dot{z} = -\sqrt{2\mathrm{tr}(S^2) - \mathrm{tr}(S)^2 + 4z^2} \nabla f(z)$.*

*Proof.* Since $\partial \phi(\theta) = [v^\top, u^\top]$ we have

$$\partial \phi(\theta) \partial \phi(\theta)^\top = \|u\|^2 + \|v\|^2.$$

Since $\mathbf{h}(\theta) := uu^\top - vv^\top$ is a conservation law of $\phi_{\mathtt{Lin}}$ for every $\theta = (u, v)$ on the trajectory one has: $S = uu^\top - vv^\top$, and therefore $S^2 = \|u\|^2 uu^\top - zuv^\top - zvu^\top + \|v\|^2 vv^\top$. Thus

$$\mathrm{tr}(S^2) = \|u\|^4 + \|v\|^4 - 2z^2.$$

As one also has: $(\|u\|^2 - \|v\|^2)^2 = \mathrm{tr}(S)^2$, one has:

$$\begin{aligned}
(\partial \phi(\theta) \partial \phi(\theta)^\top)^2 = (\|u\|^2 + \|v\|^2)^2 &= 2(\|u\|^4 + \|v\|^4) - (\|u\|^2 - \|v\|^2)^2 \\
&= 2(\mathrm{tr}(S^2) + 2z^2) - \mathrm{tr}(S)^2 \\
&= 2\mathrm{tr}(S^2) + 4z^2 - \mathrm{tr}(S)^2,
\end{aligned}$$

which concludes the proof. $\square$

## K    PROOF OF THEOREM 3.11.

**Theorem 3.11.** *If $\theta_L(0)$ satisfies the relaxed balanced condition (Definition 3.6) with $\boldsymbol{\lambda} = (\lambda_i)_i$ then during the trajectory $\theta_L(t)$ of equation 1, the matrices in equation 11 satisfy $S_j(\theta_L(t)) = Q_j(U_L(t)U_L(t)^\top)$ and $T_j(\theta_L(t)) = R_j(U_1(t)^\top U_1(t))$, where $Q_j(x) := \prod_{k=0}^{L-j-1}(x - a_k)$ with $a_0 := 0$ and $a_k := \sum_{i=1}^k \lambda_{L-i}$ for $k = 1, \cdots L-1$ and $R_j(x) := \prod_{k=0}^{j-2}(x - b_k)$ with $b_0 := 0$ and $b_k := -\sum_{i=1}^k \lambda_i$. Moreover $U_L U_L^\top$ (resp. $U_1^\top U_1$) is the unique root of $Z_L Z_L^\top = Q_0(U_L U_L^\top)$ (resp. of $Z_L^\top Z_L = R_{L-1}(U_1^\top U_1)$) with spectrum lower bounded by $\max_{0 \leq k \leq L-1} a_k$ (resp. by $\max_{0 \leq k \leq L-2} b_k$). This implies that all matrices in equation 11 are entirely characterized by $Z_L$ and the initialization, hence $\theta_L(0)$ satisfies the intrinsic dynamic property on $\mathbb{R}^D$ with respect to $\phi_{\mathtt{Lin}}$.*

*Proof.* Let us first outline the main steps of the proof. We first show that the equalities $Z_L Z_L^\top = Q_0(U_L U_L^\top)$ and $Z_L^\top Z_L = R_{L-1}(U_1^\top U_1)$ hold on the whole trajectory. Then we prove that this implies the expression of $S_j$ (resp. of $T_j$) in terms of $U_L U_L^\top$ (resp. of $U_1^\top U_1$) along the whole trajectory too. Finally we show that along the whole trajectory $U_L U_L^\top$ and $U_1^\top U_1$ (and therefore all $S_j$'s and $T_j$'s) are entirely characterized by $Z_L = \phi_{\mathtt{Lin}}(\theta_L)$ and the initial conditions (captured by $\boldsymbol{\lambda}$). This will thus imply that $\theta_L(0)$ satisfies the intrinsic dynamic property on $\mathbb{R}^D$ with respect to $\phi_{\mathtt{Lin}}$.

**Step 1: Expression of $Z_L Z_L^\top$ as a polynomial in $U_L U_L^\top$**

Since $U_{j+1}^\top U_{j+1} - U_j U_j^\top$ is a set of conservation laws for $\phi_{\mathtt{Lin}}$, the fact that the relaxed balanced conditions equation 12 hold at initialization implies that they hold along the whole trajectory.

We prove by induction on $1 \leq \ell \leq L$ that $Z_\ell := U_\ell \ldots U_1$ satisfies $Z_\ell Z_\ell^\top = P_\ell(U_\ell U_\ell^\top)$ for some polynomial $P_\ell$ of degree $\ell$ that satisfy $P_1(x) = x$ and $P_\ell(x) = xP_{\ell-1}(x - \lambda_{\ell-1})$ for $2 \leq \ell \leq L$. For $\ell = 1$ we trivially have $Z_\ell = U_\ell$ hence the result is true. Now consider $2 \leq \ell \leq L$ and assume that the result holds true for $\ell - 1$. Since $Z_\ell = U_\ell Z_{\ell-1}$ we have

$$Z_\ell Z_\ell^\top = U_\ell(Z_{\ell-1} Z_{\ell-1}^\top)U_\ell^\top = U_\ell P_{\ell-1}(U_{\ell-1} U_{\ell-1}^\top)U_\ell^\top \overset{equation\ 12}{=} U_\ell P_{\ell-1}(U_\ell^\top U_\ell - \lambda_{\ell-1}\mathtt{id})U_\ell^\top$$

where we used equation 12 for $i = \ell - 1$. Denoting $\hat{P}_{\ell-1}(x) := P_{\ell-1}(x - \lambda_{\ell-1})$ we obtain $Z_\ell Z_\ell^\top = U_\ell \hat{P}_{\ell-1}(U_\ell^\top U_\ell)U_\ell^\top = U_\ell U_\ell^\top \hat{P}_{\ell-1}(U_\ell U_\ell^\top) = P_\ell(U_\ell U_\ell^\top)$. This concludes the induction.

Given the recursion formula for $P_\ell$, another easy induction yields

$$P_\ell(x) = \Pi_{k=0}^{\ell-1}(x - \sum_{i=1}^{k}\lambda_{\ell-i}), \quad 1 \le \ell \le L. \tag{30}$$

Specializing to $\ell = L$ we obtain $P_L = Q_0$ as claimed.

**Step 2: Expression of $S_j$ (resp. of $Z_L^\top Z_L$ and $T_j$) as a polynomial in $U_L U_L^\top$ (resp. in $U_1^\top U_1$).**

It is a direct consequence of the first step, as we now explain. To show the result on $S_j$, consider the new variable $\theta' = (U'_{L-j}, \ldots, U'_1) := (U_L, \ldots, U_{j+1})$ and $Z' := U'_{L-j} \cdots U'_1 = U_L \cdots U_{j+1}$. With these notations we have $S_j = Z'Z'^\top$, and the relaxed balanced conditions imply that:

$$(U'_{i+1})^\top U'_{i+1} - U'_i(U'_i)^\top = \lambda'_i \mathrm{Id}_n, \quad 1 \le i \le L - j - 1$$

where $\boldsymbol{\lambda}' = (\lambda'_{L-j-1}, \ldots, \lambda'_1) := (\lambda_{L-1}, \ldots, \lambda_{j+1})$. By the first step we obtain the desired expression.

Similar computations with $\theta' = (U_1^\top, \ldots, U_{j-1}^\top)$, $Z' = U_1^\top \ldots U_{j-1}^\top$ and $\boldsymbol{\lambda}' = (-\lambda_1, \ldots, -\lambda_{j-2})$ show the desired expression for $T_j = Z'Z'^\top$ and $Z_L^\top Z_L$ as well.

**Step 3: Characterization of $U_L U_L^\top$ via $Z_L$ and the initial conditions.** The proof that $U_1^\top U_1$ is characterized by $Z_L$ (in fact $Z_L^\top Z_L$) and the initial conditions is similar and therefore omitted.

By the first step we have $Z_L Z_L^\top = Q_0(U_L U_L^\top)$, hence $U_L U_L^\top$ is indeed a matrix root of this equation. As both matrices $Z_L Z_L^\top$ and $U_L U_L^\top$ are real symmetric, the above expression shows that we can reduce to the scalar study of their eigenvalues.

As we show below, a consequence of the relaxed balancedness conditions equation 12 is that all eigenvalues of the positive semi-definite matrix $U_L U_L^\top$ belong to the interval $I := [\max(0, a_1, \ldots, a_{L-1}), \infty)$. Thus, considering any eigenvalue $e \ge 0$ of the positive semi-definite matrix $Z_L Z_L^\top$, it is enough to show that the polynomial equation $R(X) := Q_0(X) - e = 0$ admits a unique root in this interval.

The existence of a root in $I$ is a consequence of the mean value theorem, since $R(\max(0, a_1, \cdots, a_{L-1})) = -e \le 0$ and $\lim_{x \to \infty} R(x) = +\infty$. To prove uniqueness, we proceed by contradiction: assume that $R(X)$ admits two distinct roots $x_1 < x_2$ in $I$. By Rolle's theorem $R'(X) = Q'_0(X)$ has a root in $]x_1, x_2[$. This contradicts the fact that, by the construction of $Q_0$ and Rolle's theorem, all roots of $Q'_0(X)$ are contained in the open interval $(\min(0, a_1, \ldots, a_{L-1}), \max(0, a_1, \ldots, a_{L-1}))$.

To conclude the proof, we show that indeed all eigenvalues of $U_L U_L^\top$ belong to $I := [\max(0, a_1, \ldots, a_{L-1}), \infty)$. Denote $\sigma_i = \inf \mathrm{sp}(U_i U_i^\top)$, $1 \le i \le L$. Since each matrix $U_\ell$ is square and positive semi-definite, we have $\mathrm{sp}(U_i U_i^\top) = \mathrm{sp}(U_i^\top U_i) \subseteq [0, \infty)$ for every $1 \le i \le L$, and by equation 12 we also have $\mathrm{sp}(U_{i+1}U_{i+1}^\top) = \lambda_i + \mathrm{sp}(U_i U_i^\top)$, hence $\sigma_{i+1} = \sigma_i + \lambda_i \ge 0$ for $1 \le i \le L - 1$. An easy recursion shows that $\sigma_i \ge \max(0, \sum_{j=1}^{i-1}\lambda_j)$ for $1 \le i \le L$, hence the result. $\square$

We now anticipate a slight generalization part of the results of Theorem 3.11 that will be used later in the proof of Theorem 3.13.

**Lemma K.1** (Perturbed relaxed balanced condition). *Consider matrices $(U_k)_{k=0}^{L-1} \subset \mathbb{R}^{n \times n}$ and scalars $(\lambda_k)_{k=0}^{L-1}$. Denoting $h := 1/L$, define*

$$C_U := \max(1, \max_k \|U_k\|), \quad C_\lambda := \max_k |\lambda_k| \tag{31}$$

$$\eta := L^2 \cdot \max_{0 \le k \le L-2} \|(U_{k+1}^\top U_{k+1} - U_k U_k^\top) - h^2 \lambda_k \, \mathrm{Id}_n\| \tag{32}$$

*Fix $j \in \{0, \ldots, L-2\}$ and recall that $S_j := (U_{L-1} \cdots U_{j+1})(U_{L-1} \cdots U_{j+1})^\top$. Define $a_0 := 0$ and, for $k \geq 1$, $a_k := h^2 \sum_{i=1}^{k} \lambda_{L-1-i}$, $C_0 := 2C_\lambda$, $C_1 := (C_U^2 + \eta h^2 - 1)/h$. Then*

$$\max_j \left\| S_j - \prod_{k=0}^{L-1-(j+1)} \left( U_{L-1} U_{L-1}^\top - a_k \mathrm{Id}_n \right) \right\| \leq \left( C_0 e^{C_1} e^{C_0(1+C_1)} + e^{C_1} \right) \eta. \tag{33}$$

Before proving this lemma, we state the following lemma, as it will be essential in the proof of Lemma K.1: it provides a uniform bound on the Lipschitz constant of a class of polynomials.

**Lemma K.2** (Uniform Lipschitz bound). *Consider $C_0 > 0$, $C_1 > 0$. For any $0 < h \leq 1$, any integer $1 \leq d \leq 1/h$, any degree–$d$ polynomial*

$$Q_d(x) = \prod_{k=1}^{d}(x - c_k),$$

*with $\max_k |c_k| \leq C_0 h$, and any matrices $A$, $A + \Delta \in B_{R(h)} := \{X : \|X\| \leq R(h)\}$ where $R(h) := 1 + C_1 h$ and where $\|\cdot\|$ denotes the Frobenius norm, one has*

$$\|Q_d(A + \Delta) - Q_d(A)\| \leq \frac{K}{h} \|\Delta\|, \text{ with } K = K(C_0, C_1) = C_0 e^{C_0(1+C_1)} + e^{C_1}. \tag{34}$$

*Proof.* **Step 1: Scalar Lipschitz constant on the ball $B_R$.** For any matrix polynomial $Q(x) = \sum_{m=0}^{d} \alpha_m x^m$ one has, denoting $DQ(X)[H] = \sum_{m=1}^{d} \alpha_m \sum_{j=0}^{m-1} X^j H X^{m-1-j}$:

$$Q(A + \Delta) - Q(A) = \int_0^1 DQ(A + t\Delta)[\Delta]\, dt,$$

$$\|DQ(X)[H]\| \leq L_Q(\|X\|_{2\to2}) \|H\| \leq L_Q(\|X\|) \|H\|, \quad \forall X, \forall H$$

where $L_Q(R) := \sum_{m=1}^{d} |\alpha_m|\, m\, R^{m-1}$ (we used here that the spectral norm is bounded by the Frobenius norm).

**Step 2: Bounding $L_{Q_d}(R(h))$.** Exploiting the coefficient–root relation on $Q_d$ that is unitary yields $|\alpha_m| \leq \binom{d}{m} \beta^m$ where $\beta := C_0 h$ for any $0 \leq m \leq d-1$. Since $\alpha_d = 1$, for any $R > 0$ we obtain

$$L_{Q_d}(R) \leq \beta \sum_{m=1}^{d} \binom{d}{m} m (\beta R)^{m-1} + d R^{d-1} = d\beta(1 + \beta R)^{d-1} + d R^{d-1}.$$

Insert $d - 1 \leq d \leq 1/h$, $\beta = C_0 h$. Since $R(h) = 1 + C_1 h \leq R(1) = 1 + C_1$ (as $h \leq 1$) we get:

$$L_{Q_d}(R(h)) \leq \frac{1}{h} C_0 h (1 + C_0 h R(h))^{1/h} + d(1 + C_1 h)^{1/h} \leq C_0 e^{C_0 R(h)} + \frac{e^{C_1}}{h} \leq \frac{C_0 e^{C_0(1+C_1)} + e^{C_1}}{h}.$$

where the exponential bound uses $(1 + t)^{1/t} \leq e$ for $t > 0$. We define $K = K(C_0, C_1) := C_0 e^{C_0(1+C_1)} + e^{C_1}$.

**Step 3: Conclusion.** Applying the integral formula for Step 1 with the bound from Step 2 gives

$$\|Q_d(A + \Delta) - Q_d(A)\| \leq (K/h) \|\Delta\|,$$

for every $A, \Delta$ with $A, A + \Delta \in B_{R(h)}$, which is equation 34. $\qquad \square$

**We now prove Lemma K.1.**

*Proof.* **Step 0: Reindexing.** Work with the truncated sequence $(U_1', \ldots, U_N') := (U_{j+1}, \ldots, U_{L-2}, U_{L-1})$, where $N := L - 1 - j$. Define $Z_\ell := U_\ell' \cdots U_1'$ for $1 \leq \ell \leq N$ and $M_\ell := U_\ell' U_\ell'^\top$. Then $S_j = Z_N Z_N^\top$.

We also observe that by the definition of $\eta$ in equation 32, since $h = 1/L$, we have for each $1 \leq \ell \leq N$

$$U_\ell'^\top U_\ell' - M_{\ell-1} = U_\ell'^\top U_\ell' - U_{\ell-1}' U_{\ell-1}'^\top = h^2 \lambda_{\ell+j-1} \mathrm{Id}_n + r_{\ell-1}', \qquad \|r_{\ell-1}'\| \leq h^2 \eta. \tag{35}$$

**Step 1: Polynomial representation with a perturbation.** We prove by induction on $\ell$ that

$$E_\ell := Z_\ell Z_\ell^\top - P_\ell(M_\ell) \quad \text{satisfies } \|E_\ell\| \le \ell K C_U^{2\ell} \cdot h\eta \tag{36}$$

where the polynomials $P_\ell$ are defined by

$$P_1(x) := x, \qquad P_\ell(x) := x\, P_{\ell-1}\big(x - b_{\ell-1}\big) \quad (2 \le \ell \le N),$$

with $b_{\ell-1} := h^2 \lambda_{\ell+j-1}$ (matching the re-indexed sequence), and the constant $K$ is obtained by Lemma K.2 applied to the constants $C_0 := 2C_\lambda$ and $C_1 := (C_U^2 + \eta h^2 - 1)/h$.

*Base case $\ell = 1$.* Trivial: $Z_1 = U_1'$, so $Z_1 Z_1^\top = M_1 = P_1(M_1)$ and $E_1 = 0$.

*Induction step.* Assume equation 36 holds at rank $\ell - 1$. Since $Z_\ell = U_\ell' Z_{\ell-1}$,

$$Z_\ell Z_\ell^\top = U_\ell'\big(Z_{\ell-1} Z_{\ell-1}^\top\big) U_\ell'^\top = U_\ell' P_{\ell-1}(M_{\ell-1}) U_\ell'^\top + U_\ell' E_{\ell-1} U_\ell'^\top.$$

By induction hypothesis and the fact that the spectral norm is bounded by the Frobenius norm, the second term of the right hand side is bounded as

$$\|U_\ell' E_{\ell-1} U_\ell'^\top\| \le C_U^2 \|E_{\ell-1}\| \le C_U^2 (\ell-1) K C_U^{2(\ell-1)} h\eta \le (\ell-1) C_U^{2\ell} K h\eta,$$

hence we only need to show that

$$\|U_\ell' P_{\ell-1}(M_{\ell-1}) U_\ell'^\top - P_\ell(M_\ell)\| \le C_U^{2\ell} K \cdot h\eta.$$

Write $Q_{\ell-1}(x) := P_{\ell-1}(x - b_{\ell-1})$. From equation 35 and the definition of $M_{\ell-1} = U_{\ell-1}'[U_{\ell-1}']^\top$ one gets

$$M_{\ell-1} = U_\ell'^\top U_\ell' - b_{\ell-1} \mathrm{Id}_n - r_{\ell-1}', \qquad \|r_{\ell-1}'\| \le h^2 \eta.$$

Hence

$$
\begin{aligned}
U_\ell' P_{\ell-1}(M_{\ell-1}) U_\ell'^\top &= U_\ell' Q_{\ell-1}\big(U_\ell'^\top U_\ell' - r_{\ell-1}'\big) U_\ell'^\top \\
&= \underbrace{U_\ell' Q_{\ell-1}(U_\ell'^\top U_\ell') U_\ell'^\top}_{=M_\ell\, Q_{\ell-1}(M_\ell) = P_\ell(M_\ell)} + U_\ell'\Big(Q_{\ell-1}(U_\ell'^\top U_\ell' - r_{\ell-1}') - Q_{\ell-1}(U_\ell'^\top U_\ell')\Big) U_\ell'^\top.
\end{aligned}
$$

Thus to conclude the induction step we only need to show that

$$\left\|U_\ell'\Big(Q_{\ell-1}(U_\ell'^\top U_\ell' - r_{\ell-1}') - Q_{\ell-1}(U_\ell'^\top U_\ell')\Big) U_\ell'^\top\right\| \le C_U^{2\ell} K \cdot h\eta.$$

By the definition of $C_1$, the matrices $A = U_\ell'^\top U'$, $\Delta = -r_{\ell-1}'$, satisfy $\max(\|A\|, \|\Delta\|) \le \|A\| + \|\Delta\| \le C_U^2 + h^2\eta \le 1 + C_1 h$. Moreover, with the same induction that has led to equation 30, the polynomial $P_{\ell-1}(x)$ has all its roots bounded by $Lh^2 C_\lambda$, hence $Q_{\ell-1}(x) := P_{\ell-1}(x - b_{\ell-1})$ has all its roots bounded by $(L+1)h^2 C_\lambda \le 2C_\lambda h = C_0 h$, therefore we can apply Lemma K.2 to obtain, with $K = K(C_0, C_1) = C_0 e^{C_0(1+C_1)} + e^{C_1}$:

$$\|Q_{\ell-1}(U_\ell'^\top U_\ell' - r_{\ell-1}') - Q_{\ell-1}(U_\ell'^\top U_\ell')\| \le \frac{K}{h} \|r_{\ell-1}'\| \le K \cdot h\eta$$

$$\left\|U_\ell'\Big(Q_{\ell-1}(U_\ell'^\top U_\ell' - r_{\ell-1}') - Q_{\ell-1}(U_\ell'^\top U_\ell')\Big) U_\ell'^\top\right\| \le C_U^2 K \cdot h\eta \overset{C_U \ge 1}{\le} C_U^{2\ell} K \cdot h\eta,$$

which concludes the induction.

**Step 2: Factorisation of $P_N$.** With the same induction that has led to equation 30, we have

$$P_N(x) = \prod_{k=0}^{N-1} (x - a_k), \qquad a_0 = 0, \quad a_k = \sum_{i=1}^{k} h^2 \lambda_{L-1-i}.$$

Applying equation 36 with $\ell = N$ and recalling $S_j = Z_N Z_N^\top$ yields

$$S_j = P_N(M_N) + E_N = \prod_{k=0}^{N-1} (M_N - a_k I_n) + E_N,$$

where $\|E_N\| \le C_U^{2N} N K h\eta \le (1 + C_1 h)^L K\eta \le \exp(C_1) K\eta$.

Since $M_N = U_N' U_N'^\top = U_{L-1} U_{L-1}^\top$, we recover equation 33 as claimed. $\qquad\square$

## L  PROOF OF PROPOSITION 3.12.

**Proposition 3.12.** *For any $s \in [0,1]$, consider $\mathbf{h}_s : \theta := (\mathcal{A}_s)_{s \in [0,1]} \in \mathcal{X} \mapsto \mathcal{A}'_s + \mathcal{A}'^{\top}_s + [\mathcal{A}^{\top}_s, \mathcal{A}_s] \in \mathbb{R}^{n \times n}$, where we denote $\mathcal{A}'_s := \frac{\mathrm{d}}{\mathrm{d}s}\mathcal{A}_s$. Then for any $s \in [0,1]$, one has for any $t$: $\mathbf{h}_s(\theta(t)) = \mathbf{h}_s(\theta(0))$, where $\theta(t)$ is the maximal solution of equation 15 with initialization $\theta(0)$.*

*Proof.* For convenience we recall the state equation equation 16 for $Z_s$, where $s \in [0,1]$ indicates depth:

$$\frac{\mathrm{d}Z_s}{\mathrm{d}s} = \mathcal{A}_s\, Z_s, \quad Z_0 = \mathrm{Id}_n \text{ fixed,} \tag{37}$$

and we recall that the objective function is factorized by $\ell(\theta) = f(Z_{s=1})$, where the parameters are the family $\theta = \{\mathcal{A}_s : s \in [0,1]\}$.

Let $\theta : [t \in [0,T] \mapsto \theta(t) \in \mathcal{X}] \in \mathcal{C}^1([0,T], \mathcal{X})$ be the solution of the gradient flow given by the family of coupled ODE equation 15

$$\forall s \in [0,1], \quad \frac{\partial \mathcal{A}_s}{\partial t}(t) = -\mathfrak{g}_s(t), \quad \text{with} \quad \mathfrak{g}_s(t) := \frac{\partial \ell}{\partial \mathcal{A}_s}\big(\theta(t)\big), \tag{38}$$

with a given initialization $\theta(0)$. Our goal is to show that $\frac{\partial}{\partial t}h_s(\theta(t)) = 0$.

*Step 1: Computations of $\frac{\partial}{\partial t}h_s(\theta(t))$.*

For any $s \in [0,1]$, one has by definition

$$h_s(\theta(t)) = \frac{\partial \mathcal{A}_s(t)}{\partial s} + \left(\frac{\partial \mathcal{A}_s(t)}{\partial s}\right)^{\top} + [\mathcal{A}_s(t)^{\top}, \mathcal{A}_s(t)].$$

Taking the $t$-derivative yields

$$\begin{aligned}
\frac{\partial}{\partial t}h_s(\theta(t)) &= \frac{\partial}{\partial t}\left(\frac{\partial \mathcal{A}_s(t)}{\partial s}\right) + \frac{\partial}{\partial t}\left(\frac{\partial \mathcal{A}_s(t)}{\partial s}\right)^{\top} + \frac{\partial}{\partial t}[\mathcal{A}_s(t)^{\top}, \mathcal{A}_s(t)] \\
&= \frac{\partial}{\partial s}\left(\frac{\partial \mathcal{A}_s(t)}{\partial t}\right) + \left(\frac{\partial}{\partial s}\frac{\partial \mathcal{A}_s(t)}{\partial t}\right)^{\top} + \frac{\partial}{\partial t}[\mathcal{A}_s(t)^{\top}, \mathcal{A}_s(t)],
\end{aligned} \tag{39}$$

where the exchange of derivatives is justified in Section L.1.

Moreover one has

$$\frac{\partial}{\partial t}[\mathcal{A}_s(t)^{\top}, \mathcal{A}_s(t)] = \left[\frac{\partial \mathcal{A}_s(t)^{\top}}{\partial t}, \mathcal{A}_s(t)\right] + \left[\mathcal{A}_s(t)^{\top}, \frac{\partial \mathcal{A}_s(t)}{\partial t}\right].$$

Thus by using equation 38

$$\frac{\partial \mathcal{A}_s(t)}{\partial t} = -\mathfrak{g}_s(t),$$

we obtain

$$\begin{aligned}
\frac{\partial}{\partial t}h_s(\theta(t)) &= \frac{\partial}{\partial s}\big(-\mathfrak{g}_s(t)\big) + \left(\frac{\partial}{\partial s}\big(-\mathfrak{g}_s(t)\big)\right)^{\top} + \big[-\mathfrak{g}_s(t)^{\top}, \mathcal{A}_s(t)\big] + \big[\mathcal{A}_s(t)^{\top}, -\mathfrak{g}_s(t)\big] \\
&= -\frac{\partial \mathfrak{g}_s(t)}{\partial s} - \left(\frac{\partial \mathfrak{g}_s(t)}{\partial s}\right)^{\top} - [\mathfrak{g}_s(t)^{\top}, \mathcal{A}_s(t)] - [\mathcal{A}_s(t)^{\top}, \mathfrak{g}_s(t)].
\end{aligned} \tag{40}$$

The remaining task is to show that the sum of these terms cancels, using an expression of the gradient.

*Step 2: An expression of $\mathfrak{g}_s(t)$ using the adjoint equation.*

To compute the gradient $\mathfrak{g}_s = \frac{\partial \ell}{\partial \mathcal{A}_s}$, we introduce the adjoint variable (Pontryagin et al., 1962) $\Lambda_s(t)$, which satisfies the adjoint equation

$$\frac{\partial \Lambda_s(t)}{\partial s} = -\mathcal{A}_s(t)^{\top}\Lambda_s(t), \quad \Lambda_1(t) = \frac{\partial f}{\partial Z}\big(Z_1(t)\big). \tag{41}$$

Moreover it satisfies as shown in Section L.2:

$$\mathfrak{g}_s(t) = \Lambda_s(t)\, Z_s(t)^\top. \tag{42}$$

*Step 3: Compute $\frac{\partial}{\partial s}\mathfrak{g}_s(t)$.*

By differentiating equation 42 with respect to $s$, we get:

$$\frac{\partial}{\partial s}\mathfrak{g}_s(t) = \frac{\partial \Lambda_s(t)}{\partial s}\, Z_s(t)^\top + \Lambda_s(t)\, \frac{\partial Z_s(t)^\top}{\partial s}.$$

Then by using the adjoint equation equation 41 and the state equation equation 37, one has

$$\frac{\partial}{\partial s}\mathfrak{g}_s(t) = -\mathcal{A}_s(t)^\top \Lambda_s(t) Z_s(t)^\top + \Lambda_s(t) Z_s(t)^\top \mathcal{A}_s(t)^\top \tag{43}$$

$$= -\mathcal{A}_s(t)^\top \mathfrak{g}_s(t) + \mathfrak{g}_s(t)\, \mathcal{A}_s(t)^\top \tag{44}$$

$$= -[\mathcal{A}_s(t)^\top, \mathfrak{g}_s(t)]. \tag{45}$$

Taking the transpose,

$$\left(\frac{\partial}{\partial s}\mathfrak{g}_s(t)\right)^\top = -\mathfrak{g}_s(t)^\top \mathcal{A}_s(t) + \mathcal{A}_s(t)\, \mathfrak{g}_s(t)^\top = [\mathcal{A}_s(t), \mathfrak{g}_s(t)^\top] = -[\mathfrak{g}_s(t)^\top, \mathcal{A}_s(t)]. \tag{46}$$

*Step 4: Conclusion.* By substituting the computed expressions into equation 40, one obtains as claimed that

$$\frac{\partial}{\partial t} h_s(\theta(t)) = 0.$$

$\square$

## L.1    WE NOW DETAIL EQUATION 39.

**Theorem L.1** (Commutation of mixed derivatives). *Let*

$$\mathcal{X} = \mathcal{C}^1\big([0,1], \mathbb{R}^{n\times n}\big), \qquad \|f\|_{\mathcal{X}} := \max\{\|f\|_\infty, \|f'\|_\infty\},$$

*and set $B = \mathcal{C}^0([0,1], \mathbb{R}^{n\times n})$ with the sup–norm $\|\cdot\|_B = \|\cdot\|_\infty$. Denote $D : \mathcal{X} \longrightarrow B,\ f \mapsto f'$ the spatial derivative. Suppose $\theta(\cdot) \in \mathcal{C}^1\big([0,T], \mathcal{X}\big)$ and write $\mathcal{A}(t,s) := [\theta(t)](s)$. Then*

- *the mixed derivatives*
$$\partial_t \partial_s \mathcal{A}(t,s) \quad and \quad \partial_s \partial_t \mathcal{A}(t,s)$$
  *exist for every $(t,s) \in [0,T] \times [0,1]$ and coincide:*

$$\boxed{\partial_t \partial_s \mathcal{A}(t,s) = \partial_s \partial_t \mathcal{A}(t,s) \quad \forall\, (t,s)}$$

- *the map $s \mapsto \partial_t \partial_s \mathcal{A}(t,s)$ is continuous.*

*Proof. Step 1: $D$ is continuous.* For every $f \in \mathcal{X}$,

$$\|Df\|_B = \|f'\|_\infty \leq \max\{\|f\|_\infty, \|f'\|_\infty\} = \|f\|_{\mathcal{X}},$$

so $\|D\|_{\mathrm{op}} \leq 1$; hence $D$ is a bounded and thus a continuous linear map.

*Step 2: Temporal differentiability is preserved by $D$.* The fact that the function $\theta$ (valued in the Banach space $\mathcal{X}$) is $\mathcal{C}^1$ means precisely that its (Fréchet) derivative $\dot\theta(t) := \partial_t \theta(t) \in \mathcal{X}$ exists for each $t$ and the map $t \mapsto \dot\theta(t)$ is continuous from $[0,T]$ to $\mathcal{X}$.

Applying the *continuous and linear* operator $D$ yields by linearity

$$\frac{D(\theta(t+h)) - D(\theta(t))}{h} = D\left(\frac{\theta(t+h) - \theta(t)}{h}\right)$$

for every $t \in [0, T]$ and $h$ small enough such that $t + h \in [0, T]$, and since by continuity of $D$ the right hand side tends to $D(\dot{\theta}(t))$ when $h \to 0$, the left hand side also has a limit, showing that

$$\frac{d}{dt}\big(D(\theta(t))\big) = D\big(\dot{\theta}(t)\big) \qquad \text{for every } t \in [0, T]. \tag{47}$$

Thus the mixed derivative $\partial_t \partial_s \mathcal{A}(t, \cdot)$ exists as an element of $B$.

*Step 3: symmetry of the mixed derivatives.* Evaluating the identity equation 47 above pointwise in $s$ and writing $\mathcal{A}(t, s) = [\theta(t)](s)$ gives

$$\partial_t \partial_s \mathcal{A}(t, s) = \big[D(\dot{\theta}(t))\big](s) = \partial_s \big[\dot{\theta}(t)\big](s) = \partial_s \partial_t \mathcal{A}(t, s).$$

Hence the two mixed derivatives exist everywhere and are equal.

*Step 4: continuity of $s \mapsto \partial_t \partial_s \mathcal{A}(t, s)$.* Since $\dot{\theta}(t) \in \mathcal{X}$ for each $t$, its derivative $s \mapsto \partial_s \big[\dot{\theta}(t)\big](s)$ is continuous. As $\partial_s \big[\dot{\theta}(t)\big](s) = \partial_s \partial_t \mathcal{A}(t, s)$, by the previous step, this exactly means that $s \mapsto \partial_t \partial_s \mathcal{A}(t, s)$ is continuous. $\qquad \square$

### L.2 WE NOW SHOW EQUATION 42.

More precisely, to show equation 42, we will both prove that

$$\mathfrak{g}_s(t) = (Z_s(t)^{-1})^\top Z_1(t)^\top \nabla f(Z_1(t)) Z_s(t)^\top \tag{48}$$

and that

$$\Lambda_s(t) = (Z_s(t)^{-1})^\top Z_1(t)^\top \nabla f(Z_1(t)), \tag{49}$$

which will indeed give equation 42.

We briefly explain why for a given $t$ the matrix $Z_s(t)$ never loses its invertibility when $s \in [0, 1]$ varies, by showing that the determinant can never reach 0. As

$$\partial_s Z_s(t) = \mathcal{A}_s(t) Z_s(t), \qquad Z_0(t) = \mathrm{Id}_n.$$

Jacobi's rule gives

$$\frac{\mathrm{d}}{\mathrm{d}s} \det Z_s(t) = \mathrm{tr}\big(\mathcal{A}_s(t)\big) \det Z_s(t), \quad \det Z_0(t) = 1.$$

Solving this scalar ODE,

$$\det Z_s(t) = \exp\!\Big(\int_0^s \mathrm{tr}\big(\mathcal{A}_\tau(t)\big)\, d\tau\Big) \neq 0, \qquad s \in [0, 1].$$

Therefore $Z_s(t) \in \mathrm{GL}(n)$ for every $s$.

Since $t$ is fixed, in the following we lighten notations by dropping it from the equations. The proof of equation 49 is direct by showing that $\Lambda_s$ and $(Z_s^{-1})^\top Z_1^\top \nabla f(Z_1)$ satisfy the same ODE equation 41 with the same value at $s = 1$. Thus we only need to show equation 48.

*Proof.* To show equation 48 we will use Riesz theorem to identify the expression of the gradient. We thus will consider the Hilbert space

$$L^2 := L^2\big([0, 1], \mathbb{R}^{n \times n}\big), \qquad \langle U, V \rangle_{L^2} := \int_0^1 \mathrm{tr}\big(U_s^\top V_s\big)\, ds,$$

in which the parameter $\theta = \{\mathcal{A}_s \in \mathbb{R}^{n \times n} : s \in [0, 1]\} \in \mathcal{C}^1([0, 1], \mathbb{R}^{n \times n}) =: \mathcal{X} \subseteq L^2$ lives.

We recall that $Z_s(\theta)$ is the unique solution of the state equation equation 16:

$$\partial_s Z_s = \mathcal{A}_s Z_s, \quad Z_0 = \mathrm{Id}_n, \qquad \forall s \in [0, 1], \tag{50}$$

and that the cost $\ell$ is factorized by the flow map $Z_1(\theta)$ with a smooth scalar field $f : \mathbb{R}^{n \times n} \to \mathbb{R}$, *i.e*, $\ell(\theta) := f\big(Z_1(\theta)\big)$.

*1st step: expression of the Gateaux variation of the flow*

Let $\theta = \{\mathcal{A}_s : s \in [0,1]\} \in \mathcal{X}$ be fixed and pick an arbitrary $\delta\theta \in \mathcal{X}$. For $\varepsilon \in \mathbb{R}$ define the perturbed coefficient $\theta^\varepsilon := \theta + \varepsilon\,\delta\theta$, denoting its components $\theta^\varepsilon = \{\mathcal{A}_s^\varepsilon : s \in [0,1]\}$. Denote by $Z_s^\varepsilon := Z_s(\theta^\varepsilon)$ the flow that satisfies the associated ODE:

$$\partial_s Z_s^\varepsilon = \mathcal{A}_s^\epsilon Z_s^\epsilon,\ Z_0^\epsilon = \mathrm{Id}_n, \qquad \forall s \in [0,1]. \tag{51}$$

As $(s,\epsilon,Z) \mapsto \mathcal{A}_s^\epsilon Z \in \mathcal{C}^1$, th function $(s,\epsilon) \mapsto Z_s^\epsilon$ is $\mathcal{C}^1$ using the Cauchy–Lipschitz theorem with a parameter. In particular for any $s \in [0,1]$, $\epsilon \mapsto Z_s^\epsilon$ is $\mathcal{C}^1$. Introduce the first variation

$$\delta Z_s = \left. \frac{d}{d\varepsilon} Z_s^\varepsilon \right|_{\varepsilon=0} =: \Delta_s,$$

which corresponds to the Gateaux derivative of $\theta' \mapsto Z_s(\theta')$ at $\theta$ in the direction $h = \delta\theta$. We now show that $\Delta_s$ satisfies the following inhomogeneous ODE:

$$\partial_s \Delta_s = \mathcal{A}_s \Delta_s + \delta\mathcal{A}_s\, Z_s, \qquad \Delta_0 = 0, \qquad \forall s \in [0,1]. \tag{52}$$

where $\delta\mathcal{A}_s := \left. \frac{d}{d\varepsilon} \mathcal{A}_s^\varepsilon \right|_{\varepsilon=0}$.

Indeed let us consider $q_s^\varepsilon := \frac{Z_s^\varepsilon - Z_s}{\varepsilon}$ for any $0 < \varepsilon \leq 1$. In particular one has $q_s^\varepsilon \xrightarrow[\varepsilon \to 0]{} \Delta_s$. Moreover one has:

$$q_s^\varepsilon = \varepsilon^{-1} \int_0^s (\mathcal{A}_u^\varepsilon Z_u^\varepsilon - \mathcal{A}_u Z_u)\mathrm{d}u = \int_0^s B_u^\epsilon Z_u \mathrm{d}u + \int_0^s \mathcal{A}_u^\varepsilon q_u^\varepsilon \mathrm{d}u, \tag{53}$$

where $B_u^\epsilon := \frac{\mathcal{A}_u^\varepsilon - \mathcal{A}_u}{\varepsilon} = \frac{\mathcal{A}_u^\varepsilon - \mathcal{A}_u^0}{\varepsilon}$ satisfies $B_u^\varepsilon \xrightarrow[\varepsilon \to 0]{} \left. \frac{d}{d\varepsilon} \mathcal{A}_u^\varepsilon \right|_{\varepsilon=0} = \delta\mathcal{A}_u$ (as $\epsilon \mapsto \mathcal{A}_u^\varepsilon$ is $\mathcal{C}^1$) and where $\varepsilon \in [0,1] \mapsto B_u^\varepsilon$ is continuous (at 0, we define $B_u^0 = \delta\mathcal{A}_u$) as $\varepsilon \mapsto \mathcal{A}_u^\varepsilon$ is $\mathcal{C}^1$), and thus is bounded on $[0,1]$ by a constant that does not depend on $\varepsilon$. By dominated convergence, when $\varepsilon \to 0$ in equation 53 one obtains the limit:

$$\Delta_s = \int_0^s (\mathcal{A}_u \Delta_u + \delta\mathcal{A}_u\, Z_u)\mathrm{d}u,$$

which coincides with the unique solution of equation 52.

Since $Z_s$ is a solution for the homogeneous part $\partial_s Z_s = \mathcal{A}_s Z_s$ with $Z_0 = \mathrm{Id}_n$, by the variation-of-parameters method, one obtains (as $\Delta_0 = 0$):

$$\Delta_s = Z_s \int_0^s Z_\tau^{-1}\, \delta\mathcal{A}_\tau\, Z_\tau\, d\tau.$$

Evaluating at $s = 1$ gives

$$\delta Z_1 = \Delta_1 = \int_0^1 Z_1 Z_\tau^{-1}\, \delta\mathcal{A}_\tau\, Z_\tau\, d\tau. \tag{54}$$

*2d step: Differential of $\ell$ and identification of the gradient.*

Because $f \in \mathcal{C}^1(\mathbb{R}^{n \times n}, \mathbb{R})$ its (Fréchet) differential at $M \in \mathbb{R}^{n \times n}$ is

$$Df(M)[H] = \langle \nabla f(M), H \rangle_F, \qquad \forall H \in \mathbb{R}^{n \times n}. \tag{55}$$

Applying the chain rule to $\ell = f \circ Z_1$ with the Gateaux differentials $D_G$ at $\theta$ and in the direction $h = \delta\theta$, one obtains,

$$D_G \ell(\theta)[\delta\theta] = D_G f\big(Z_1(\theta)\big)\big[\delta Z_1\big].$$

But as by hypothesis both $\ell$ and $f$ are Frechet differentiable, one has:

$$D\ell(\theta)[\delta\theta] = Df\big(Z_1(\theta)\big)\big[\delta Z_1\big].$$

Using equation 55 with $M = Z_1(\theta)$ and $H = \delta Z_1$,

$$D\ell(\theta)[\delta\theta] = \big\langle \nabla f\big(Z_1(\theta)\big), \delta Z_1 \big\rangle_F. \tag{56}$$

By inserting the expression of $\delta Z_1$ from equation 54 into equation 56, one has:

$$D\ell(\theta)[\delta\theta] = \int_0^1 \mathrm{tr}\Big(\nabla f(Z_1)^\top Z_1 Z_\tau^{-1}\, \delta\mathcal{A}_\tau\, Z_\tau\Big)\, d\tau.$$

Because $\operatorname{tr}(RS) = \operatorname{tr}(SR)$, one get

$$\operatorname{tr}\big(\nabla f(Z_1)^\top Z_1 Z_\tau^{-1} \delta \mathcal{A}_\tau Z_\tau\big) = \operatorname{tr}\big(Z_\tau \nabla f(Z_1)^\top Z_1 Z_\tau^{-1} \delta \mathcal{A}_\tau\big),$$

and thus by defining for each $\tau$

$$G_\tau^\top := Z_\tau \nabla f(Z_1)^\top Z_1 Z_\tau^{-1}, \tag{57}$$

one finally has

$$D\ell(\theta)[\delta\theta] = \int_0^1 \operatorname{tr}\big(G_\tau^\top \delta A_\tau\big)\, d\tau = \langle G, \delta\theta \rangle_{L^2}.$$

By Riesz theorem, the *gradient in* $L^2$ (i.e. the Fréchet gradient) is the unique element $G \in L^2$ verifying $D\ell(\theta)[\delta\theta] = \langle G, \delta\theta \rangle_{L^2}$ for every $\delta\theta$:

$$\nabla \ell(\theta) = G.$$

The transpose in equation 57 finally yields the required formula. $\qquad\square$

### L.3  LINK WITH CONSERVATION LAWS IN FINITE DEPTH (INFORMAL).

We assume that $\theta_L := (U_L, \cdots, U_1)$ satisfies the relaxed balanced conditions:

$$U_{i+1}^\top U_{i+1} - U_i U_i^\top = \frac{H_i}{L^2},$$

then as $U_k = \mathrm{Id} + \frac{1}{L} A_k$, using that $A_{k+1} = A_k + \frac{1}{L} A_k' + o\left(\frac{1}{L}\right)$ we get:

$$
\begin{aligned}
\frac{H_k}{L^2} &= (\mathrm{Id} + \frac{1}{L} A_{k+1})^\top (\mathrm{Id} + \frac{1}{L} A_{k+1}) - (\mathrm{Id} + \frac{1}{L} A_k)(\mathrm{Id} + \frac{1}{L} A_k^\top) \\
&= \frac{A_{k+1}^\top + A_{k+1}}{L} - \frac{A_k + A_k^\top}{L} - \frac{A_k A_k^\top}{L^2} + \frac{A_{k+1}^\top A_{k+1}}{L^2} \\
&= \frac{A_k'^\top + A_k' - A_k A_k^\top + A_k^\top A_k}{L^2} + o\left(\frac{1}{L^2}\right) \\
&= \frac{h_{s_k}(\theta)}{L^2} + o\left(\frac{1}{L^2}\right),
\end{aligned}
$$

Thus $h_s$ is such that $h_{s_k}(\theta) = H_k + o(1)$.

In particular if $\theta_L$ satisfies the quasi balanced condition

$$U_{i+1}^\top U_{i+1} - U_i U_i^\top = \frac{\lambda_i}{L^2} \mathrm{Id},$$

then one can choose $h_s$ as:

$$h_s(\theta) = \lambda(s) \mathrm{Id}_n,$$

with $\lambda$ a function such that $\lambda(s_k) = \lambda_k$. We say in that case that $\theta$ satisfies the relaxed balanced condition.

## M  PROOF OF THEOREM 3.13.

### M.1  PROOF OF THE THEOREM

**Theorem 3.13.** *If the initialization $\theta(0)$ satisfies that for each $s \in [0,1]$ $\mathbf{h}_s(\theta(0)) = \lambda(s)\mathrm{Id}_n$ for some $\lambda(\cdot) \in \mathcal{C}^0([0,1], \mathbb{R})$, then one has*

$$\dot{Z}_1 = -\int_0^1 (Z_1 Z_1^\top)^{1-s} \exp(\gamma(s)) \nabla f(Z_1)(Z_1^\top Z_1)^s \mathrm{d}s,$$

*with $\gamma(s) := (1-s)\psi_1(1) - \psi_1(1-s) - s\psi_2(1) + \psi_2(s)$, where $\psi_1 : s \in [0,1] \mapsto \int_0^s \int_0^u \lambda(1-v)\mathrm{d}v\mathrm{d}u$ and $\psi_2 : s \in [0,1] \mapsto \int_0^s \int_0^u \lambda(v)\mathrm{d}v\mathrm{d}u$. If $\lambda(\cdot) \equiv 0$ (balanced-condition), then $\gamma(\cdot) \equiv 0$.*

*Proof.* For any $t$ and any integer $L \geq 1$ we define $s_k := s_k^L = \frac{k}{L}$ for $k = 0, \cdots, L-1$ and:

$$X_{k+1}(t) = X_k(t) + h\mathcal{A}_{s_k}(t)X_k(t), \quad \text{with } h := \frac{1}{L} \text{ and } X_0(t) = I_n.$$

Since this corresponds exactly to the Euler explicit method with step $h$ for the ODE

$$\partial_s Z_s(t) = \mathcal{A}_s(t)Z_s(t), \quad Z_0(t) = \mathrm{Id}_n, \quad s \in [0,1],$$

one has for any $t$ and $L$ (computations postponed in Section M.2):

$$\sup_{0 \leq k \leq L-1} \|X_k(t) - Z_{s_k}(t)\| = \mathcal{O}(h) \tag{58}$$

$$\|\partial_t Z_1(t) - \partial_t X_L(t)\| = \mathcal{O}(h), \tag{59}$$

with $\|\cdot\|$ any matrix norm on $\mathbb{R}^{n \times n}$, the implicit constant in the notation $\mathcal{O}(\cdot)$ is independent of $k$ and $L$ while it can depend on $t$.

We now fix some $t$ and observe that $X_{k+1}(t) = U_k(t)X_k(t)$ with

$$U_k(t) := \mathrm{Id}_n + h\mathcal{A}_{s_k}(t). \tag{60}$$

so that (from now on we drop the $t$ variable for brevity)

$$\partial_t X_L = h \sum_{j=0}^{L-1} (U_{L-1}\cdots U_{j+1})(\partial_t \mathcal{A}_{s_j})X_j(t) = h\sum_{j=0}^{L-1}(U_{L-1}\cdots U_{j+1})(\partial_t \mathcal{A}_{s_j})U_{j-1}\cdots U_0$$

By equation 15 and the relation equation 48 (shown in Section L.2) we have for any $s \in [0,1]$

$$\partial_t \mathcal{A}_s = -\mathfrak{g}_s = -(Z_s^{-1})^\top Z_1^\top \nabla f(Z_1)Z_s^\top$$

hence

$$\partial_t X_L = -h\sum_{j=0}^{L-1}(U_{L-1}\cdots U_{j+1})(Z_{s_j}^{-1})^\top Z_1^\top \nabla f(Z_1)Z_{s_j}^\top U_{j-1}\cdots U_0$$

As $U_{L-1}\cdots U_{j+1} = X_L X_{j+1}^{-1} = (Z_1 + \mathcal{O}(h))(Z_{s_{j+1}} + \mathcal{O}(h))^{-1} = Z_1 Z_{s_{j+1}}^{-1} + \mathcal{O}(h)$ since the invertibility of $Z_s$ and continuity of $s \mapsto Z_s$ implies that $\|Z_s^{-1}\|$ is uniformly bounded) and $Z_{s_{j+1}}Z_{s_j}^{-1} = \mathrm{Id}_n + \mathcal{O}(h)$ (since $Z_{s_{j+1}} = Z_{s_j} + h\mathcal{A}_{s_j}Z_{s_j} + \mathcal{O}(h^2)$), we deduce that

$$\begin{aligned}
(Z_{s_j}^{-1})^\top Z_1^\top &= (Z_1 Z_{s_j}^{-1})^\top = [(Z_1 Z_{s_{j+1}}^{-1})Z_{s_{j+1}}Z_{s_j}^{-1}]^\top \\
&= [(U_{L-1}\cdots U_{j+1} + \mathcal{O}(h))(I_n + \mathcal{O}(h))]^\top \\
&= (U_{L-1}\cdots U_{j+1})^\top + \mathcal{O}(h),
\end{aligned}$$

where in the last line we used that since with any relevant matrix norm since $\max_k \|U_k\| = 1 + \mathcal{O}(h) = 1 + \mathcal{O}(1/L)$ we have $\|U_{L-1}\ldots U_j\| \leq [1 + \mathcal{O}(1/L)]^L = \mathcal{O}(1)$. Similarly we also have $\|U_{j-1}\ldots U_0\| = \mathcal{O}(1)$ hence

$$\partial_t X_L = -h\sum_{j=0}^{L-1}(U_{L-1}\cdots U_{j+1})(U_{L-1}\cdots U_{j+1})^\top \nabla f(Z_1)Z_{s_j}^\top U_{j-1}\cdots U_0 + \underbrace{h\sum_{j=0}^{L-1}\mathcal{O}(h)}_{=\mathcal{O}(h) \text{ since } h=1/L}$$

Similarly as $U_{j-1}\cdots U_0 = X_j = Z_{s_j} + \mathcal{O}(h)$ by equation 58, we get $Z_{s_j}^\top = (U_{j-1}\cdots U_0)^\top + \mathcal{O}(h)$ hence

$$\begin{aligned}
\partial_t X_L &= -h\sum_{j=0}^{L-1}(U_{L-1}\cdots U_{j+1})(U_{L-1}\cdots U_{j+1})^\top \nabla f(Z_1)(U_{j-1}\cdots U_0 + \mathcal{O}(h))^\top (U_{j-1}\cdots U_0) \\
&= -h\sum_{j=0}^{L-1}(U_{L-1}\cdots U_{j+1})(U_{L-1}\cdots U_{j+1})^\top \nabla f(Z_1)(U_{j-1}\cdots U_0)^\top (U_{j-1}\cdots U_0) + \mathcal{O}(h)
\end{aligned}$$

$$\tag{61}$$

We also have

$$
\begin{aligned}
U_{k+1}^\top U_{k+1} - U_k U_k^\top &= (\mathrm{Id}_n + h\mathcal{A}_{s_{k+1}})^\top (\mathrm{Id}_n + h\mathcal{A}_{s_{k+1}}) - (\mathrm{Id}_n + h\mathcal{A}_{s_k})(\mathrm{Id}_n + h\mathcal{A}_{s_k}^\top) \\
&= h(\mathcal{A}_{s_{k+1}}^\top + \mathcal{A}_{s_{k+1}}) - h(\mathcal{A}_{s_k} + \mathcal{A}_{s_k}^\top) - h^2(\mathcal{A}_{s_k}\mathcal{A}_{s_k}^\top) + h^2(\mathcal{A}_{s_{k+1}}^\top \mathcal{A}_{s_{k+1}}) \\
&= h^2(\mathcal{A}_{s_k}'^\top + \mathcal{A}_{s_k}' - \mathcal{A}_{s_k}\mathcal{A}_{s_k}^\top + \mathcal{A}_{s_k}^\top \mathcal{A}_{s_k}) + o\left(h^2\right) \\
&= h^2 \mathbf{h}_{s_k}(\theta) + o\left(h^2\right) \\
&= h^2 \lambda(s_k)\mathrm{Id}_n + o\left(h^2\right),
\end{aligned}
\tag{62}
$$

as $\theta(0)$ satisfies the quasi-balanced condition and $\mathcal{A}_{s_{k+1}} = \mathcal{A}_{s_k} + h\mathcal{A}_{s_k}' + o(h)$, where the implicit $o(1)$ function in the notation $o(h^2) = h^2 o(1)$ is still independent of $k$ and $L$ as $s \in [0,1] \mapsto \mathcal{A}_s'$ is continuous on the compact set $[0,1]$ and is thus uniformly continuous.

By Lemma K.1 with $\lambda_k = \lambda(s_k)$ and one has:

$$
(U_{L-1} \cdots U_{j+1})(U_{L-1} \cdots U_{j+1})^\top = \prod_{k=0}^{L-1-(j+1)} (U_{L-1}U_{L-1}^\top - a_k \mathrm{Id}_n) + E_{L-1-j},
\tag{63}
$$

with

$$
a_0 := 0, \quad a_k = h^2 \sum_{i=1}^{k} \lambda(s_{L-1} - s_i) \text{ for } k \geq 1, \quad \text{and } \|E_{L-1-j}\| \leq K\eta
\tag{64}
$$

where $K := (C_0 \exp(C_1)\exp(C_0(C_1 + 1) + \exp(C_1))$ with $C_0 := 2C_\lambda$, $C_1 := (C_U^2 + \eta h^2 - 1)/h$,

$$
C_U := \max(1, \max_k \|U_k\|), \quad C_\lambda := \max_k |\lambda_k|
\tag{65}
$$

$$
\eta := L^2 \cdot \max_{0 \leq k \leq L-2} \|(U_{k+1}^\top U_{k+1} - U_k U_k^\top) - h^2 \lambda_k \, \mathrm{Id}_n\|.
\tag{66}
$$

As $\lambda(\cdot)$ is continuous, $C_\lambda \leq \|\lambda\|_\infty < \infty$ for any $L$. Similarly, we already used that as $s \in [0,1] \mapsto \mathcal{A}_s$ is continuous, $C_U = 1 + \mathcal{O}(h)$, and thus $C_U^2 = 1 + \mathcal{O}(h)$, again with implicit constant independent of $L$. Moreover by equation 62, $\eta h^2 = \eta/L^2 = o(h^2)$, and we obtain $C_1 = (C_U^2 + \eta h^2 - 1)/h = (\mathcal{O}(h) + o(h^2))/h = \mathcal{O}(1)$, hence $C_1$ is bounded uniformly. Finally we obtain

$$
\max_j E_{L-1-j} = o(1)
\tag{67}
$$

where the implicit function $o(1)$ is still independent of $L$.

We denote

$$
F_i(U_{L-1}U_{L-1}^\top) := \prod_{k=0}^{i} (U_{L-1}U_{L-1}^\top - a_k \mathrm{Id}_n)
\tag{68}
$$

and use the shorthand $A_k := A_k^L(t) := \mathcal{A}_{s_k}(t) \in \mathbb{R}^{n \times n}$, for $0 \leq k \leq L-1$. Since $U_k = \mathrm{Id}_n + hA_k$ with $h = 1/L$ and $s_{L-1} - s_i = \frac{L-1}{L} - \frac{i}{L} = 1 - \frac{i+1}{L}$ for each integer $i$, we have (using Riemann

integration as $\lambda(\cdot)$ is continuous): since all the matrices in the product equation 68 commute

$$F_j(U_{L-1}U_{L-1}^\top) = \exp\left(\sum_{k=0}^{j} \log(U_{L-1}U_{L-1}^\top - a_k\mathrm{Id}_n)\right)$$

$$\overset{(60)-(64)}{=} \exp\left(\sum_{k=0}^{j} \log\left(\mathrm{Id}_n + \frac{A_{L-1}+A_{L-1}^\top}{L} + o\left(\frac{1}{L}\right) - \frac{1}{L}\left(\underbrace{\frac{1}{L}\sum_{i=1}^{k}\lambda(1-\tfrac{i+1}{L})}_{=\int_0^{s_k}\lambda(1-v)\mathrm{d}v+o(1)}\right)\mathrm{Id}_n\right)\right)$$

$$= \exp\left(\sum_{k=0}^{j}\left(\frac{A_{L-1}+A_{L-1}^\top}{L} - \frac{1}{L}\int_0^{s_k}\lambda(1-v)\mathrm{d}v\cdot\mathrm{Id}_n + o\left(\frac{1}{L}\right)\right)\right)$$

$$= \exp\left(s_j(\mathcal{A}_1+\mathcal{A}_1^\top) - \underbrace{\frac{1}{L}\sum_{k=0}^{j}\int_0^{s_k}\lambda(1-v)\mathrm{d}v}_{=\int_0^{s_j}\int_0^{u}\lambda(1-v)\mathrm{d}v\mathrm{d}u+o(1)}\cdot\mathrm{Id}_n + o(1)\right).$$

We denote $\psi_1 : s \in [0,1] \mapsto \int_0^s \int_0^u \lambda(1-v)\mathrm{d}v\mathrm{d}u$. By equation 63 and the above derivations one has

$$Z_1 Z_1^\top = \lim_{L\to+\infty} X_L X_L^\top = \lim_{L\to+\infty} F_{L-1}(U_{L-1}U_{L-1}^\top) = \exp((\mathcal{A}_1+\mathcal{A}_1^\top) - \psi_1(1)\cdot\mathrm{Id}_n),$$

and thus

$$\mathcal{A}_1 + \mathcal{A}_1^\top = \log(Z_1 Z_1^\top) + \psi_1(1)\cdot\mathrm{Id}_n$$

Thus

$$F_j(U_{L-1}U_{L-1}^\top) = (Z_1 Z_1^\top)^{s_j}\exp\left(s_j\psi_1(1) - \psi_1(s_j)+o(1)\right). \tag{69}$$

Similarly as before and by adapting the proof of Lemma K.1 one gets:

$$(U_{j-1}\cdots U_0)^\top(U_{j-1}\cdots U_0) = \prod_{k=0}^{j-1}(U_0^\top U_0 - b_k\mathrm{Id}_n) + o(1) =: G_j(U_0^\top U_0) + o(1) \tag{70}$$

with $b_k = -h^2\sum_{i=0}^{k-1}\lambda(s_i)$ and $b_0 = 0$, and where we denote

$$G_j(U_0^\top U_0) := \prod_{k=0}^{j-1}(U_0^\top U_0 - b_k\mathrm{Id}_n) \tag{71}$$

Similarly as above:

$$G_{j+1}(U_0^\top U_0) = \exp\left(\sum_{k=0}^{j}\log(U_0^\top U_0 - b_k\mathrm{Id}_n)\right)$$

$$= \exp\left(\sum_{k=0}^{j}\log\left(\mathrm{Id}_n + \frac{A_0+A_0^\top}{L} + o\left(\frac{1}{L}\right) + \frac{1}{L}\left(\underbrace{\frac{1}{L}\sum_{i=0}^{k-1}\lambda(s_i)}_{=\int_0^{s_k}\lambda(v)\mathrm{d}v+o(1)}\right)\mathrm{Id}_n\right)\right)$$

$$= \exp\left(\sum_{k=0}^{j}\left(\frac{A_0+A_0^\top}{L} + \frac{1}{L}\int_0^{s_k}\lambda(v)\mathrm{d}v\cdot\mathrm{Id}_n + o\left(\frac{1}{L}\right)\right)\right)$$

$$= \exp\left(s_j(\mathcal{A}_0+\mathcal{A}_0^\top) + \underbrace{\frac{1}{L}\sum_{k=0}^{j}\int_0^{s_k}\lambda(v)\mathrm{d}v}_{=\int_0^{s_j}\int_0^{u}\lambda(v)\mathrm{d}v\mathrm{d}u+o(1)}\cdot\mathrm{Id}_n + o(1)\right).$$

We denote $\psi_2 : s \in [0,1] \mapsto \int_0^s \int_0^u \lambda(v) \mathrm{d}v \mathrm{d}u$. By equation 70 and the above derivations we have

$$Z_1^\top Z_1 = \lim_{L \to +\infty} X_L^\top X_L = \lim_{L \to +\infty} G_L(U_0^\top U_0) = \exp((\mathcal{A}_0 + \mathcal{A}_0^\top) + \psi_2(1) \cdot \mathrm{Id}_n),$$

and thus

$$\mathcal{A}_0 + \mathcal{A}_0^\top = \log(Z_1^\top Z_1) - \psi_2(1) \cdot \mathrm{Id}_n$$

It follows that

$$G_{j+1}(U_0^\top U_0) = (Z_1^\top Z_1)^{s_j} \exp(-s_j \psi_2(1) + \psi_2(s_j) + o(1)). \tag{72}$$

Finally, combining equation 63-equation 67-equation 68-equation 69 and equation 70-equation 71-equation 72, we obtain

$$\partial_t X_L \stackrel{(61)}{=} -h \sum_{j=0}^{L-1} (U_{L-1} \cdots U_{j+1})(U_{L-1} \cdots U_{j+1})^\top \nabla f(Z_1)(U_{j-1} \cdots U_0)^\top (U_{j-1} \cdots U_0) + \mathcal{O}(h)$$

$$= -h \sum_{j=0}^{L-1} \left[ (Z_1 Z_1^\top)^{1-s_j} \nabla f(Z_1)(Z_1^\top Z_1)^{s_j} \cdot \exp(\gamma(s)) + o(1) \right] + \mathcal{O}(h)$$

$$= -\int_0^1 (Z_1 Z_1^\top)^{1-s} \nabla f(Z_1)(Z_1^\top Z_1)^s \exp(\gamma(s)) \mathrm{d}s + o(1),$$

where

$$\gamma(s) := (1 - s)\psi_1(1) - \psi_1(1 - s) - s\psi_2(1) + \psi_2(s).$$

Thus one has:

$$\partial_t Z_1(t) = -\int_0^1 (Z_1 Z_1^\top)^{1-s} \exp(\gamma(s)) \nabla f(Z_1)(Z_1^\top Z_1)^s \mathrm{d}s,$$

which concludes the proof. $\qquad\square$

## M.2 PROOF OF EQUATION 58-EQUATION 59

We now show that equation 58-equation 59 hold for any $t$.

*Proof.* First we recall that

$$X_{k+1}(t) = X_k(t) + h\mathcal{A}_{s_k}(t)X_k(t),$$

with $X_0(t) = \mathrm{Id}_n$. This corresponds exactly to the Euler explicit formulation of the ODE:

$$\partial_s Z_s(t) = \mathcal{A}_s(t)Z_s(t), \quad Z_0(t) = \mathrm{Id}_n, \quad s \in [0,1] \tag{73}$$

with step $h = 1/L$.

We now show both items at once. We fix some $t$. Set

$$W(s) := \begin{pmatrix} Z_s \\ Y_s \end{pmatrix} \in \mathbb{R}^{2n \times n}, \qquad Y_s := \partial_t Z_s, \qquad W(0) = \begin{pmatrix} \mathrm{Id}_n \\ 0 \end{pmatrix},$$

so that

$$\frac{\mathrm{d}}{\mathrm{d}s} W(s) = \begin{pmatrix} \mathcal{A}_s(t) & 0 \\ \partial_t \mathcal{A}_s(t) & \mathcal{A}_s(t) \end{pmatrix} W(s). \tag{74}$$

The corresponding explicit-Euler discretization with step $h = 1/L$ reads

$$W_{k+1} = W_k + h \begin{pmatrix} \mathcal{A}_{s_k}(t) & 0 \\ \partial_t \mathcal{A}_{s_k}(t) & \mathcal{A}_{s_k}(t) \end{pmatrix} W_k, \tag{75}$$

which coincides component-wise with the recursions for $X_k$ and $T_k = \partial_t X_k$.

Because the right-hand side of equation 74 $(s, W) \mapsto \begin{pmatrix} \mathcal{A}_s(t) & 0 \\ \partial_t \mathcal{A}_s(t) & \mathcal{A}_s(t) \end{pmatrix} W$ is $\mathcal{C}^1$ (indeed both $s \mapsto \mathcal{A}_s(t)$ and $s \mapsto \partial_t \mathcal{A}_s(t)$ are $C^1$ (cf Theorem L.1) for each $t$), the Euler explicit scheme converges at order one (see e.g. (Berthelin, 2017, Proposition 10.30)):

$$\max_{0 \leq k \leq L} \left\| W(s_k) - W_k \right\| = \mathcal{O}(h).$$

In particular one get that for any $k$: $X_k(t) = Z_{s_k}(t) + O(h)$ and (reading the second bloc row at the final index $k = L$):

$$\left\| \partial_t Z_{s=1}(t) - \partial_t X_L(t) \right\| = \left\| Y_{s=1} - T_L \right\| = \mathcal{O}(h).$$

$\qquad\square$

# N    LLM USAGE

The authors of this paper used Large Language Models to aid and polish the writing of this paper and as a tool to make some of the proofs.

