# OpenReview forum: "Intrinsic training dynamics of deep neural networks"
_ICLR.cc/2026/Conference — ICLR 2026 Poster_

### Official Review · Reviewer_xeEd · 2025-10-30

**Soundness:** 4
**Presentation:** 3
**Contribution:** 3
**Rating:** 6
**Confidence:** 3

**Summary:**

This paper studied conditions under which the gradient flow in minimizing $\ell(\theta)$ can be rewritten as an autonomous flow in $z=\phi(\theta)$ for some reparametrization map $\phi$. Based on conservation laws, the authors derived and analyzed two sufficient conditions: the intrinsic metric property and the intrinsic recoverability property. They showed for general ReLU networks that, parameter points in a dense set admit an intrinsic flow in a neighborhood. For linear networks, they showed that the map from all weight matrices to the end-to-end matrix satisfies the intrinsic metric property under relaxed balanced initializations, but not in general. The authors derived explicit forms of the intrinsic flow for linear networks and an infinitely deep linear network with relaxed balanced initialization, and a depth-three scalar ReLU network.

**Strengths:**

The proposed framework provides a systematic way to study the intrinsic dynamics of gradient flow. It continues two lines of works:

* The paper considered reparameterized flows that are not necessarily mirror flows, thereby extending the framework in Li et al. (2022), and providing an intermediate step for applying tools developed in Azulay et al. (2021).

* It presented several descriptions of the intrinsic metric and intrinsic recoverability properties based on conservation laws, which I find natural and insightful. This connects with the framework in Marcotte et al. (2023).

I therefore believe this work is valuable for the community studying optimization dynamics and implicit bias.

Additionally, the intrinsic dynamics derived for the depth-three ReLU network are interesting and have not been identified in prior work, to my knowledge.

**Weaknesses:**

* The negative results for linear network are not strong to me. First, only the sufficient condition, intrinsic metric property, is declined, not the existence of intrinsic dynamic. Second, only a specific reparametrization map, $\phi_{\mathrm{lin}}$, is considered. It remains unclear whether $\phi_{\mathrm{lin}}$ induces an intrinsic flow, and if not, whether other reparameterization maps might.

* The intrinsic flow for ReLU networks is established only locally, which makes its utility unclear. Current analyses of implicit bias, even those not relying on mirror flows, still require the reparametrized flow to be globally defined, for example those in Azulay et al. (2021).

**Initial recommendation**

Despite the above points, I find this paper valuable to the community and my initial recommendation is acceptance.

**Questions:**

* Is Assumption 2.1 not satisfied by $\phi_{\mathrm{ReLU}}$? Consider $g(x,\theta)= u \sigma(v^\top x)=\mathbb{1}(v^\top x)uv^\top x$. When $\theta$ lies on {$v^\top x=0$}, $\phi_{\mathrm{ReLU}}(\theta)=u v^\top$ is not a reparametrization in any neighborhood of $\theta$, since the indicator function $\mathbb{1}(v^\top x)$ is not determined by $u v^\top$.

* How is the gradient flow solution defined for ReLU networks, for example, the "maximal solution" in Definition 2.6 and in Proposition 3.10? In particular, how is the non-differentiability of the ReLU activation at zero addressed? Does a gradient flow solution exist on $t\in[0,+\infty)$ for any initialization, and is the solution unique?

* (Following the previous question) Does Proposition 3.10 indicate that, regardless of how the derivative of ReLU at zero, $\sigma'(0)$, is defined, the flow in $Z$ is always well-defined on $\Theta$, and the definition of $\sigma'(0)$ only affects $\nabla f(Z)$ in the $Z$-flow?


**Minor suggestions**

* I find Equation (7) quite interesting. It would be helpful if the authors could provide a geometric interpretation of this condition, in particular an interpretation of the kernel of $\partial M$.

* In the abstract and the Contributions, there are claims similar to "For general ReLU networks, we show that, for any initialization, it is possible to rewrite the flow as an intrinsic dynamic." According to Corollary 3.9, I would suggest using "for a dense set of initializations" instead of "any initialization", as these two notions can differ a lot.

* It might make more sense to group Sections 2, 3.1, and 3.2 together as they concern the general framework, and group Sections 3.3 and 4 together as they concern specific networks.

---

> ### Author Response · Authors · 2025-11-24
> **(1/2)**
>
> We would like to thank the reviewer for the positive comments and constructive suggestions.
>
> > **W1a** The negative results for linear network are not strong to me. First, only the sufficient condition, intrinsic metric property, is declined, not the existence of intrinsic dynamic.
>
> This is a very good question! Actually, we can show via the Chow-Rashevskii theorem that in the considered linear case *a slightly stronger variant of the intrinsic dynamic property implies the intrinsic metric property*. We've updated the preprint (see page 21 for the proof). So the two properties are indeed almost equivalent (intrinsic metric is necessary and sufficient) in this case. As a consequence, *the negative result is actually about the (strongly) intrinsic dynamic*. This has been clarified.
>
> > **W1b** Second, only a specific reparametrization map is considered. It remains unclear whether  $\phi_{Lin}$  induces an intrinsic flow, and if not, whether other reparameterization maps might.
>
> a) a consequence of our answer to **W1a** is that at every initialization that is not relaxed balanced $\phi_{Lin}$ *does not* induce a flow with a (strongly) intrinsic dynamic.
>
> b) this impossibility result for  $z := \phi_{\text{Lin}}(\theta)$ easily extends to $u = \gamma(z)$  for any diffeomorphism $\gamma$.
>
> > **W2** The intrinsic flow for ReLU networks is established only locally, which makes its utility unclear. Current analyses of implicit bias, even those not relying on mirror flows, still require the reparametrized flow to be globally defined, for example those in Azulay et al. (2021).
>
> In fact, there is a set $\mathcal{Z}$ (an algebraic manifold, characterized in Appendix G) of measure zero such that the result is global as soon as the trajectory $\theta(t)$ stays away from $\mathcal{Z}$. Indeed, the reparametrization $\phi$ is defined globally (you can always factorize with $\phi$ as it captures rescaling invariances for any parameters, only $f$ depends on the activations). Thus, the matrix $M (\theta) = \partial \phi( \theta) \partial \phi(\theta)$ is also defined globally and drives the dynamics of $z = \phi(\theta)$ (even at points where $f$ is not differentiable, using chain rules with the selection Jacobian of $f$ as defined in https://arxiv.org/pdf/2006.02080). For ReLU networks, for all $\theta$ away from an algebraic manifold $\mathcal{Z}$ of measure zero, the function $\psi: \theta \mapsto (\mathbf{h}(\theta), \phi(\theta))$ is invertible (the important fact here is that *all conservation laws, as well as the reparametrization $\phi$, are defined globally* so that $\theta = \psi^{-1} (\mathbf{h}(\theta), z)$, and thus as soon as the trajectory remains away from this algebraic manifold one has $ \theta(t) = \psi^{-1} (\mathbf{h}(\theta_0), z(t))$ and thus $M(\theta(t))$ can be expressed only as a function of the trajectory of $z(t)$ and the init, which is the crucial object for implicit bias analysis that captures the associated potential.
>
>
> > **Q1** Is Assumption 2.1 not satisfied by $\phi_{ReLU}$?
>
> Thank you for raising this point. Actually, Assumption 2.1 *unnecessarily* required (2) to hold in the neighbourhood of every $\theta_0$, while *we only need it to hold in a given $\Omega$*, that may be loss-dependent (i.e, in ML, dataset-dependent). In the final version it is replaced by the assumption (on $\ell$, $\phi$ and $\Omega$) that there exists a $C^2$ function $f$ such that (2) holds on some given set $\Omega$. Example 2.3 has been updated with \Omega being defined (given some training collection $x_i$) as the set of parameters for which $v_j^\top x_i \neq 0$ for every $i,j$.
>
> > **Q2** how is the gradient flow solution defined for ReLU networks, for example, the "maximal solution" in Definition 2.6 and in Proposition 3.10? In particular, how is the non-differentiability of the ReLU activation at zero addressed? Does a gradient flow solution exist on $t\in [0, \infty)$,  for any initialization, and is the solution unique?
>
> With the new version of Assumption 2.1 “on a domain $\Omega$”, the maximal solution *on $\Omega$* is in the sense of the Cauchy-Lipschitz theorem *on the domain $\Omega.$* This has been slightly reworded for better clarity everywhere needed.
>
> NB: in our paper, we did not set out to engage with issues arising from the non-differentiability introduced by the ReLU activation. Instead, we deliberately reformulated all relevant expressions so as to bypass this difficulty altogether. That said, we are confident that our theoretical framework naturally extends to ODEs defined even at points where $f$ is non-differentiable. This can be handled through chain rules based on selection Jacobians (see our answer to **W2**), combined with existing results that guarantee the existence and uniqueness of continuous solutions to such non-smooth ODEs (page 14 of https://bolte.perso.math.cnrs.fr/Loja.pdf).

---

> > ### Author Response · Authors · 2025-11-24
> > **(2/2)**
> >
> > > **Q3** (Following the previous question) Does Proposition 3.10 indicate that, regardless of how the derivative of ReLU at zero, $\sigma’(0)$, is defined, the flow in  $Z$ is always well-defined on $\Theta$, and the definition of $\sigma’(0)$ only affects $\nabla f(Z)$ in the $Z$-flow?
> >
> > As hopefully clarified by our answer to **Q1**+**Q2**, the considered maximal solution is on the domain $\Omega$ where (the new) Assumption 2.1 is valid, which is dataset dependent. Doing so, we *don’t* need the existence / uniqueness of a solution *defined* via the gradient flow equation, in particular around times at which at least one ReLU activation crosses zero (such points do not belong to $\Omega$). Accordingly,  Proposition 3.10 (now Proposition 3.3) has been reworded purely in terms of intrinsic dynamics, without reference to “a maximal solution”.
> >
> > NB: as mentioned in our answer to **Q2**, our analysis could be extended to include nondifferentiable points of ReLU with only minor adaptations. One could work with the *original* Assumption 2.1 (so that the domain does not depend on the activation pattern, and is simply $\mathbb{R}^D$), assume $f$ is *piecewise* $C^2$ (which is precisely the setting of our new “strong intrinsic dynamic” property introduced in Appendix J page 19), and then consider the maximal solution of the corresponding differential inclusion. Using the strong intrinsic dynamic property, the flow in $z$ can then be derived exactly as in Proposition 3.10 (now Proposition 3.3) via chain rules applied to Jacobian selections. And indeed, this indicates that *regardless of how the derivative of ReLU at zero is defined, the flow in $Z$ is always well-defined on $\Theta$, and the definition of $\sigma’(0)$ only affects $\nabla f(Z)$ in the $Z$-flow*.
> >
> > **About minor suggestions**
> >
> > > I find Equation (7) quite interesting. It would be helpful if the authors could provide a geometric interpretation of this condition, in particular an interpretation of the kernel of $\partial M$
> >
> > Each kernel can be thought as the tangent space to the level sets of the corresponding function, so the left-hand-side of (7) is the intersection of two tangent spaces, thus the tangent space associated to the level sets of $M$ must contain them. We’ve addressed accordingly the other suggestions you’ve proposed, thank you!

---

> > > ### Comment · Reviewer_xeEd · 2025-11-28
> > >
> > > Thank you for the detailed responses! I acknowledge the improvements, particularly regarding the equivalence between the strong intrinsic dynamic property and the intrinsic metric property.
> > >
> > > I still have some questions/concerns about applying the framework to ReLU networks. I understand that addressing the non-differentiability of ReLU is not the main goal of this work. Though I wanted to better understand to what extent the proposed framework can be actually used to analyze GF in ReLU networks.
> > >
> > > Regarding the set $\mathcal{Z}$, if I understand correctly, the intrinsic flow is established only when the entire trajectory does not intersect $\mathcal{Z}$, i.e., it stays within a single connected component of $\Theta=\mathbb{R}^D \setminus \mathcal{Z}$. However, I do not expect that this holds in general practical training. I think that a measure zero set is negligible when talking about random initialization, but is not obviously negligible when talking about the entire trajectory. For example, in Proposition 3.3., staying in a single component of $\Theta$ means that, every coordinate of the parameter vector does not change its sign throughout the training, which appears to me a stringent condition.
> > >
> > > Regarding the non-differentiability, I am not convinced that the results in [1] are applicable (e.g., whether the condition $\mathcal{H}_1$ in page 13 is satisfied). In fact, according to [2], the ODE solutions under ReLU are not unique; see, [2], page 3: "Another important difficulty of this non-differentiability is that Cauchy-Lipschitz theorem does not apply and uniqueness is not ensured" and, page 18: "Because of the non-differentiability of the ReLU activation, the gradient flow is not uniquely defined". Is it true that the intrinsic flow is compatible with all admissible solutions (that are outside $\mathcal{Z}$)?
> > >
> > > I would also like to point out that the notions of algorithmic differentiation and Jacobian selection in [3] are not common in the analysis of ReLU networks. Clarke's sub-differential is more commonly used. These two notions are different as noted by the authors of [3].
> > >
> > >
> > > [1] [https://bolte.perso.math.cnrs.fr/Loja.pdf](https://bolte.perso.math.cnrs.fr/Loja.pdf)
> > >
> > > [2] Boursier, Etienne, Loucas Pillaud-Vivien, and Nicolas Flammarion. "Gradient flow dynamics of shallow relu networks for square loss and orthogonal inputs." Advances in Neural Information Processing Systems 35 (2022): 20105-20118.
> > >
> > > [3] [https://arxiv.org/pdf/2006.02080](https://arxiv.org/pdf/2006.02080)

---

> > > > ### Author Response · Authors · 2025-12-03
> > > >
> > > > We thank the reviewer for the discussion and their careful review. Although they can no longer reply, we address their last message here.
> > > >
> > > > > Regarding the set $\mathcal{Z}$, if I understand correctly, the intrinsic flow is established only when the entire trajectory does not intersect $\mathcal{Z}$, i.e., it stays within a single connected component of $\Theta=\mathbb{R}^D \setminus \mathcal{Z}$ (...)
> > > >
> > > > Indeed, even if a set $\mathcal{Z}$ has measure zero, it may still be large enough to force $\mathbb{R}^D \setminus \mathcal{Z}$ to have several connected components. While Proposition G.1 was designed to simply exhibit a measure zero (algebraic) set $\mathcal{Z}$ with the required property on $\text{dim} \text{Lie}(W_\phi (\theta))$, it is an interesting question to understand to what extent (for some DAG ReLU network architectures) the proposition can be refined to obtain a finer set with more controlled topological / dimensionality properties, so as to ensure for example that $\mathbb{R}^D \setminus \mathcal{Z}$ has a single connected component. This would require using finer structural properties $\phi_{ReLU}$ beyond its polynomial nature and goes beyond the scope of the present work.
> > > >
> > > > > "Regarding the non-differentiability, I am not convinced that the results in [1] are applicable (e.g., whether the condition $\mathcal{H}_1$  in page 13 is satisfied). In fact, according to [2], the ODE solutions under ReLU are not unique (...)  Is it true that the intrinsic flow is compatible with all admissible solutions (that are outside $\mathcal{Z}$)?" + “I would also like to point out that the notions of algorithmic differentiation and Jacobian selection in [3] are not common in the analysis of ReLU networks. Clarke's sub-differential is more commonly used. These two notions are different as noted by the authors of [3].”
> > > >
> > > > Thank you for pointing this out and for the reference. You are correct that the condition $\mathcal{H}_1$ is not satisfied in the ReLU setting, which indeed means that uniqueness of gradient-flow solutions is not guaranteed.
> > > > Nevertheless, the intrinsic flow remains compatible with any admissible solution of the differential inclusion.
> > > >  For a given solution, one may work with a conservative gradient selection along that trajectory, using the fact that the loss is path-differentiable in this setting, which ensures that the chain rule applies. Both Clarke’s subdifferential and the Jacobian selections in [3] provide valid conservative gradient selections for this purpose, even though they differ.
> > > >
> > > > Again, while we are reasonably confident that the intrinsic-flow reasoning extends to this non-smooth setting, a full discussion of such differential-inclusion questions is beyond the scope of the present work.

---

### Official Review · Reviewer_1Gdc · 2025-10-30

**Soundness:** 3
**Presentation:** 2
**Contribution:** 2
**Rating:** 4
**Confidence:** 3

**Summary:**

The goal of this paper is to find whether a high-dimensional gradient flow can be written as an intrinsic Riemannian flows in lower-dimensional spaces. The authors analyzed three related properties: intrinsic dynamics, metric, and recoverability. They then connect these properties: if we have iinstrinsic recoverability (connected with conservation laws), then the intrinsic metric property is satisfied, which then "verifies" the intrinsic dynamic property. The application to ReLU networks and that to linear networks are also discussed.

**Strengths:**

This paper has demonstrated a high level of mathematical rigor. The authors clearly define the involved properties formally, and use a series of lemmas and theorems to formally establish their connections. I find the results provided in Theorem 3.3 technically sound, which provides a viable way to investigate the possibility of studying the low-dimensional intrinsic Riemannian flow. The scope of the framework is somewhat general.

In addition, the application to deep linear networks (the relaxed balanced initialization compared to prior works) is interesting, which explains why this kind of initialization is necessary rather than only a clever or convenient choice.

**Weaknesses:**

1. However, in my view, this paper does not provide fundamentally new insight. The authors indeed broaden the scope of the connection between conservation laws and intrinsic metric for low-dimensional flow, however, this idea has been broadly discussed by prior works (e.g., Bah et al., (2022); Marcotte et al. (2023)) and the authors do not simplify the description. Hence, there lacks an adequate motivation for what this new framework can provide. As a result, I think the scope of contribution of this paper is limited (providing a new characterization for the path-lifting metric).

2. Another major weaknesses is the presentation. I think this paper is very challenging to read. It is too dense. The chain of definitions, lemmas, and theorems is very long, yet the main flow is constantly interrupted by examples and propositions that have already been discussed by other works. This is even more confusing when I find that the authors in fact have one separate section for examples. In addition, the link between high-level concepts is difficult to follow, and there lacks sufficient intuitive explanation for, e.g., why these high-level properties should be equivalent and can be connected.

**Questions:**

1. Can the authors provide some novel insights about neural network behavior that was unknown before applying this new framework?

2. One motivation of studying the intrinsic flow suggested by the authors is the connection to implicit bias. Then how do the results appeared in this paper inspire the study of the implicit bias? As one advantage of the framework is the relaxation of the requirement of the mirror flow representation, which in fact helps finding the implicit bias, I'm a bit confused about this question.

---

> ### Author Response · Authors · 2025-11-24
> **(1/2)**
>
> We would like to thank the reviewer for their feedback.
>
> > **W1** However, in my view, this paper does not provide fundamentally new insight. The authors indeed broaden the scope of the connection between conservation laws and intrinsic metric for low-dimensional flow, however, this idea has been broadly discussed by prior works (e.g., Bah et al., (2022); Marcotte et al. (2023)) and the authors do not simplify the description. Hence, there lacks an adequate motivation for what this new framework can provide. As a result, I think the scope of contribution of this paper is limited (providing a new characterization for the path-lifting metric).
>
> We respectfully but firmly disagree with this assessment. Bah et al. (2022) consider intrinsic dynamics *only for linear networks under a balancing condition* and do not address the general problem. Marcotte et al. (2023) focus on the related but different question of conservation laws/invariances, but only marginally study intrinsic dynamics. To the best of our knowledge, our work is the first to clearly formulate the general problem of intrinsic dynamical characterization and the first to resolve it in several practically relevant settings.
>
> We now clarify in detail how intrinsic dynamics are used (or not!) in the two cited papers:
>
> **Bah et al. (2022).** The authors use the known result from Arora et al. (2018) stating that *balanced initializations yield intrinsic dynamics*, and from this, they deduce that the rank of the product of weight matrices remains constant along training. This is the only place where intrinsic dynamics appear, and only in the balanced regime. The derivation of intrinsic dynamics from conservation laws is not discussed; the paper simply relies on the preexisting fact that balanced initializations guarantee this property. We stress that this clarification is not meant to diminish the value of their excellent contribution: their main result is a technically impressive global convergence theorem for two-layer linear networks under almost all initializations. Their argument does not rely on intrinsic dynamics - nor could it, since one implication of our results is that *for almost all initializations in linear networks, no intrinsic dynamics exist.*
>
> **Marcotte et al. (2023).** Marcotte et al. do establish a link between conservation laws and intrinsic dynamics, but only through the *intrinsic recoverability* condition. This condition is *assumed* (whereas we *prove it* for *arbitrary DAG ReLU* networks) and is somewhat too strong:
> - i) linear networks do not satisfy intrinsic recoverability, yet
> - ii) balanced initializations were already known (Arora et al., 2018) to yield intrinsic dynamics.
>
> Our work introduces the weaker intrinsic metric property, which turns out to be exactly the right structural notion: it provides the sharp and correct connection between conservation laws and intrinsic dynamics.
>
> Crucially, this finer property allows us to (a) identify a strictly larger class of initializations that admit intrinsic dynamics, and (b) show that in certain settings these are precisely the only initializations with this property. This demonstrates that the intrinsic metric property is the appropriate level of granularity for characterizing intrinsic dynamics.
>
> > **W2** Another major weaknesses is the presentation. I think this paper is very challenging to read. It is too dense. The chain of definitions, lemmas, and theorems is very long, yet the main flow is constantly interrupted by examples and propositions that have already been discussed by other works. This is even more confusing when I find that the authors in fact have one separate section for examples. In addition, the link between high-level concepts is difficult to follow, and there lacks sufficient intuitive explanation for, e.g., why these high-level properties should be equivalent and can be connected.
>
> As suggested by reviewer xeEd, we have reorganized the paper to improve readability: Sections 2, 3.1, and 3.2 are now grouped together as the general framework, and Sections 3.3 and 4 are grouped to present the two main examples (ReLU and linear networks).
>
> The examples and propositions from prior work that appear in the main text are included intentionally for pedagogical clarity: we believe it provides intuition and illustrates key notions precisely when they are introduced. We fear that removing them from the main flow would make the underlying concepts more difficult to understand.
>
> We believe that the revised structure makes the overall exposition significantly clearer.

---

> > ### Author Response · Authors · 2025-11-24
> > **(2/2)**
> >
> > > **Q1** Can the authors provide some novel insights about neural network behavior that was unknown before applying this new framework?
> >
> > A first insight is that, in the linear case with non-relaxed balanced initialization, our proof that no intrinsic dynamic is possible implies that it is worthless to conduct any mirror-like analysis. Our analysis also opens new research avenues to understand how the relaxed balanced parameters $\lambda$ concretely impact the geometry of the trajectories.
> >
> > In the ReLU case, while pre-existing qualitative analyses on feature learning/lazy/rich regimes were conducted in the two-layer case thanks to the corresponding known metric, our proof of the existence of an intrinsic metric in the general deep case also opens new avenues to investigate how the initialization impacts the corresponding geometries.
> >
> > We also refer the reviewer to our dedicated contribution section, which provides a complete summary.
> >
> > > **Q2** One motivation of studying the intrinsic flow suggested by the authors is the connection to implicit bias. Then how do the results appeared in this paper inspire the study of the implicit bias? As one advantage of the framework is the relaxation of the requirement of the mirror flow representation, which in fact helps finding the implicit bias, I'm a bit confused about this question.
> >
> > In fact, studying implicit bias is *one* motivation but *not* the main focus of this paper. Indeed, our results provide new tools that directly support such analyses, and they also show, importantly, that in some settings the standard approach to implicit bias *cannot* work (e.g. for non relaxed balanced initialization for linear networks). We believe that both enabling results and impossibility results are valuable to the community: the former open new avenues, while the latter clarify the fundamental limits of existing techniques.
> >
> > More concretely, current theoretical analyses of implicit bias systematically rely on an intrinsic flow. Nearly all existing works follow the same two-step strategy:
> > - i) rewrite the gradient flow on parameters $\theta$ as an intrinsic flow on $z=\phi(\theta)$;
> > - ii) show that this intrinsic flow is (exactly or approximately) a mirror flow, typically via a time-warped Hessian (Azulay et al., 2021).
> >
> > This strategy applies broadly, but *Step 1 had not been fully resolved prior to our work.* Our paper provides the missing Step 1 for both ReLU and linear architectures of any depth.
> >
> > Since intrinsic flows are a necessary ingredient in every known implicit-bias proof (e.g., Azulay et al., Li et al.), our results both (a) enable future implicit-bias analyses in settings where this was previously out of reach, and (b) establish that in some regimes the standard approach must fundamentally fail. We view both aspects as important contributions for the ML theory community.

---

> > > ### Comment · Reviewer_1Gdc · 2025-11-27
> > >
> > > I thank the authors for their detailed response. I've updated my score.

---

### Official Review · Reviewer_E9V6 · 2025-10-30

**Soundness:** 4
**Presentation:** 3
**Contribution:** 3
**Rating:** 6
**Confidence:** 4

**Summary:**

The paper defines a three increasingly stronger conditions that all imply the existence of intrinsic dynamics. These conditions require the knowledge of the set of invariants / conserved quantities, however a necessary and an equivalent condition that is more easily computable is given for two of these conditions.

The authors then use this framework to give a number of example of intrinsic dynamics, in linear networks and ReLU networks.

**Strengths:**

The paper presents a number of ideas that englobe many previous result in the same framework, and seems to also show that one cannot do better than these previous results (there are no more invariants than those already known).

Some of the intrinsic dynamics provided are to my knowledge new, though they are all very close to already known works.

**Weaknesses:**

This paper is a bit of a typical example of "proof by abstract non-sense", it mainly proves mainly already existing results using very abstract (and arguably quite complex) tools. I appreciate that this type of approach can tell us that there are not more invariants and therefore simplifications than the ones we already knew, but I am not convinced that I need all these tools to find the next invariants on a new model, because it seems that people have been able to identify these invariants naively without issues.

For linear networks, the relaxed balanced conditions has also been used in this paper https://arxiv.org/abs/2405.17580, which also proves that this condition arises naturally in the infinite width limit. The infinite depth linear network dynamics description is more novel, but it is very similar to the two layer case.

For ReLU networks, the invariants are just the differences between the norms of the incoming and outcoming weights, which were known for a while too.

**Questions:**

- Are there any model type where you believe that these techniques could discover some yet unknown invariants and lead to simplified dynamics?
- There are very few invariants in ReLU networks (the number of invariant is the squared root of the number of parameters), which suggests that there is no hope to just use invariants to obtain truly lower dimensional dynamics, do you still see hope? The biggest successes at finding simplified dynamics in DNNS is taking infinite width limits, e.g. NTK / mean-field limits. Can these fit in your framework?

---

> ### Author Response · Authors · 2025-11-24
> **(1/2)**
>
> We would like to thank the reviewer for their positive feedback.
>
> > **W1** This paper is a bit of a typical example of "proof by abstract non-sense", it mainly proves mainly already existing results using very abstract (and arguably quite complex) tools. I appreciate that this type of approach can tell us that there are not more invariants and therefore simplifications than the ones we already knew, but I am not convinced that I need all these tools to find the next invariants on a new model, because it seems that people have been able to identify these invariants naively without issues.
>
> By “invariants” are you referring to conservation laws? If yes, then there seems to be a misunderstanding: the question of whether there are more invariants than the ones already known for gradient flows was investigated (and solved) in Marcotte et al. 2023, using Lie bracket computations, and extended to other models by Marcotte et al. 2024 to find new conservation laws for NMF or ICNN.  Our paper rather uses conservation laws (whose study and analysis were completed in other papers) to address a different question: intrinsic dynamics.
>
> > **W2a** For linear networks, the relaxed balanced conditions has also been used in this paper https://arxiv.org/abs/2405.17580, which also proves that this condition arises naturally in the infinite width limit.
>
> We agree that the *balanced* condition is not new and arises naturally in the infinite-width setting. However, *relaxed balanced* appears to be new. It applies only when the number of hidden neurons is smaller than the output or input dimension ($r \leq \min(n, m)$, see our response to **W1** of Reviewer cmEp), so it cannot arise in the mean-field infinite-width setting. In particular, **relaxed balanced conditions are *not* studied in the given reference.**
>
> > **W2b** The infinite depth linear network dynamics description is more novel, but it is very similar to the two layer case.
>
> A key novelty of the infinite-depth linear network results is that the intrinsic metric admits an explicit closed-form expression. In contrast, for finite depth $n> 4$, no such closed form exists under relaxed balanced initialization: the intrinsic metric can only be computed algorithmically by finding the zeros of certain polynomials.
>
> Moreover:
>
> - both the study of intrinsic dynamics in finite-depth linear networks under relaxed balanced initialization (Theorem 4.6 → now Theorem 3.9) and the analysis of infinitely deep linear networks (Theorem 4.8 → now Theorem 3.11) are new contributions;
>
> - the structure of the intrinsic metrics is in fact *very different* from the two-layer case, both when comparing depth-2 with depth $L>2$, and when comparing finite depth $L>2$ with the infinite-depth regime. The infinite-depth setting exhibits qualitatively distinct behavior that cannot be inferred from the two-layer (nor $L>2$-layer)  analysis.
>
> > **W2c** "For ReLU networks, the invariants are just the differences between the norms of the incoming and outcoming weights, which were known for a while too" + **Q1** "Are there any model type where you believe that these techniques could discover some yet unknown invariants and lead to simplified dynamics?"
>
> As explained in our answer to **W1**, our contribution lies in using conservation laws (invariants) to characterize intrinsic dynamics - not in establishing these invariants themselves. This perspective allows us to derive **new simplified intrinsic *dynamics* that were previously unknown**, including:
> - (i) implicit intrinsic dynamics for general ReLU networks with $L>2$  layers;
> - (ii) an explicit intrinsic dynamic for 3-layer ReLU networks;
> - (iii) an explicit intrinsic dynamic for 2-layer linear networks with $r < \min(n,m)$ under relaxed balanced initialization;
> - (iv) implicit (and almost explicit algorithmically) intrinsic dynamics for $L>2$-layer linear networks under relaxed balanced initialization;
> - (v) an explicit intrinsic dynamic for infinitely deep linear networks.
>
> Importantly, knowing all conservation laws is **not** sufficient to guarantee the existence of an intrinsic dynamic. For instance, in 2-layer linear networks, the number of independent conservation laws is the same for *all* initializations, yet **only** relaxed-balanced initializations yield an intrinsic dynamic.
>
> One of the main goals of our paper is precisely to uncover the exact role played by conservation laws in enabling intrinsic dynamics, clarifying when intrinsic dynamics exist and when they provably cannot.

---

> > ### Author Response · Authors · 2025-11-24
> > **(2/2)**
> >
> > > **Q2a** There are very few invariants in ReLU networks (the number of invariant is the squared root of the number of parameters), which suggests that there is no hope to just use invariants to obtain truly lower dimensional dynamics, do you still see hope?
> >
> > Thank you for this comment. We agree that the dimension reduction in the intrinsic dynamics of ReLU networks is limited, but this is expected: additional conservation laws would overly constrain the dynamics to an even smaller manifold. Our work establishes the study of intrinsic dynamics, thereby paving the way for future analyses of implicit bias in ReLU networks and more general classes of linear-network initializations. The extent of the reduction is not a limiting factor for this objective. The main challenge ahead is to analyze the inverse of the resulting metric and determine whether it can be expressed as a *warped Hessian* (as in Azulay et al.), or under an even weaker formulation. We view this as an exciting direction for future work.
> >
> > > **Q2b** The biggest successes at finding simplified dynamics in DNNS is taking infinite width limits, e.g. NTK / mean-field limits. Can these fit in your framework?
> >
> > The invariances and reduced dynamics that we develop in our work persist in the mean field regime, interpreted as constraints on the support of the parameter distribution, whenever this notion is well defined. For two-layer networks, similar ideas already appear in earlier studies. In particular, Chizat and Bach (2018) make essential use of this structure in the ReLU case (often described as the “2-homogeneous setting”), where the training dynamics can be reformulated as an evolution on the sphere, a viewpoint that plays a key role in their convergence analysis. Although this aspect is not the main focus of our paper, we will briefly discuss it in the final version in order to point toward worthwhile research directions.

---

### Official Review · Reviewer_cmEp · 2025-11-03

**Soundness:** 4
**Presentation:** 4
**Contribution:** 3
**Rating:** 8
**Confidence:** 3

**Summary:**

This paper introduces a unifying theoretical framework to support implicit bias-type analyses of deep neural networks. The paper's goal is to establish when the gradient flow on the original parameters $\theta$ implies a corresponding flow on a lifted variable $z = \phi(\theta)$ depending on initialization $\theta(0)$ and the architecture --- this is defined formally through the "intrinsic dynamic" property. The authors define two further properties of the pair $\theta(0)$ and $\phi$, which can ensure intrinsic dynamics on $z$. These properties are related through a chain of implications, with necessary and/or sufficient conditions provided for parts of this chain to hold. Finally, the framework is applied to establish

 * that any initialization $\theta_0$ of DAG ReLU networks of arbitrary depth satisfies the intrinsic dynamic property with respect to the path-lifting $\phi$
 * that the relaxed balanced initialization for arbitrarily deep fully connected linear neural networks (LNN) with square weights satisfies the intrinsic dynamic property with respect to a commonly used reparametrization $\phi$. This result is extended to infinitely deep LNNs under the same initialization for which the dynamics $\dot{z}$ is made explicit.
 * the explicit dynamics of $\dot{z}$ for a 3 layer ReLU MLP.
 * the explicit dynamics of $\dot{z}$ for two-layer fully connected linear networks with relaxed balanced initialization

**Strengths:**

This paper makes a solid contribution to the line of work studying the implicit bias of NNs. The reasons are multifold:

* The proposed framework is novel and provides a unified approach for establishing that a reparametrization of the weights admits intrinsic dynamics, which is essential to implicit bias analyses. This result is very useful as, to my knowledge, existing literature relies on ad-hoc approaches that come with a difficult-to-track fine-grained variation in assumptions. While the difficulty of making $\dot{z}(t)$ explicit remains, this framework can provide shortcuts to validating potential architecture-initialization-reparametrization combinations.
* To the best of my knowledge, the related literature is appropriately covered. Connections to prior work are precise and detailed, and paint a clear picture of the current state of theoretical progress in this area.
* The proofs are presented clearly and, though I have not examined every detail, their main steps appear correct.
* The paper is well-structured and thoughtfully written, mathematically sound, and with a good balance of grounding examples among the dense theoretical results. The authors did a great job at condensing a high amount of information in a way that is easy to follow and pleasant to read.

**Weaknesses:**

A balancing discussion on the limitations of this framework / current results is mostly missing. For example,
 * What is the main difficulty in extending Theorem 4.3 to arbitrary $r$?
 * The ReLU architecture and $\phi_{\mathrm{ReLU}}$ pair seems to allow for stronger properties leading to intrinsic dynamics, i.e., intrinsic recoverability, compared to linear neural networks with relaxed balanced $\theta_0$ and $\phi_{\mathrm{Lin}}$. Is it coincidental, or does it point to a deeper-rooted limitation of the factorization $\phi_{\mathrm{Lin}}$?

**Questions:**

Please see the above section.

---

> ### Author Response · Authors · 2025-11-24
>
> We would like to thank the reviewer for the positive comments and constructive suggestions.
>
> > **W1** What is the main difficulty in extending Theorem 4.3 to arbitrary $r$?
>
> Thank you for raising this point. Since the main assumption of Theorem 4.3 (now Theorem 3.6) is the relaxed-balanced initialization assumption, it already handles arbitrary $r$: indeed, if $\theta_0$ is relaxed-balanced, i.e. if $U^\top U - V^\top V = \lambda I$, we deduce that:
> - either $\lambda = 0$, and the intrinsic dynamic property is already known (see Eq. 13).
> - or $\lambda > 0$, and $U^\top U$ must be positive definite, implying $r \leq n$;
> - or $\lambda < 0$, and $V^\top V$ must be positive definite, implying $r \leq m$.
>
> In the last two cases, we necessarily have $r \leq \min(n,m)$, which is precisely explicited in the Theorem.
> We added a comment after the Theorem for completeness.
>
> > **W2** The ReLU architecture and  $\phi_{\texttt{Lin}}$ pair seems to allow for stronger properties leading to intrinsic dynamics, i.e., intrinsic recoverability, compared to linear neural networks with relaxed balanced  $\theta_0$ and $\phi_{\texttt{Lin}}$. Is it coincidental, or does it point to a deeper-rooted limitation of the factorization?
>
> Did the reviewer mean the pair “ReLU architecture and $\phi_{\texttt{Relu}}$”? If so, we agree: intrinsic recoverability is a stronger property than intrinsic dynamic; further it holds essentially for every initialization. When the architecture is linear, both $\phi_{\texttt{Lin}}$  and $\phi_{\texttt{Relu}}$ are admissible factorizations. The second one always leads to an intrinsic dynamic, the first one is somewhat more natural and lower-dimensional, but requires relaxed balancedness to lead to intrinsic dynamics. The difference comes from the fact that $\phi_{\texttt{Lin}}$ is not involutive.

---

### Comment · Area_Chair_K6EW · 2025-11-17
**Some additional points**

I have looked at the paper and skimmed through the reviews. I believe, in addition to what the reviewers have mentioned, the authors should address the following problems (that are related to each other)

1. I think the authors ought to discuss their limitations more. The whole work is about gradient flow, but in reality, we never have GF. The authors need to discuss the range of applicability of their theory
2. I think the following work should be discussed in some depth when addressing point 1, which essentially shows that most of the conservation laws discussed in this submission are broken in interesting ways: arxiv.org/abs/2402.07193
3. I think it may be a good idea to adapt the title a bit. For example, I think it is more appropriate to add the qualifier "...under gradient flow" to the title to accurately reflect the scope of the paper. Of course, the authors are free to propose alternatives, but I believe something related to "gradient flow" should be clear from the title


AC

---

> ### Author Response · Authors · 2025-11-24
>
> We would like to thank the AC for their suggestions.
>
> > **Q1** I think the authors ought to discuss their limitations more. The whole work is about gradient flow, but in reality, we never have GF. The authors need to discuss the range of applicability of their theory
>
> Thank you for the comment. We acknowledge that restricting our analysis to the gradient-flow (GF) setting is a limitation. However, this is not a specific assumption of our approach: the GF regime is the standard framework in which intrinsic dynamics are studied (both in works on implicit bias and in studies of the dynamics themselves), and to our knowledge, all prior works in this line of research operate in this GF setting (Bah et al. 2022, Li et al. 2022, Marcotte et al. 2023, Arora et al. 2018, Achour et al. 2025, Azulay et al. 2021 etc.)
>
> Extending these analyses to the gradient-descent (GD) setting is significantly more challenging, as the discrete dynamics do not lend themselves to the same tractable continuous-time tools. A promising direction would be to build on recent approaches based on quasi-conserved quantities (e.g., Marcotte et al. 2025), but adapting such techniques in our setting is non-trivial and falls outside the scope of this work.
>
> Going beyond GD to optimizers with momentum, natural-gradient–based methods, or weight decay is also an interesting research direction. The analysis of conservation laws for such optimization methods has been investigated in Marcotte et al. (2024, 2025), and a key challenge for future work would be to adapt the current analysis of intrinsic dynamics to these more complex training regimes.
>
> > **Q2** I think the following work should be discussed in some depth when addressing point 1, which essentially shows that most of the conservation laws discussed in this submission are broken in interesting ways: arxiv.org/abs/2402.07193
>
> We thank the AC for pointing out this relevant prior work, which we will discuss in the final version. It studies a different type of dynamics, closely related to Langevin dynamics, where the injected noise is independent of the data. This breaks the invariances of the model and explains why one cannot expect intrinsic dynamics to emerge in that setting. Although this lies beyond the scope of our paper, we note that SGD, in contrast, approximately preserves the conservation laws (see Marcotte et al. 2025). This suggests that approximate intrinsic dynamics could be investigated in future work.
>
> > **Q3** I think it may be a good idea to adapt the title a bit. For example, I think it is more appropriate to add the qualifier "...under gradient flow" to the title to accurately reflect the scope of the paper. Of course, the authors are free to propose alternatives, but I believe something related to "gradient flow" should be clear from the title
>
> We appreciate the suggestion. While we agree that explicitly mentioning gradient flow helps clarify the scope, we would prefer to keep a short and readable title, since we believe the abstract states clearly that our analysis is carried out entirely in the gradient-flow setting.

---

### Meta-Review · Area_Chair_62BX · 2026-01-07

**Summary:**

The article studies how gradient training could lead to parameters in lower dimensional sets.

The reviewers are generally positive about this work, praising it as a valuable / solid contribution to implicit bias discussion, presenting ideas that englobe many previous results, having a high level of mathematical rigor.

Therefore I recommend accept.

**Reviewer Concerns:**

Reviewer 1Gdc:
in my view, this paper does not provide fundamentally new insight.
-> Response in rebuttal

Another major weaknesses is the presentation
-> Response in rebuttal

**Reviewer Scores:**

For each review, specify how you think the reviewer would have changed their score if they had been able to participate fully in the discussion.

Reviewer cmEp: 8 -> 8
Reviewer E9V6: 6 -> 6
Reviewer 1Gdc: 4 -> 6
Reviewer xeEd: 6 -> 6

---

### Decision · Program_Chairs · 2026-01-26

Accept (Poster)